# Cryptographic Rational Secret Sharing Schemes over General Networks

**Alfonso Labao *** and **Henry Adorna**

Department of Computer Science, University of the Philippines Diliman, Quezon City 1101, Philippines
* Correspondence: alfonso.labao@upd.edu.ph

**Abstract:** We propose cryptographic rational secret sharing protocols over general networks. In a general network, the dealer may not have direct connections to each player, and players may not have direct connections to each of the other players. We present conditions on the network topology for which our proposed protocols are computational strict Nash equilibria and $(k-1)$-resilient, along with analysis on their round and communication complexity. We also present new notions of equilibria such as $\Phi$-resilient computational Nash equilibria, whereby a protocol is resilient to coalitions that satisfy conditions in $\Phi$, regardless of the coalition's size. We also propose $(n-1)$-key leakage-tolerant equilibria applicable to cryptographic protocols involving secret keys, whereby the equilibrium holds even if some players acquire $(n-1)$ tuples of secret keys.

**Keywords:** rational secret sharing; algorithmic game theory; network security; protocol mechanism design

## 1. Introduction

Secret sharing schemes address the problem of securely disseminating a secret among several participants, which is a relatively old problem in cryptography. Perhaps the most popular early secret sharing scheme is the $(n,k)$ secret sharing scheme by [1], which is also termed as a $(n,k)$ threshold sharing scheme. In this secret sharing scheme, the setting involves a dealer who wants to share a secret among $n$ players. The dealer subdivides the secret into $n$ pieces (i.e., shares) and sends a piece to each player. If at least $k$ players cooperate and share their shares, then the secret can be efficiently reconstructed. However, if less than $k$ players cooperate, their shares reveal no information about the secret. To achieve these conditions, the scheme of [1] uses properties of polynomials and Lagrange interpolation, and it is shown to be secure under the formalized security notion of a secret sharing scheme [2]. Since this invention by [1], several other secret sharing schemes have been proposed [3], many of which are closely related to the field of secure multiparty computation [4–7].

The setting for standard $(n,k)$ secret sharing, however, assumes that players are either completely honest or malicious [8], and security is guaranteed against completely malicious players (termed adversaries). In a paper by [9], however, players are instead modeled as rational in the game-theoretic sense [10], i.e., players have associated utility functions, and the goal of each player is to maximize their own utility as a function of the game's outcome—while taking into account the effects of the actions of other players in determining the outcome of the game. It is shown in [9], that standard non-rational secret sharing schemes would fail to obtain the desired objective of having all players learn the secret if participants are modeled as rational under natural assumptions on their utility functions. Thus, non-rational protocols have to be modified in order to factor-in the utility-maximizing behavior of players and the widened action space that comes from rationality. This notion of a rational player by [9] paved the way for the research area of rational secret sharing, where solutions are expressed in the form of protocols that

induce Nash equilibria [11]. In particular, the rational secret sharing scheme in [9] is a protocol where players have an incentive to follow the protocol and learn the secret together, rather than for a player to deviate from the protocol and learn the secret by itself. In this regard, Ref. [9] showed that their scheme is not only a Nash equilibrium but is also not weakly dominated [11], which, in some instances, involves a stronger condition than Nash equilbrium. Moreover, [9] showed that no rational secret sharing scheme exists for $n = 2$ players, but such a scheme exists for $n > 2$ by taking advantage of randomness and uncertainty over the game's outcome. Several other papers on rational secret sharing followed after [9]. The scheme of [12] is a simple rational secret sharing scheme that allows the dealer to either draw a true secret from some subset of a field, or draw a false secret—which is a simplification from the original protocol of [9]. This random drawing by the dealer gives uncertainty in players' point of view, such that for the players, the more viable and less risky option is to comply with the protocol. Another paper by [13] considers the dependence of schemes on various notions of utility. The chapter of [14] claims that rational secret sharing contributed a new notion of equilibrium to the field of game theory, which is the $(k-1)$-resilient equilibrium. In particular, a protocol induces a $(k-1)$-resilient equilibrium if it is a Nash equilibrium and if any coalition of less than $k$ players has no incentive to deviate from the protocol. Other rational secret sharing schemes are presented in [15–18].

The schemes of [9,12,19] consider settings where the dealer has a direct connection to each of the players to send each players' share. In addition, players have access to a simultaneous broadcast channel, whereby any transmission sent over the channel is automatically received by all the players (although [12] presented a sketch in the end of his paper over an asynchronous broadcast channel). These assumptions are relaxed in [20], whereby players still have access to a broadcast channel, but transmissions are performed asynchronously. In addition, ref. [20] showed that the schemes of [9,12] are not exactly Nash equilibria if players are allowed to perform a superpolynomial number of computations—which is not at all a given requirement in games according to game-theory literature (i.e., some games are even assumed to be infinite [21]). Ref. [20] thus presented a scheme that is a Nash equilibrium in an information-theoretic sense by drawing shares from an unbounded domain. The scheme of [20], however, assumes that players are allowed to receive shares of arbitrary size. The results of [20] have theoretical appeal, but as per [8], coming up with rational secret schemes where participants are constrained to compute in polynomial time, i.e., cryptographic rational secret schemes, are still meaningful. This led [8] to formulate notions of computational Nash equilibria, computationally strict Nash equilibria, as well as $(k-1)$-resilient computational Nash equilibria, which are modified notions of Nash equilibria over games that constrain its participants to operate in polynomial-time. Moreover, the equilibrium notions of [8] are defined in terms of actions cast as information transmissions relative to each participants' point-of-view—disregarding any hidden internal computations done by other participants. The scheme of [8] is asynchronous and operates over point-to-point networks instead of broadcast channels. In particular, [8] uses cryptographic primitives termed verifiable random functions (VRFs) [22,23].

The setting considered in [8], however, assumes that the dealer has access to each of the players, and each player has access to all other players over a point-to-point network. In this paper, we consider rational secret sharing schemes over general networks, which is a further relaxation from the networks considered in [8,20]. In particular, in a general network, the dealer is not guaranteed to have direct access to each of the players, and players are not guaranteed to have direct access to each of the other players. This implies that transmissions from the dealer or from a player may have to pass through some other player nodes in the network before it reaches its intended recipient. The work of [24,25] deals with the problem of securely disseminating a player's individual share of the secret given that the dealer is not directly connected with each player. In particular, Ref. [24] specifies a graphical property of the network, namely, the $k$-path disjoint property, as a condition for securely disseminating a player's share despite general network constraints. The work

of [26] presents a non-rational secret sharing scheme that is secure on general networks and has much less communication complexity—under the condition that the corresponding graph describing the network topology is *k*-propagating [26]. Both the schemes of [24,26], however, deal more with the first phase of a secret sharing scheme, namely, the secret generation and share/key dissemination phase.

In Section 4.1, we discuss the limitations of the secret sharing schemes surveyed in the above paragraphs. As discussed, the rational secret sharing schemes [8,9,12] assume a broadcast channel or a point-to-point network, by which participants can send messages to one another (whether simultaneous or asynchronously). However, in Section 4.1, we show that in some instances of a general network, equilibrium guarantees of these schemes would fail to hold. On the other hand, non-rational secret sharing schemes (as in [24,26]) are not valid in the case of rational participants, as given rationality and natural assumptions on utility, players are better off by not sharing their shares—as discussed in [9] and described in Section 2.3. It is the goal of the paper, then, to present protocols which provide equilibrium guarantees (under certain conditions of the network topology), even in the combined case of a general network topology over rational participants for all phases of a secret sharing protocol. In particular, our contributions are as follows:

*Our Contributions*

1.  In this paper, we provide protocols that guarantee equilibrium even in the combined case of a general network topology over rational participants for all phases of a secret sharing protocol. We likewise state the required graphical properties of such general networks in order for such equilibria to hold. Thus, our protocols are able to overcome the limitations of existing protocols that are either non-rational or which assume broadcast channels/point-to-point connections among participants—albeit under some conditions on the network topology. In particular, we present three protocols. The first protocol uses a pseudorandom function cryptographic primitive [2] and induces a computational Nash equilibrium given an online dealer, i.e., the dealer transmits information throughout the protocol. For the second protocol, we use the verifiable random functions as conducted in [8], which also results in a computational Nash equilibrium but requires only a semi-online dealer, i.e., the dealer transmits information only at certain phases of the protocol, but is not needed throughout the protocol's execution. The second protocol, however, has much higher round complexity compared to the first scheme. The equilibria of each scheme borrows a technique proposed by [8], which is to randomly draw the value of a definitive iteration from a geometric distribution but to delay the moment when players discover the definitive iteration to create uncertainty. In addition, we apply a scheme inspired by [24] to distribute a secret perfectly in a general network. However, in Section 4.1, we mention that additional mechanisms are required in order for computational Nash equilibrium to provably hold—and we show reasons why the equilibrium is not clear under a straightforward combination of the schemes of [8,24]. Moreover, we mention the required graph-theoretic properties of the general network required for such equilibria , which we term as the *k*-disjoint property, where each pair of nodes in the graph has at least *k* disjoint paths connecting them.
2.  Aside from computational Nash equilbrium, we also show that our proposed protocol induces stronger notions of Nash equilibrium, i.e., computationally strict Nash equilibrium and $(k-1)$-resilient computational Nash equilibrium following [8]. For each equilibrium notion, we present the required properties of the network topology needed for the equilibrium to hold. These properties are expressed using graph theoretical concepts.
3.  We present new notions of the computational Nash equilibrium. The first is termed a $\Phi$-resilient computational Nash equilibrium, whereby a protocol is a $\Phi$-resilient if it is a computational Nash equilibrium and if it is resilient to any coalition that satisfies the properties listed in $\Phi$, regardless of the coalition's size, where the properties in $\Phi$

are expressed using graph theoretical concepts. We present a third protocol which is a $\Phi$-resilient computational Nash equilibrium and derive the result that a $k$-resilient protocol may be resilient to some coalitions of size greater than $k$, as long as such coalitions satisfy the graphical properties required in $\Phi$. The second equilibrium notion is termed $(n-1)$-key leakage resilient equilibrium, whereby a rational secret sharing scheme is still a computational Nash equilibrium in spite of some players acquiring $(n-1)$ secret keys.

## 2. Model and Definitions

Let $\kappa \in \mathbb{N}$ denote a security parameter, where the notion of a security parameter relative to a cryptographic scheme is explained in detail in [2]. A function $f : \mathbb{N} \to \mathbb{R}$ is *negligible* if, for all $c > 0$, there is a $\kappa_c > 0$ such that $f(\kappa) < 1/\kappa^c$ for all $\kappa > \kappa_c$. Throughout the paper, the notation $x \leftarrow X$ refers to $x$ being randomly drawn from the probability distribution of random variable $X$, but it is also sometimes used as $y \leftarrow f(x)$, where $f$ is some probabilistic function.

Let $\mathcal{A}$ be any probabilistic polynomial-time algorithm. The *advantage* of $\mathcal{A}$ is defined to be its capacity to distinguish between the probability distributions of two collections of random variables. For instance, let $\mathcal{X} = \{X_\kappa\}_{\kappa \in \mathbb{N}}$ and $\mathcal{Y} = \{Y_\kappa\}_{\kappa \in \mathbb{N}}$ be two collections of random variables indexed by $\kappa$. The advantage of algorithm $\mathcal{A}$ in this instance is $|\Pr[\mathcal{A}(1^\kappa, x) = 1] - \Pr[\mathcal{A}(1^\kappa, y) = 1]|$ for $x \leftarrow X_\kappa$ and $y \leftarrow Y_\kappa$. Two collections of random variables $\mathcal{X}$ and $\mathcal{Y}$ are *computationally indistinguishable* if the advantage of any polynomial-time algorithm is negligible in $\kappa$.

An $(n,k)$ *secret sharing scheme* $\Pi$ for domain $\mathcal{S}$ is a polynomial-time protocol carried out by a *dealer* $d$ and a set of $n$ *players* $\{p_1, p_2, \ldots, p_n\}$, where the time spent by the protocol and the size of $\mathcal{S}$ are functions of $\kappa$. In particular, $|\mathcal{S}|$ has to be superpolynomial with respect to $\kappa$ in order for a secret sharing scheme to be secure in the cryptographic sense. The protocol $\Pi$ is given by polynomial-time algorithms $(S_G, S_R)$, where $S_G$ is a *share generation* algorithm, and $S_R$ is a *secret reconstruction* algorithm. To securely disseminate a secret $s$ among the $n$ players, the protocol proceeds in two phases. The first phase is the *secret generation and share (or key) dissemination phase*, where the dealer uses $S_G$ on input $s \in \mathcal{S}$ to generate $n$ shares $\{s_1, s_2, \ldots, s_n\} \in \mathcal{S}$. The dealer gives $s_i$ to player $p_i$ for $i \in [n]$. In the second phase, termed the *secret reconstruction phase*, a subset of players of size $n_a \leq n$, termed the *active players* are meant to collaborate in reconstructing $s$, such that given any set consisting of at least $k$ shares, the secret $s$ can be efficiently and correctly reconstructed using algorithm $S_R$. This is termed the *correctness* property of secret sharing schemes. Moreover, secret sharing schemes satisfy the *secrecy* requirement, whereby any data that provide information on less than $k$ shares reveal nothing about $s$.

### 2.1. Game Theory Definitions

Following standard notions in game-theory [11], a game is described by: (1) a set of participants who have associated utility functions (which are termed as *players*), and possibly other participants without utility functions (for instance, nature as described in [10]); (2) the possible actions available to each participant; (3) rules that determine the order in which participants make their moves; (4) a rule that determines the outcome of every possible game ending; and (5) a definition of the utility function associated with each player in the game. Several forms of games have been considered, but here we consider the extensive form of a game with imperfect information following [20]. Namely, an extensive form game $\mathcal{G}$ with imperfect information is a tuple

$$(N, (A_i)_{i \in [|N|]}, \Omega_H, f_{\texttt{next}}, (\mathcal{I}_i)_{i \in [|N|]}, \boldsymbol{o}, (\mu_i)_{i \in [|N|]})$$

where:

1.     $N$—a finite set of players denoted as $\{p_1, p_2, \ldots, p_n\}$ with $n = |N|$.

2. $A_i$—the action space available to player $p_i$ with an element denoted as $\mathrm{act} \in A_i$. $A_i$ can be finite or infinite.

3. $\Omega_H$—a set of sequences (termed *histories*) with elements $\omega := (\mathrm{act}_1, \mathrm{act}_2, \ldots, \mathrm{act}_m)$ (for some $m > 0$) of actions taken by players that satisfy the following: (1) $\emptyset \in \Omega_H$ and (2) for any $m > 0$, if $(\mathrm{act}_1, \mathrm{act}_2, \ldots, \mathrm{act}_m) \in \Omega_H$ and $m' < m$, then $(\mathrm{act}_1, \mathrm{act}_2, \ldots, \mathrm{act}_{m'}) \in \Omega_H$. A history $(\mathrm{act}_1, \mathrm{act}_2, \ldots, \mathrm{act}_m)$ is terminal if there is no $\mathrm{act}_{m+1}$ such that

$$(\mathrm{act}_1, \mathrm{act}_2, \ldots, \mathrm{act}_{m+1}) \in \Omega_H.$$

The set of actions for player $p_i$ after a non-terminal history $\omega := (\mathrm{act}_1, \mathrm{act}_2, \ldots, \mathrm{act}_m)$ is denoted as $A_i(\omega) := \{\mathrm{act}_{m+1} | (\mathrm{act}_1, \mathrm{act}_2, \ldots, \mathrm{act}_m, \mathrm{act}_{m+1}) \in \Omega_H\}$.

4. $f_{\mathtt{next}}$—a function $f_{\mathtt{next}} : \Omega_H \to N$ for which $f_{\mathtt{next}}(\omega)$ is the player who takes action after history $\omega \in \Omega_H$.

5. $\mathcal{I}_i$—the information partition for player $p_i$, which is a partition of $\{\omega \in \Omega_H | f_{\mathtt{next}}(\omega) = p_i\}$ with the property that $A_i(\omega) = A_i(\omega')$ if $\omega$ and $\omega'$ are both in the same element of the partition. An element of $\mathcal{I}_i$ is denoted as $I$, which is termed an *information set*. The set of actions for $p_i$ after reaching $I$ is $A_i(I)$.

6. $o$—a set of outcomes, where an outcome is a description of events in the game once a terminal history is reached.

7. $\mu_i$—a utility function from the set of terminal histories to $\mathrm{R}$, which determines $p_i$'s gain depending on the game's outcome.

**Definition 1.** *Given an extensive form of game $\mathcal{G}$ with imperfect information, a behavioural strategy (or simply strategy) is denoted as a vector $\sigma := \{\sigma_1, \sigma_2, \ldots, \sigma_n\}$, where for $i \in [n]$, $\sigma_i$ is the strategy of player $p_i$. Each $\sigma_i$ for $i \in [n]$ is a function mapping $I$ to a probability distribution over $A_i(I)$.*

The definition of strategy given in Definition 1 is the standard definition in game-theory [11], whereby actions are functions of histories or information sets. An equivalent (and perhaps more intuitive) definition of strategy for player $p_i \in N$ views actions $A_i(I)$ taken by $p_i$ under information set $I$ as conditional on the *information contained in I*. For instance, a history in an information set $I$ may consist of past actions of a player's internal computations, along with past actions of other players consisting of transmissions sent over a network. In this case, the set of information contained in $I$ consists of the outputs of these internal computations plus the content of transmissions from other players. Strategy in this case is defined as actions taken by a player conditional on the information contained in $I$ after reaching information set $I$. This notion of information contained in an information set is denoted as $\phi_i(I)$ for $p_i \in N$ and is defined below.

**Definition 2.** *Let $p_i \in N$ reach information set $I$. The information from $I$ or information in $I$ is denoted as $\phi_i(I)$, which consists of all possible information from the point of view of $p_i$ upon reaching $I$. The set of actions for $p_i$ after reaching $I$ and conditional on $\phi_i(I)$ is denoted as $A_i(\phi_i(I))$ and $A_i(\phi_i(I)) = A_i(I)$, i.e., the difference between $A_i(\phi_i(I))$ and $A_i(I)$ is merely conceptual.*

**Definition 3.** *Given an extensive form game $\mathcal{G}$ with imperfect information, a behavioural strategy (or simply strategy) is denoted as a vector $\sigma := \{\sigma_1, \sigma_2, \ldots, \sigma_n\}$, where for $i \in [n]$, $\sigma_i$ is the strategy of player $p_i$. Each $\sigma_i$ for $i \in [n]$ is a function mapping the space of $\phi_i(I)$ to a probability distribution over $A_i(I)$.*

**Definition 4.** *Define: $\sigma_{-i} := (\sigma_1, \ldots, \sigma_{i-1}, \sigma i + 1, \ldots, \sigma_n)$, and similarly, define $(\sigma'_i, \sigma_{-i}) = (\sigma_1, \ldots, \sigma_{i-1}, \sigma'_i, \sigma_{i+1}, \ldots, \sigma_n)$, i.e., the strategies of all players are the same as in $\sigma$, except for player i, who changed his strategy to $\sigma'_i$.*

### 2.2. Graph Theory Definitions

Recall that a *graph* $G = (V, E)$ consists of a set of *nodes V* and a set of *edges* $E \subseteq V \times V$, such that two nodes $a_1, a_2 \in V$ are *joined* or are *adjacent* to each other if $(a_1, a_2) \in E$. In this setting, graphs are assumed to be undirected. A *walk* from node *a* to node *b* is a finite sequence of edges $((a_1, b_1), (a_2, b_2), \ldots, (a_m, b_m))$ for some $m > 0$ (i.e., all walks in this setting are assumed to end and we do not consider infinite walks), such that $a_1 = a$, $b_m = b$, and $b_l = a_{l+1}$ for $l \in [m-1]$. The *first edge* of a walk $((a_1, b_1), (a_2, b_2), \ldots, (a_m, b_m))$ is the edge $(a_1, b_1) \in E$. Given a walk $((a_1, b_1), (a_2, b_2), \ldots, (a_m, b_m))$, the nodes $\{a_1, a_2, \ldots, a_m, b_m\}$ comprise the *node sequence* of the walk. A *path* from *a* to *b* is a walk in which all elements of its node sequence are distinct, and the first and last nodes in the node sequence are *a* and *b*, respectively. Given a path from *a* to *b*, the path is said to *originate* at *a*, and the node *a* is termed the *origin-node*, or the *origin*, while the node *b* is termed the *end-receiver node* or the *end-node*. Two distinct nodes $a, b \in V$ are *connected* if there exists a path from *a* to *b*, in which case the path is *connecting a* to *b*. Two paths are *completely disjointed* if their respective node sequences have empty intersection (i.e., they do not cross each other). Aside from these standard graph theory definitions, we also define special types of paths and graphs that will be used in this setting. Let $a, b \in V$ be a pair of distinct nodes.

**Definition 5.** *A set of paths from a to b is internally disjoint if: (1) the node sequences of the paths have a as the origin and b as the end-receiver and (2) if, aside from the beginning and end, the node sequences of the paths do not share any node in common. Furthermore, given a graph $G(V, E)$, let a, b be two distinct pair of nodes in V. A set of k paths from a to b is a set of k-disjoint paths from a to b if they are internally disjoint. Lastly, given a graph $G(V, E)$, let $\bar{V} \subset V$. The set of nodes $\bar{V}$ is k-disconnected if, for each distinct pair of nodes $a, b \in \bar{V}$, we have: (1) $(a, b) \notin E$ and (2) for any path connecting a and b, the size of the node sequence is at least $k + 2$.*

While dense clique graphs are likely to be path-disjoint, it is not necessary for a graph to be a clique in order to be path-disjoint. As shown in Figure 1, we have a graph that is 3-path disjoint even though it is not a clique. A useful property of *k*-path disjoint graphs is stated in Lemma 1, which will be used in the proofs in the Appendix.

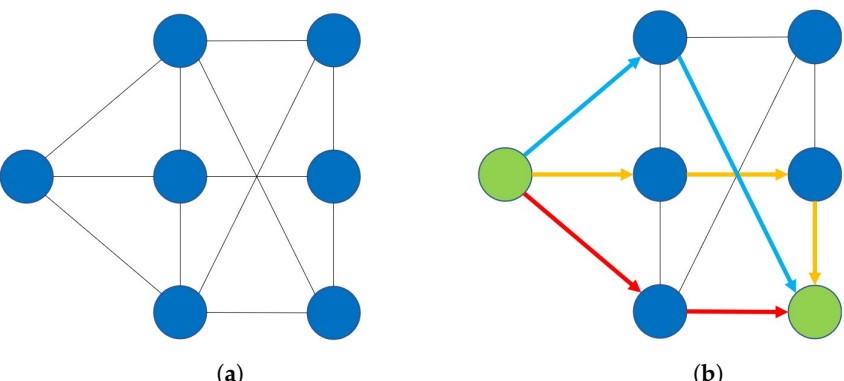

| (**a**) | (**b**) |

**Figure 1.** The left figure (**a**) shows a graph that is $(k = 3)$-path-disjoint even if it is not a clique. An example of a 3-disjoint paths from one green node to another green node given the graph in (**a**) is shown in the right figure (**b**).

**Lemma 1.** *Given a k-path-disjoint graph $G(V, E)$, let $\bar{V} \subset V$ be a set of size $k - 1$. For each distinct pair of nodes $a, b \in V$, any set of k-disjoint paths from a to b contains a path that does not contain nodes belonging to $\bar{V}$.*

**Proof.** Let $a, b$ be an arbitrary pair of distinct nodes in $V$. Let $\bar{V} \subset V$ be an arbitrary subset of nodes of $V$ of size $k - 1$. Suppose that there exists a set of *k*-disjoint paths from *a* to *b* such that each path contains nodes belonging to $\bar{V}$. Since this particular set of paths is internally disjoint, this implies that there are *k* paths whose first edges are distinct from each other

and which originate at $a$. Distribute the members of $\bar{V}$ to these $k$ paths. However, since $|\bar{V}| < k$, some paths do not contain nodes belonging to $\bar{V}$, which is a contradiction.   □

### 2.3. Rational Secret Sharing

Early secret sharing schemes' model players are either completely honest or malicious [1]. In a *rational secret sharing scheme*, however, players are *rational* in the game-theoretic sense and are associated with utilities depending on outcomes of a game [9]. Thus, a protocol $\Pi$ in rational secret sharing corresponds to a prescribed strategy over a game. In particular, in a rational secret sharing game, there are $n + 1$ participants consisting of $n$ players who wish to reconstruct the secret and have associated utility functions, plus a dealer without an associated utility function. However, among these $n$ players, only a subset of $n_a \le n$ players are willing to participate in the protocol, namely, the *active players*. In the setting of [9], each active player has access to a *broadcast channel*, whereby if an active player transmits information in this channel, all other active players in the game learn the transmitted information automatically. An important result of [9] (and described in Section 4.1), is that standard non-rational cryptographic protocols fail if participants are modeled as rational instead of plainly honest or malicious.

The secret sharing game described in [9] proceeds in several *iterations*, and each iteration consists of multiple *communication rounds*. At the beginning of each iteration, the dealer privately distributes information to each of the $n$ players. Afterwards, the subset of $n_a$ active players run the protocol among themselves by *simultaneously broadcasting* messages in a series of rounds. At the end of an iteration, the protocol either terminates or proceeds to the next iteration. At the beginning and throughout the game, it is assumed that the dealer and each of the players know the identities of the $n_a$ active players.

The strategies of the game's active players in [9] can be viewed as probabilistic interactive Turing machines [27] that operate in polynomial-time following [8]. In this context, the dealer and the active players can perform arbitrary polynomial-time probabilistic computations internally in each round. In addition, in each round, the dealer and the active players can either (1) broadcast information (i.e., a share) or (2) abstain from broadcasting information (players only). In addition, players can (3) abort the game or (4) output a guess of the secret. If all active players abort, the game ends, and the outcome of a game is described in terms of the outputs of each active player. Following [9], the value of the utility function $\mu$ of a player increases if it correctly outputs the secret $s$. Each active player, however, prefers that the number of active players who correctly outputted $s$ be as small as possible, as shown in Definition 6 below. For simplicity, however, in all that follows in this paper, we assume that all players are active, i.e., $n_a = n$, so that if some player is referred to as performing some action or strategy or whose utility is being computed, it is automatically assumed that the player is an active player.

**Definition 6.** *Let $\boldsymbol{o}$ denote an outcome vector of length $n$ such that $o_i = 1$ if player $p_i$ outputs the secret $s$. If a player outputs $s$ correctly, it is considered to have learned $s$, without the need to look into its internal computations. If $p_i$ outputs a wrong secret or aborts without any output, $p_i$ is considered to not have learned the secret and $o_i = 0$. Let $\mu_i(\boldsymbol{o})$ denote the utility of player $i$ given outcome $\boldsymbol{o}$. Following [8], let $\boldsymbol{o} = \{o_1, o_2, \dots, o_n\}$ and $\boldsymbol{o}' = \{o_1', o_2', .., o_n'\}$ be two distinct outcomes. For each player $p_i \in \boldsymbol{P}$, we have: (1) if $o_i > o_i'$ then $\mu_i(\boldsymbol{o}) > \mu_i(\boldsymbol{o}')$, and (2) if $o_i = o_i'$ and $\sum_{i \in [n]} o_i < \sum_{i \in [n]} o_{i'}'$, then $\mu_i(\boldsymbol{o}) > \mu_i(\boldsymbol{o}')$.*

**Definition 7.** *Given an outcome $\boldsymbol{o}$, let $u_i(\boldsymbol{o})$ denote player $i$'s expected utility function, where expectation is taken over the value of $s$ (which is assumed to be chosen uniformly by the dealer at the beginning of the game), the randomness of the dealer, and the randomness of each player's strategy.*

**Definition 8.** *Let $s \in \mathcal{S}$ be a secret. Following [8,12], define $U_i^+ := \mu_i(\boldsymbol{o})$ if $o_i = 1$, and $o_{i'} = 0$ for all $i' \in [n] \setminus i$, i.e., player $p_i$ learns the secret but no other player does. On the other hand, for any $\boldsymbol{o}$ such that $o_i = 1$, and $\sum_{i' \in [n] \setminus i} o_{i'} > 0$, i.e., player $p_i$ learns the secret and at least some other player*

*does as well, we define the resulting utility as a single value* $U_i := \mu_i(\boldsymbol{o})$. *Lastly, for any* $\boldsymbol{o}$ *such that* $o_i = 0$, *i.e., player* $p_i$ *does not learn the secret, we define the resulting utility as a single value* $U_i^- := \mu_i(\boldsymbol{o})$. *For each player* $p_i \in N$, *define* $U_{\texttt{random}}$ *as* $U_{\texttt{random}} := (1/|\mathcal{S}|)U_i^+ + (1 - 1/|\mathcal{S}|)U_i^-$, *which is the expected utility of a player who outputs a random guess of s if other parties abort or output a wrong guess.*

*For this setting, the functions* $U_i^+$, $U_i^-$ *and* $U_i$ *are the same for all players so that we can refer to them simply as* $U^+$, $U^-$ *and* $U$. *For this paper, we assume that* $U^+ > U > U^-$. *Moreover, it is required that* $U > U_{\texttt{random}}$ *since, otherwise, players will have no incentive to participate in the game as shown in [8].*

**Definition 9.** *A protocol* $\Pi$ *in a rational secret sharing game has an online dealer if the dealer continually sends transmissions at each iteration until the secret is reconstructed, i.e., the dealer's continual transmissions at each iteration throughout the game is required for players to reconstruct the secret. A protocol has a semi-online dealer if the dealer sends transmissions for a finite number of iterations, after which, the dealer stops sending any additional transmission even if the secret is still not yet constructed by the players, i.e., the players are left to reconstruct the secret on their own (without the dealer) at some point in the game.*

*2.4. $A_{GN}$ Rational Secret Sharing*

The rational secret sharing schemes above consider games where players have access to broadcast channels, and where the dealer can directly transmit individual shares to each player. In this setting, we relax the assumption that the dealer can directly transmit individual shares to each player. Rather, the dealer has direct access to a certain number of players in the network (which may not necessarily include each player). In addition, players may be unable broadcast information to all other players at once. Rather, a player can only transmit information directly to a certain number of players (which may not necessarily include each player). This leads to the notion of *asynchronous general network* ($A_{GN}$) rational secret sharing, which is a generalization of a rational secret sharing game. To express these notions better, we use some concepts from graph theory.

We denote an $A_{GN}$ rational secret sharing game associated with a graph $G(V, E)$ with $n + 1$ participants (i.e., 1 dealer and $n$ players) in Definition 10. The placement of the dealer and each of the players in the general network's topology is represented by $G$, where the dealer and each of the players are assigned a node in $V$ so that $|V| = n + 1$. If an edge in $E$ joins two nodes of $V$, this implies that the player (or dealer) represented by the origin-node can send a transmission using the network to the other player represented by the end-node. In the description of $\mathcal{G}$ below, we switch between referring to the participants as computational models (i.e., Turing machines), and as nodes in the graph $G$. However, it will be understood from the context that if the dealer or a player performs some computations, it is doing so internally in its capacity as a computational model, while if the dealer node or a player node sends a transmission to another player node, the participants are sending transmissions with reference to their representations as nodes in $G$.

**Definition 10.** *An asynchronous general network ($A_{GN}$) rational secret sharing game $\mathcal{G}$ associated with a graph $G(V, E)$ and domain $\mathcal{S}$ is described by the following:*

1.　*The game has $n + 1$ participants consisting of $n$ players $N := \{p_1, p_2, \ldots, p_n\}$, where each player $p_i$ is associated with utility function $\mu_i$ for $i \in [n]$, and a dealer d who does not have an associated utility function. The utility function $\mu_i$ for $p_i \in N$ follows the utility function described in Definition 6.*
2.　*The participants of the game are represented by the nodes $V$ of $G$. An edge $(a, b) \in E$ implies that node a (i.e., a player or the dealer) can directly transmit information to node b (another player). The dealer is required to have at least one edge joining its node with another player's node.*
3.　*The game proceeds in phases. The first set of phases is termed the key and share a generation/dissemination phase, while the next set of phases is termed the secret reconstruction*

*phase. A protocol of the game should take care of letting players know when a phase ends and when the next phase begins. The key and share generation/dissemination phase is viewed as a single iteration of the game, i.e., iteration 0 and consists of several communication rounds. In iteration 0, the dealer samples a secret $s \in S$ and distributes shares of the secret along with other arbitrary forms of information (i.e., secret/public keys) to the players.*

4.   *The secret reconstruction phase consists of a sequence of iterations $1, 2, \ldots$. Each iteration consists of a sequence of communication rounds (or round for short). In each round, the dealer and the players can internally perform arbitrary polynomial-time and size probabilistic computations, and can either (1) transmit information to several other player nodes with whom its node is joined according to E or (2) abstain. In addition, players can (3) output a guess of the secret key or (4) abort. If a player aborts, it leaves the game and no longer has access to information from subsequent iterations/rounds in the game.*

5.   *In each round in the key and share generation/dissemination phase, and in each round in an iteration in the secret reconstruction phase, the player and the dealer can transmit information to several other player nodes (with whom its node is joined in E) simultaneously. After transmitting information, a player can no longer transmit again within the round, i.e., transmission is performed simultaneously and once within a round. After transmission of information, a player receives information simultaneously from other players with whom it is joined in E. With this rule, it follows that information received by a player in one round can only be used in computations/transmissions in the next round.*

6.   *The value of iteration and each round within an iteration is common knowledge among all participants throughout the game. Likewise, a protocol of the game should take care of letting all participants know when the current iteration ends and when the next iteration begins.*

7.   *The game ends once all players abort. Once a game ends, its outcome is defined as a vector $\boldsymbol{o} = \{o_1, o_2, \ldots, o_n\}$ such that $o_i = 1$ if player $p_i$ `outputs` the secret $s$.*

8.   *The expected utility $u_i$ of player $p_i$ given outcome $\boldsymbol{o}$ for $i \in [n]$ follows the expected utility function described in Definition 6.*

From above definition, the graph in a rational secret sharing game with broadcast and dealer access to each player [9] can be seen as a special instance of an $A_{GN}$ rational secret sharing game, where the associated graph is fully connected, i.e., each player node has edges to all other player nodes, and the dealer has edges to each of the players. From the description of an $A_{GN}$ game above, it could be seen that the action space is very large since it includes all possible internal computations at each round as well as all possible transmissions among players. With a very large action space, listing down a function that maps information sets $I$ to a probability distribution over a player's actions is not feasible. This where the notion of $\phi_i(I)$ becomes useful, whereby actions are dependent on the information contained in an information set $I$, where actions of a player are decided for each round. As a result, to define a strategy, we only need to define actions dependent on certain relevant information that directly affects its utility rather than specifying each possible information set. With this, let the participants of an $A_{GN}$ rational secret sharing game $\mathcal{G}$ be indexed by the set $0 \cup [n]$ such that the dealer has index 0 and player $p_1$ has index 1, player $p_2$ has index 2, etc. We define strategies and secret sharing schemes in the context of an $A_{GN}$ rational secret sharing game as follows.

**Definition 11.** *Let $\mathcal{G}$ be an $A_{GN}$ rational sharing game associated with a graph $G(V, E)$ and domain $\mathcal{S}$. A polynomial-time strategy $\boldsymbol{\sigma} = \{\sigma_0, \sigma_1, \ldots, \sigma_n\}$ is a set of polynomial-time strategies for each participant that—conditional on information $\phi_i(I)$ in information set $I$—defines at each round the participant's (1) internal probabilistic computations, (2) transmissions (or lack of transmissions) among participants with whom it is joined by an edge in E, and (3) output and abort actions.*

**Definition 12.** *Let $\mathcal{G}$ be an $A_{GN}$ rational sharing game associated with a graph $G(V, E)$ and domain $\mathcal{S}$. Given a polynomial-time protocol $\Pi$ over $\mathcal{G}$, the strategy $\boldsymbol{\sigma} = \{\sigma_0, \sigma_1, \ldots, \sigma_n\}$ corresponding to $\Pi$ is a set of polynomial-time strategies for each participant that define its actions at*

*each round, such that the participant's actions follow* $\Pi$. *In this case,* $\sigma$ *is termed as the strategy prescribed by* $\Pi$.

**Definition 13.** *Let* $\mathcal{G}$ *be an* $A_{GN}$ *rational sharing game associated with a graph* $G(V, E)$ *and domain* $\mathcal{S}$, *and let* $s \in \mathcal{S}$ *denote the secret chosen by the dealer at iteration* 0. *A protocol* $\Pi$ *over* $\mathcal{G}$ *is an* $(n, k)$ $A_{GN}$ *secret sharing scheme (not yet considering rationality) if it corresponds to a polynomial-time strategy* $\sigma$, *such that if players follow the actions prescribed by* $\sigma$ *and obtain information that reveal at least k shares, they can reconstruct the secret s efficiently (correctness). If players obtain information that reveal less than k shares, the probability of correctly outputting s is* $1/|\mathcal{S}|$ *(secrecy).*

## 3. Equilibrium Notions

The standard notion of equilibria in a game-theoretic setting is the *Nash equilibrium*, and a protocol is said to induce a Nash equilibrium if no player can gain any advantage by deviating from the protocol—assuming that all other players follow the protocol. However, as observed in [8,9], the standard Nash equilibrium concept is inadequate (too weak) in the setting of rational secret sharing. This led [9] to consider more specialized versions of the Nash equilibrium, such as *equilibrium surviving iterated deletion of weakly dominated strategies [11]*. However, even this notion of equilibrium is not without problems [8,20], leading [20] to consider further refinements in the equilibrium such as the *strict Nash equilibrium*. In this paper, we adopt notions of *computational* equilibrium from [8], which have the merit of closely retaining the properties of a *strict Nash equilibrium* while considering computational constraints. For this, let $\mathcal{G}$ be an $A_{GN}$ rational sharing game associated with a graph $G(V, E)$ and domain $\mathcal{S}$. Let protocol $\Pi$ denote a $(n, k)$ $A_{GN}$ secret sharing scheme over $\mathcal{G}$. Let $\sigma = \{\sigma_0, \sigma_1, \ldots, \sigma_n\}$ denote the strategy corresponding to $\Pi$. Let $f$ denote a negligible function over $\kappa$. We have the following:

**Definition 14.** $\Pi$ *induces a computational Nash equilibrium over* $\mathcal{G}$ *if, for each player* $p_i$ *for* $i \in [n]$ *in* $\mathcal{G}$, *we have* $u_i(\sigma'_i, \boldsymbol{\sigma_{-i}}) \leq u_i(\boldsymbol{\sigma}) + f(\kappa)$ *for any other polynomial time strategy* $\sigma'_i$ *for player* $p_i$.

**Definition 15.** *From [8], we define* $\mathtt{view}_{-i}^{\Pi}$ *as follows. Let* $\mathtt{script}_d$ *denote the transmissions of the dealer to its adjacent nodes across all rounds of the game. Let* $\mathtt{script}_i$ *denote the transmissions of* $p_i$ *to its adjacent nodes (across all rounds of the game), but which do not include transmissions after* $p_i$ *outputs a guess of the secret s. Let* $\mathtt{script}_{-i}$ *denote the set of transmissions of* $p_{i'}$ *for* $i' \in [n]$ *with* $i' \neq i$ *to its adjacent nodes (across all rounds of the game). Let all participants follow the strategies prescribed by* $\Pi$. $\mathtt{view}_{-i}^{\Pi}$ *is defined as information which includes* $\mathtt{script}_d$, $\mathtt{script}_i$, *and* $\mathtt{script}_{-i}$, *plus all randomness involved in the computations of* $p_{i'}$ *for* $i' \in [n]$ *with* $i' \neq i$ *across all rounds.*

**Definition 16.** *Let* $\rho_i$ *be another strategy of* $p_i$ *with* $\rho_i \neq \sigma_i$. *Let all participants (except* $p_i$) *follow the strategies prescribed by* $\Pi$. *For its part, player* $p_i$ *follows strategy* $\rho_i$. *Given this set of strategies, let* $\mathtt{script}_d$, $\mathtt{script}_i$, *and* $\mathtt{script}_{-i}$ *be defined as in Definition 15. Let T be some polynomial-time algorithm that knows the entire view of* $p_i$ *as it follows* $\rho_i$ *(i.e., player* $p_i$'s *randomness, its computations, its transmissions as written in* $\mathtt{script}_i$, *and any transmissions received from other participants) and which outputs a truncation* $\mathtt{script}'_i$ *of* $\mathtt{script}_i$. *We define* $\mathtt{view}_{-i}^{T, \rho_i, \Pi}$ *as information which includes* $\mathtt{script}_d$, $\mathtt{script}'_i$, *and* $\mathtt{script}_{-i}$, *plus all randomness involved in the computations of* $p_{i'}$ *for* $i' \in [n]$ *with* $i' \neq i$ *across all rounds. Similarly, define* $\mathtt{view}_{-i}^{\rho_i, \Pi}$ *as the same information contained in* $\mathtt{view}_{-i}^{T, \rho_i, \Pi}$ *but which excludes reference to T.*

**Definition 17.** *Let $f$ denote a negligible function in $\kappa$. For $i \in [n]$, a strategy $\rho_i$ is equivalent with respect to $\Pi$ or $\rho_i \sim \Pi$ if there exists a polynomial-time algorithm $T$ such that for all polynomial-time distinguishers $D$, we have:*

$$\left| \Pr\left[ D(1^{\kappa}, \mathtt{view}_{-i}^{T, \rho_i, \Pi}) = 1 \right] - \Pr\left[ D(1^{\kappa}, \mathtt{view}_{-i}^{\Pi}) = 1 \right] \right| \leq f(\kappa)$$

**Definition 18.** *Let protocol $\Pi$ denote a $(n,k)$ $A_{GN}$ secret sharing scheme over $\mathcal{G}$. Let $\sigma = \{\sigma_0, \sigma_1, \ldots, \sigma_n\}$ denote the strategy corresponding to $\Pi$. We say that $\Pi$ induces a computational strict Nash equilibrium: (1) if it induces a computational Nash equilibrium and (2) if, for any polynomial-time strategy $\sigma_i'$ for which $\sigma_i' \not\sim \Pi$, there is a $c > 0$ such that $u_i(\sigma) \geq u_i(\sigma_i', \sigma_{-i}) + 1/\kappa^c$ for infinitely many values of $\kappa$.*

Having considered the above notions of equilibrium, we now consider an extension of these equilibrium concepts in the presence of coalitions. Namely, given an $A_{GN}$ secret sharing game $\mathcal{G}$ with $n + 1$ participants, a coalition $\mathcal{C} \subseteq \boldsymbol{P}$ is a set of players whose strategies are coordinated arbitrarily. The output of $\mathcal{C}$ is a single value which represents the individual outputs of each member of $\mathcal{C}$. The utility function of $\mathcal{C}$ is denoted as $\mu_{\mathcal{C}}$, and the expected utility function is $u_{\mathcal{C}}$. Similarly, denote by $\sigma = (\sigma_{\mathcal{C}}, \sigma_{-\mathcal{C}})$ the resulting strategy if members of $\mathcal{C}$ follow $\sigma_{\mathcal{C}}$ while other players that are not members of $\mathcal{C}$ follow $\sigma_{-\mathcal{C}}$. Let protocol $\Pi$ denote a $(n,k)$ $A_{GN}$ secret sharing scheme over $\mathcal{G}$. Let $\sigma = (\sigma_{\mathcal{C}}, \sigma_{-\mathcal{C}})$ be a strategy that corresponds to $\Pi$. Let $f$ denote a negligible function over $\kappa$.

**Definition 19.** *$\Pi$ induces a $(k - 1)$-resilient computational Nash equilibrium if, for any $\mathcal{C} \subseteq \boldsymbol{P}$ with $|\mathcal{C}| < k$, for any polynomial-time strategy $\sigma_{\mathcal{C}}'$ such that $\sigma_{\mathcal{C}}' \neq \sigma_{\mathcal{C}}$, we have $u_{\mathcal{C}}(\sigma_{\mathcal{C}}', \sigma_{\mathcal{C}}) \leq u_{\mathcal{C}}(\sigma) + f(\kappa)$.*

For completeness, coalition versions of the above definitions are stated in Appendix A.

*Additional Equilibrium Notions*

We now present two novel equilibrium notions, for which some of our proposed protocols satisfy. The first equilibrium notion (Definition 20) is a $(n - 1)$-*key leakage-tolerant computational Nash equilibrium*, which is a computational Nash equilibrium that is resistant to secret key leakage—given a scheme which uses cryptographic primitives involving secret keys. The second equilibrium is the notion of a $\Phi$-equilibrium (Definition 21). This notion states that a $(k - 1)$-computational Nash equilibrium can hold even in the presence of large coalitions whose size is larger than $k$—as long as these coalitions satisfy the graphical properties listed in $\Phi$. This is in contrast to standard definitions of $(k - 1)$-resilient computational Nash equilibria whereby an upper bound on the size of any coalition is imposed.

**Definition 20.** *Let $\mathcal{G}$ be an $A_{GN}$ rational secret sharing game with $n + 1$ participants associated with a graph $G(V, E)$ and domain $\mathcal{S}$. Let $\Pi$ be a cryptographic protocol that uses cryptographic primitives involving a set of secret keys $\boldsymbol{sk} := \{\boldsymbol{sk}_i\}_{i \in [n]}$, where $\boldsymbol{sk}_i$ is a tuple of secret keys of player $p_i$. $\Pi$ induces an $(n - 1)$-key leakage-tolerant computational Nash equilibrium over $\mathcal{G}$ if it is a computational Nash equilibrium, even if each player acquires up to $n - 1$ tuples of secret keys.*

We note that as per Definition 20, each player is constrained to obtain up to $n - 1$ secret keys, where the secret keys may be obtained through arbitrary means, i.e., by sharing of keys within a coalition or through side-channel attacks. This rules out the case whereby a certain player who currently has $n - 1$ secret keys forms a coalition with the remaining player whose secret key it does not yet have in order to obtain $n$ secret keys in total. Such cases are ruled out by the definition of the $n - 1$-key leakage-tolerant computational Nash equilibrium.

**Definition 21.** *Let $\mathcal{G}$ be an $A_{GN}$ rational secret sharing game with $n + 1$ participants associated with a graph $G(V, E)$ and domain $\mathcal{S}$. Let $\Phi$ be a set of conditions over $V$ relative to $E$. $\Pi$ induces a $\Phi$-resilient computational Nash equilibrium over $\mathcal{G}$ if, for any arbitrary coalition $\mathcal{C} \subseteq N$ whose respective nodes in $G$ satisfy the conditions in $\Phi$, for any polynomial-time strategy $\sigma'_{\mathcal{C}}$ such that $\sigma'_{\mathcal{C}} \neq \sigma_{\mathcal{C}}$, we have $u_{\mathcal{C}}(\sigma'_{\mathcal{C}}, \sigma_{\mathcal{C}}) \leq u_{\mathcal{C}}(\sigma) + f(\kappa)$.*

## 4. Protocols

### 4.1. Overview of Existing Protocols

Existing protocols in the literature are listed in Table 1. These protocols can be grouped into two major categories: those that allow for rational participants and those that do not (i.e., non-rational protocols). From Table 1, we discuss the limitations of these schemes as follows.

**Table 1.** Rational refers to whether the scheme considers participants as rational or not. Bounded refers to whether the shares used in the scheme are finite or infinite. Async refers to whether the scheme allows for asynchronous communication among participants. B/p2p refers to whether the scheme assumes that players are connected by either a broadcast or a point-to-point network. General refers to whether the scheme allows for participants to be connected under a general network topology. The schemes of [24,26] are marked with yes* under the "general" column since they work on a general network where the dealer may not have direct connections to all players during the share dissemination phase. However, it is not clear in [24,26] how players communicate their shares to each other and how the network topology would be during the secret reconstruction phase.

| Scheme | Rational | Bounded | Async | b/p2p | General |
|---|---|---|---|---|---|
| Halpern and Teague [9] | yes | yes | no | yes | no |
| Gordon and Katz [12] | yes | yes | yes | yes | no |
| Fuchsbauer et al. [8] | yes | yes | yes | yes | no |
| Kol and Naor [20] | yes | no | yes | yes | no |
| Shah et al. [26] | no | yes | yes | no | yes* |
| Dolev et al. [24] | no | yes | yes | no | yes* |
| Ours | yes | yes | yes | no | yes |

1. *Rational schemes assume broadcast channels/point-to-point networks*. The existing rational schemes [8,9,12,20] are not designed to operate on a general network since they assume that the dealer $d$ along with $n$ players have access to either a broadcast channel or a point-to-point network (i.e., all participants are pairwise connected), for which these schemes achieve $(k - 1)$ equilibrium given some $k < n$. For reference, the algorithm of [8] is listed in detail in Appendix E. If applied to some instances of a general network, however, the equilibrium guarantees that these schemes would fail. For instance, in Figure 2, $d$ is directly connected to only $l = 3$ players, and yet, $d$ needs to send at least 12 messages to all $n = 12$ players in order to share the secret in a fair manner following the p2p/broadcast protocol (i.e., since all of these schemes make the dealer directly send a message to each player). Given this topology, $d$ is forced to use only $l$ connections to send all of its messages. As a result, one player that is directly connected to $d$ (say player $p_i$) is bound to receive at least $d/l$ messages. If $d/l \geq k - 1$, $p_i$ learns the secret. In this example, it follows that the equilibrium guarantees of these schemes would fail for some values of $k$. The same analogy could be applied to some player communicating information to another player in the secret reconstruction phase, i.e., several players may send information to one player who is in a network bottleneck.

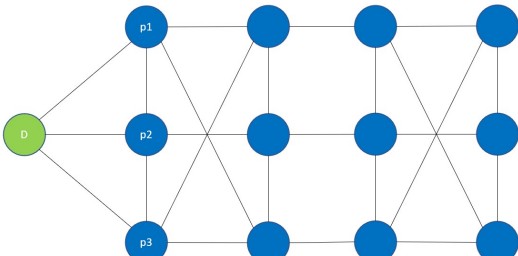

**Figure 2.** An instance of a general network where the equilibrium guarantees of broadcast/p2p-network rational secret sharing schemes would fail. Here, the dealer (green node) is only directly connected to 3 players, $p1, p2, p3$, whereas there are 12 players (blue nodes) in total. Given that in a broadcast/p2p-network rational secret sharing scheme, the dealer has to communicate messages to all players, the dealer in this case is forced to course at least 12 messages through the set of players $p1, p2, p3$ (many of which are not designed to be seen by $p1, p2, p3$). It follows that at least one of $p1, p2, p3$ would eventually obtain at least 4 messages from the dealer that provide information on the secret, breaking the equilibrium guarantees for all $k < 4$.

2. *Non-rational schemes*. On the other hand, the protocols of [24,26] are secure for general networks but assume that participants are non-rational. Specifically, [24] presents the SMT algorithm which addresses the problem of securely disseminating the shares of each player during the secret generation/share dissemination phase. Briefly, for each share outputted by the share generation algorithm, the SMT treats each share as a *new secret*, and breaks it down into another $k$ sub-shares. For each player, SMT sends these $k$ sub-shares along $k$-disjoint paths, for which each player is able to securely reconstruct its individual share (not yet the secret). The protocol of [26] improves upon the SMT concept by lowering communication complexities. Both [24,26], however, deal with the problem of disseminating shares in a general network during the secret generation and share dissemination phase. However, it is not clear in their paper how the secret reconstruction phase would proceed, i.e., whether players are still connected over a general network once they communicate shares to each other. In our proposed protocols, however, we assume that in both the secret generation/key dissemination phase, and the reconstruction phase, all participants are constrained by a general network. However, perhaps a more fundamental problem with non-rational cryptographic protocols is pointed out in [8,9]. In particular, if players are modeled as rational with natural assumptions on their utilities, such non-rational schemes would fail during the secret reconstruction phase. This is due to the widened action space of rational players, along with their utility maximizing behaviour (compared to plain honest players). For instance, suppose that utility is modeled whereby all players want to learn the secret, but prefer that the smallest number of other players learn the secret as possible (following Section 2.3). It can be shown that each player does no worse (and could even do better) by withholding from sharing his secret (this action is now possible since the player is no longer plainly honest, but rational). To see this, suppose that the non-rational scheme corresponds to an $(n, k)$ secure secret sharing scheme and consider a player $p_i$, $i \in [n]$. If less than $k - 1$ players share the secret, $p_i$ would not learn the secret regardless of his actions. If more than $k - 1$ players share the secret, $p_i$ would learn the secret regardless of his actions as well. If exactly $k - 1$ players share the secret, then $p_i$ is better off by not sharing his secret since he can reconstruct the secret given his hidden share along with the $k - 1$ other shares.

From the discussion above, the equilibrium results of existing rational secret sharing schemes need to be qualified in the case of a general network. On the other hand, existing non-rational schemes for general network have to be modified if rational participants are allowed. As such, the goal of the proposed secret sharing protocols below is to operate over a general network in all phases given rational participants. In the process, the specific

network conditions (i.e., topology) that allow for the existence of desirable equilibrium where all players learn the secret are specified.

### 4.2. High-Level Overview of Our Protocols

The protocol of [8] is shown in detail in Appendix E. In summary, Ref. [8]'s protocol relies on two components to achieve *computationally strict Nash equilibrium*, namely: (1) *uncertainty on the definitive stage* and (2) *protocol compliance checking*. Given $n$ players, the first component (1) is achieved by drawing two random polynomials, $G$ and $H$, such that $G(0) = s$ and $H(0) = 0$. In addition, we have $\{g_i^* := G(i) \oplus V_E(sk_i, r^*)\}_{i \in [n]}$ and $\{h_i^* := H(i) \oplus V_E(sk_i, r^* + 1)\}_{i \in [n]}$, where $r^*$ represents the definitive iteration and $V_E$ is an algorithm of a secure VRF (Appendix D). With this, players are able to discover the definitive iteration only at iteration $r^* + 1$, since they can reconstruct $H$ and evaluate $H(0) = 0$. This delay of 1 iteration from $r^*$ results in a *computational Nash equilibrium*. The second component, i.e., protocol compliance checking results in a further *computationally strict Nash equilibrium* as players can use the VRF to check any deviations in transmissions from the protocol. However, implementing [8]'s protocol directly in a general network setting results in some problems, such as:

1.  The protocol of [8] assumes that the dealer is able to send shares/secret keys to each player directly at the beginning of the game in the share/key generation and dissemination phase. In a general network, the dealer may not have this ability, and as described in the previous section, the protocol of [8] may lead the dealer to concentrate transmissions to some player nodes.
2.  In addition, with rational participants, the action space widens in the first key dissemination phase. For instance, players may maul the share/secret keys from the dealer or refrain from sending the share/secret keys to the desired recipients. Given this larger action space of players, it is not clear if a certain combination of the SMT protocol to the protocol of [8] would result directly in an equilibrium, and additional mechanisms may be needed. In particular, in Appendix E.1, we show how a certain combination of the SMT protocol with [8] over an instance of a general network results in a strategy that is dominated by some other strategy.
3.  Moreover, in the secret reconstruction phase, point-to-point transmissions between players may not be available, and transmissions may have to pass through intermediate players. As a result, some players may maul or modify transmissions along the way. Once again, it is not clear if [8]'s protocol would still induce an equilibrium under this enlarged action space of players in the secret reconstruction phase.

To fix the preceding issues, one way for equilibrium to be preserved in a general network is to include additional coordination mechanisms among participants. However, additional coordination mechanisms imply that there have to be additional *protocol compliance checking* steps in order for a player to check if all other players are indeed following the coordination mechanism. Bearing these in mind, we developed the following approach for our protocols $\Pi_1, \Pi_2, \Pi_{2.1}$—as described from a high level.

1.  To guarantee computational Nash equilibrium under rational players in the share generation/key dissemination phase, we include the additional mechanism by which the dealer includes in its messages an explicit set of instructions referring to the path by which the message will be delivered. Together with this, we implement a form of protocol compliance checking by which each player receives several duplicate messages from the dealer sent along $k$-disjoint paths. If any player sees a discrepancy from messages it received, it knows that some player deviated from the protocol, and it is able to abort immediately. We note that this mechanism also prevents concentration of transmissions from the dealer.
2.  In the secret reconstruction phase, for our first proposed protocol ($\Pi_1$), we force the players to duplicate their transmissions along $k$-disjoint paths as another form of protocol compliance checking. This way, players are able to check if all duplicates

they received are equal. If any player sees a discrepancy, it is able to abort since this indicates that some other player deviated from the protocol (i.e., by modifying or mauling a transmission along the way). However, for $\Pi_1$, without access to a VRF (see Appendix D) for all participants, the dealer needs to be online in the secret reconstruction phase in order to impose strict protocol compliance checking in all players (As noted in Lemma 2).

3. In the secret reconstruction phase, for our next protocols, $(\Pi_2)$ and $(\Pi_{2.1})$, we implement a VRF in order to achieve the same type of protocol compliance checking as $\Pi_1$, but with lower communication complexity under a semi-online dealer. However, compared to $\Pi_1$, the dealer in $\Pi_2$ and $\Pi_{2.1}$ includes a specific set of instructions by which players would send their transmissions to each other.

4. Finally, we implement uncertainty in the definitive stage by letting players discover the definitive iteration $r^*$ only at iteration $r^* + 1$. This is done using a pseudorandom function (see Appendix C) and random polynomials in $\Pi_1$, and through a secure VRF with the pseudorandom property in $\Pi_2$ and $\Pi_{2.1}$ following [8]. Moreover, the number of rounds in each iteration in $\Pi_1$, $\Pi_2$, and $\Pi_{2.1}$ are fixed a priori in order for players to synchronize and know when an iteration begins and when it ends, and by which it can unambiguously determine in a finite amount of time if some player deviated from the protocol by not sending any needed transmission, or when the definitive iteration has already been reached.

This combination of protocol compliance checking and uncertainty on the definitive stage results in an equilibrium for $\Pi_1, \Pi_2$, and $\Pi_{2.1}$, as we state in Theorems 1–6.

*4.3. Proposed Protocol $\Pi_1(n,k)$: With Online Dealer*

We now proceed to describe the first proposed protocol of this paper. This protocol ($\Pi_1$) uses a standard pseudorandom function (as defined in Appendix C) along with the Shamir secret sharing scheme (as defined in Appendix B) in order to achieve computational Nash equilibrium (and also leakage-tolerant equilibrium) in a general network whose corresponding graph is a $k$-path-disjoint. This is our first attempt to come up with a secret sharing protocol that can operate over a specific general network given rational participants. The protocol $\Pi_1$, however, assumes that the dealer is online. This requirement will be relaxed in the succeeding protocol $\Pi_2$.

Given a security parameter $\kappa \in \mathbb{N}$, denote by $\nu := \nu(\kappa)$ the value of a polynomial in $\kappa$. Let $(S_G, S_R)$ correspond to polynomial-time algorithms that give a secure $(n, k)$ Shamir Secret Sharing scheme, where $S_G : \{0,1\}^\kappa \to \{0,1\}_1^\nu \times \{0,1\}_2^\nu \times \cdots \times \{0,1\}_k^\nu$ and $S_R : \{0,1\}_1^\nu \times \{0,1\}_2^\nu \times \cdots \times \{0,1\}_k^\nu \to \{0,1\}^\kappa$. Let $\Lambda : \{0,1\}^\nu \times \{0,1\}^\nu \to \{0,1\}^\nu$ denote a standard secure pseudorandom function. Let $\mathcal{G}$ be an $A_{GN}$ rational secret sharing game associated with a $k \leq n$-path-disjoint graph $G(V,E)$ and domain $\mathcal{S} := \{0,1\}^\nu$, with $n+1$ participants consisting of a dealer $d$ and $n$ players $\{p_i\}_{i \in [n]} := N$. Given $k \leq n$, the first protocol proposed in this paper, $\Pi_1(n,k)$, is described as follows, which assumes that the dealer is online.

**Protocol. $\Pi_1(n,k)$.**

**Phase 0. Dealer Initialization//Secret Generation.** The dealer $d$ performs the following to share a secret $s \in \{0,1\}^\nu$:

1. Choose $r \in \mathbb{N}$ according to a geometric distribution with parameter $\beta$;
2. Generate secret keys $\{sk_1, sk_2, \ldots, sk_n\}$;
3. For $i \in [n]$, the dealer computes $\{s_{i,1}, s_{i,2}, \ldots, s_{i,k}\} = S_G(sk_i)$;
4. Choose random $(n-1)$-degree polynomials $G \in \mathbb{F}_{2^\nu}[x]$ and $H \in \mathbb{F}_{2^\nu}[x]$ with $G(0) = s$ and $H(0) = 0$;
5. Compute $\{g_i^* := G(i) \oplus \Lambda(sk_i, r^*)\}_{i \in [n]}$ and $\{h_i^* := H(i) \oplus \Lambda(sk_i, r^* + 1)\}_{i \in [n]}$.

**Phase 1. Keys dissemination**. Let $s^0$ be some uniformly sampled number from $\{0,1\}^\nu$ for each player $p_i$, $i \in [n]$. Let `max_l` denote the length of the longest path between any pair of nodes in $G$:

1.  For $i \in [n]$, and for $j \in [k]$, the dealer computes $\{s_{i,1}, s_{i,2}, \ldots, s_{i,j}, \ldots, s_{i,k}\} \leftarrow S_G(sk_i)$. Afterwards, the dealer $d$ selects arbitrary $k$ disjoint paths from $d$ to $p_i$, and each path is given a *path encoding* corresponding to $\text{path}_{i,j} := (a_0 = d, a_1, a_2, \ldots, a_m = p_i)$ for $j \in [k]$ and for some $m \leq \text{max}_l$. The dealer $d$ sends $\{(s_{i,j}, \{\text{path}_{i,j}\}_{j \in [k]}, \{g_i^*\}_{i \in [n]}, \{h_i^*\}_{i \in [n]})\}_{j \in [k]}$ to $p_i$ along the $k$ disjoint paths from $d$ to $p_i$.
2.  For $i \in [n]$, if $p_i$ received a transmission from some other node $p_{i'}$ containing $\{\text{path}_{i,j}\}_{j \in [k]}$, it checks if its own node is actually in a path encoding corresponding to $\text{path}_{i,j}$ for some $j \in [k]$ (this is unique given that the $k$ paths are disjoint). If not, $p_i$ outputs $s^0$ and aborts. If true, $p_i$ checks if it is meant to receive a transmission from $p_{i'}$. If not, $p_i$ outputs $s^0$ and aborts. Otherwise, if $p_i$ is the end-receiver according to $\text{path}_{i,j}$, it keeps the transmission. If $p_i$ is not the end-receiver, it sends the transmission to the next node according to $\text{path}_{i,j}$.
3.  For $i \in [n]$, if $p_i$ did not receive exactly $k$ tuples of the form

    $$(s_{i,j}, \{\text{path}_{i,j}\}_{j \in [k]}, \{g_i^*\}_{i \in [n]}, \{h_i^*\}_{i \in [n]})$$

    after `max_l` rounds such that the origin-node of each path encoding is $d$ and the end-node is $p_i$, it outputs $s^0$ then aborts. Otherwise, it verifies that all copies of $\{\text{path}_{i,j}\}_{j \in [k]}, \{g_i^*\}_{i \in [n]}$ and $\{h_i^*\}_{i \in [n]}\}_{j \in [k]}$ it received are equal. If not, it outputs $s^0$ then aborts. Otherwise, it reconstructs $sk_i = S_R(s_{i,1}, s_{i,2}, \ldots, s_{i,k})$.
4.  After `max_l` rounds, if all checks in (3) above do not fail, all participants move on to phase 2.

**Phase 2. Secret Reconstruction**. For iteration $r = 1, 2, \ldots$, the players and the dealer perform the following (where Phase 2.0 can be performed simultaneously with Phase 2.1):

1.  **Phase 2.0**: Dealer transmits as origin-node to each player.

    (a)  The dealer computes $h' = \oplus_{i \in [n]} \Lambda(sk_i, r)$. Afterwards, the dealer selects arbitrary $k$ disjoint paths from $d$ to $p_i$, where each path is given a path encoding corresponding to $\text{path}_{i,j} := (a_0 = d, a_1, a_2, \ldots, a_m = p_i)$ for $j \in [k]$ and for some $m \leq \text{max}_l$. The dealer $d$ sends $\{\{\text{path}_{i,j}\}_{j \in [k]}, h'\}_{j \in [k]}$ to $p_i$ along the selected $k$ disjoint paths from $d$ to $p_i$.

2.  **Phase 2.1**: Players transmit information to each other.

    (a)  For $i \in [n]$, if $p_i$ received any transmission from some other node $p_{i'}$ containing a path encoding, it checks if its own node is actually in the encoded path, and if it is meant to receive a transmission from $p_{i'}$. If any of these are not true, it outputs $s^{r-1}$ and aborts. Otherwise, if $p_i$ is the end-receiver according to the path encoding, it keeps the transmission. If $p_i$ is not the end-receiver, it sends the transmission to the next node according to the path encoding.

    (b)  For $i \in [n]$, if $p_i$ does not receive exactly $k$ sets of information of the form $(\{\text{path}_{i,j}\}_{j \in [k]}, h')$, such that the origin-node of each $\text{path}_{i,j}$ for $j \in [k]$ is $d$ and the end-node is $p_i$ after `max_l` rounds, it outputs $s^{r-1}$ then aborts. Otherwise, it verifies that all $k$ copies of $(\{\text{path}_{i,j}\}_{j \in [k]}, h')$ it received are equal. If not, it outputs $s^{r-1}$ then aborts.

    (c)  For $i \in [n]$, $p_i$ computes $g_i^r = \Lambda(sk_i, r)$ and $h_i^r = \Lambda(sk_i, r + 1)$. For every other player $p_l$, ($l \in [n]$, $i \neq l$), $p_i$ selects arbitrary $k$ disjoint paths from $p_i$ to $p_l$, where each path is given an encoding corresponding to $\text{path}_{l,j} := (a_0 = p_i, a_1, a_2, \ldots, a_m = p_l)$ for some $m \leq \text{max}_l$. Afterwards, $p_i$ sends

    $$\{(\{\text{path}_{l,j}\}_{j \in [k]}, g_i^r, h_i^r)\}_{j \in [k]}$$

to $p_l$ along the selected $k$ disjoint paths for all other players $p_l$, $l \in [n] \setminus i$.

(d) For $i \in [n]$, and for $l \in [n] \setminus i$, $p_i$ checks if it has received (within `max_1` rounds)) exactly $k$ tuples of the form $(\{\text{path}_{l,j}\}_{j \in [k]}, g_l^r, h_l^r)$ ($j \in [k]$) such that the origin-node of each path encoding is $p_l$ and the end-node is $p_i$. If not, $p_i$ outputs $s^{r-1}$ then aborts. Otherwise, for $l \in [n] \setminus i$, it verifies that all $k$ copies of $(\{\text{path}_{i,j}\}_{j \in [k]}, g_i^r, h_i^r)$ it received (whose origin-node is $p_l$) are equal. If not, $p_i$ outputs $s^{r-1}$ then aborts. Otherwise, once $p_i$ receives information from all players, $p_i$ checks if $\oplus_{i \in [n]} h_i^r = h'$. If not, $p_i$ outputs $s^{r-1}$ then aborts.

Otherwise, $p_i$ computes $\{h_i^p := h_i^* \oplus h_i^r\}_{i \in [n]}$. It then interpolates an $n-1$ polynomial $H^r$ using $\{h_i^p\}_{i \in [n]}$ and checks if $H^r(0) = 0$. If $H^r(0) = 0$, it outputs $s^{r-1}$ then halts. Otherwise, it computes $\{g_i' := g_i^* \oplus g_i^r\}_{i \in [n]}$, then interpolates an $n-1$-degree polynomial $G^r$ using $\{\hat{g}_i\}_{i \in [n]}$. Afterwards, it sets $s^r = G^r(0)$.

3. After `max_1` rounds, if all checks above do not fail for any participant, all participants move on to the next iteration of phase 2.

Intuitively, the protocol $\Pi_1$ works by using redundancies in paths provided by the $k$-path-disjoint graph $G$ as shown in Figure 3. Since $G$ is $k$-path-disjoint, any transmission from either the dealer or a player to another player has to pass through $k$ disjoint paths. In phase 1, the dealer breaks the share of each player into $k$ pieces using the Shamir Secret Sharing scheme and sends these $k$ pieces along $k$ disjoint paths. Any player that sees a piece of a share does not have $k-1$ other pieces and cannot reconstruct the secret key by himself. Moreover, each transmission contains a copy of the path encoding and the public keys $\{g_i^*\}_{i \in [n]}$ and $\{h_i^*\}_{i \in [n]}$. Given that each player acquires $k$ copies of a transmission, it knows that the path encoding and $\{g_i^*\}_{i \in [n]}$ and $\{h_i^*\}_{i \in [n]}$ are correct if all $k$ copies of them match. This provides incentives for players not to deviate from $\Pi_1$ by modifying any content of a transmission in phase 1 given that they know such behaviour will be detected. This renders $\Pi_1$ secure against $k-1$-sized coalitions given that, as per Lemma 1, any set of $k$ transmissions from one player to another has to pass through at least one path not belonging to the coalition, and any deviations by the coalition will be detected. In addition, the dealer uses an $n$-degree polynomial in phase 0 to make it secure against $n-1$ secret key leakage (which is inspired by a note in [8]).

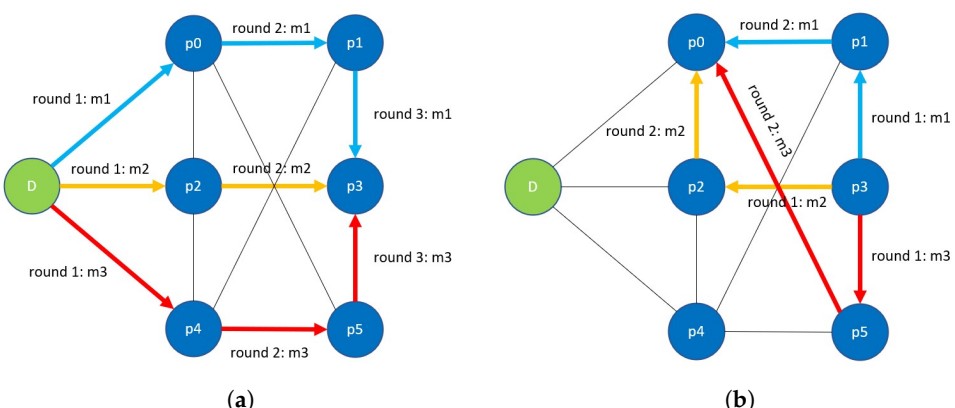

(a)          (b)

**Figure 3.** The graphs in (**a**,**b**) show a ($k = 3$)-path disjoint network graph `max_1 = 3`. The left figure (**a**) shows an example of the dealer (green node) $d$ sending messages $m1$, $m2$, $m3$ to player $p_3$ (a blue node) along 3 disjoint paths. In phases 1 and 2.1 of protocol $\Pi_1$, we have $m1 = m2 = m3$, so that $p_3$ should receive 3 copies of the same message by the 3rd round. The right figure (**b**) shows an example of a player ($p_3$) sending messages $m1 = m2 = m3$ to player $p_0$ along 3-disjoint paths, which corresponds to the steps performed by each player in phase 2.1 of $\Pi_1$.

For phase 2, the same reasoning applies, whereby the dealer sends a *check variable h'* to each player along $k$ disjoint paths, and each player sends a transmission of the form

$(\{\texttt{path}_{l,j}\}_{j\in[k]}, g_l^r, h_l^r)$ for some $l \in [n]$ and $j \in [k]$ to all other players along $k$ disjoint paths. By the same principle, players can use the $k$ copies received from each player to verify the correctness of the transmission. We note that in $\Pi_1$, the check variable $h'$ is crucial for verifying the correctness of the transmission given that, without $h'$, some strategy strictly dominates $\Pi_1$, as shown in the following Lemma.

**Lemma 2.** *Without the check $\oplus_{i\in[n]} h_i^r \neq h'$ in step 2.d of $\Pi_1(n,k)$, there exists a polynomial-time strategy for $p_i$ that strictly dominates $\Pi_1$, assuming all other players follow strategies prescribed by $\Pi_1$.*

**Proof.** Let $p_i$ take the following strategy: follow $\Pi_1$ in all aspects, except that $p_i$ changes $h_i^r$ to some random number then sends it to all other players. Other players will not detect this since the check $\oplus_{i\in[n]} h_i^r \neq h'$ is not implemented. With non-negligible probability, at $r = r^* + 1$, all other players will have $H^r(0) \neq 0$ given that they did not receive the real $h_i^r$ from $p_i$. However, $p_i$ will know that the current iteration is $r^* + 1$ since it has the real $h_i^r$ needed to interpolate the correct polynomial $H^r$ such that $H^r(0) = 0$. $p_i$ would then output $G^r(0) = s$ and receive utility $U^+$ (given that all other players are not aware that $r = r^* + 1$). $\square$

Finally, the equilibrium of $\Pi_1$ relies on the fact that players are not aware of the value of $r^*$ until they reach iteration $r^* + 1$ following [8]. This generates uncertainty among the players such that, given a sufficiently low parameter $\beta$ in the geometric distribution from which $r^*$ is sampled, players prefer to follow $\Pi_1$ rather than deviate. Given this, the following results regarding $\Pi_1$ arrive at whose proofs are in the Annex.

**Theorem 1.** *Given $\kappa \in \mathbb{N}$, let $\nu := \nu(\kappa)$ denote the value of a polynomial in $\kappa$. Let $\mathcal{G}$ be an $A_{GN}$ game with $n+1$ participants associated with a k-path-disjoint graph $G(V, E)$ for $k \leq n$ and domain $\mathcal{S} := \{0,1\}^\nu$. The protocol $\Pi_1(n,k)$ is a computational Nash equilibrium, and is also an $(n-1)$-key leakage-tolerant equilibrium provided that $[(\beta \times U^+) + (1-\beta) \times U_{\texttt{rand}} - U] < 0$, where $\beta$ is the parameter of a geometric distribution. Given a maximum path length of $\texttt{max\_l}$ in G, the average round complexity of $\Pi_1(n,k)$ is $[1 + (1/\beta)] \times \texttt{max\_l}$, with a communication complexity of at most $n \times \nu \times (k + 2n + 1)$ per round.*

**Theorem 2.** *Given $\kappa \in \mathbb{N}$, let $\nu := \nu(\kappa)$ denote the value of a polynomial in $\kappa$. Let $\mathcal{G}$ be an $A_{GN}$ game with $n+1$ participants associated with a k-path-disjoint graph $G(V, E)$ for $k \leq n$ and domain $\mathcal{S} := \{0,1\}^\nu$. The protocol $\Pi_1(n,k)$ is a computational strict Nash equilibrium provided that $[(\beta \times U^+) + (1-\beta) \times U_{\texttt{rand}} - U] < 0$.*

**Theorem 3.** *Given $\kappa \in \mathbb{N}$, let $\nu := \nu(\kappa)$ denote the value of a polynomial in $\kappa$. Let $\mathcal{G}$ be an $A_{GN}$ game with $n+1$ participants associated with a k-path-disjoint graph $G(V, E)$ for $k \leq n$ and domain $\mathcal{S} := \{0,1\}^\nu$. Suppose that no player can acquire other secret keys unless information related to it is shared by another player through a transmission. The protocol $\Pi_1(n,k)$ is a $(k-1)$-resilient computational Nash equilibrium provided that $[(\beta \times U^+) + (1-\beta) \times U_{\texttt{rand}} - U] < 0$.*

### 4.4. Proposed Protocol $\Pi_2(n,k)$: With Semi-Online Dealer

We now proceed to describe the second proposed protocol ($\Pi_2$) of this paper, which does not require an online dealer but only a semi-online one. Due to this limitation, compared to $\Pi_1$, this protocol requires an additional VRF cryptographic primitive (as defined in Appendix D). $\Pi_2$ is inspired by the protocol of [8] (see Appendix E), but $\Pi_2$ includes several additional steps in order to accommodate a general network topology over the participants. Thus, given a graph $G(V, E)$, assume that it is $k$-path-disjoint. The protocol assumes that for each pair $a, b \in V$ representing distinct nodes of participants in the game, any transmission from $a$ to $b$ will be sent through $k$ disjoint paths connecting $a$ and $b$ according to some order that could be known by all participants using a publicly known

polynomial-time algorithm. For this purpose, we define two types of ordering termed `path_ordering` and `transmission_ordering` as follows:

**Definition 22.** *Given a graph $G(V, E)$ and a positive integer $k$, a* `path_ordering` *from $a$ to $b$, with $a, b \in V$, $a \neq b$, is a unique sequence of $k$ disjoint paths from the origin-node $a$ to the end-node $b$ that can be efficiently constructed given some rule on the choice of paths.*

**Definition 23.** *Given an $A_{GN}$ game $\mathcal{G}$ with $n + 1$ participants associated with a graph $G(V, E)$, a* `transmission_ordering` *for $\mathcal{G}$ is a unique sequence of paths that can be efficiently constructed given: (1) a rule on the ordering of pairs of distinct nodes in $V$ and (2) a* `path_ordering` *for each distinct pair of origin-nodes and end-nodes. In addition,* `transmission_ordering` *marks the origin-nodes and end-nodes of each path in* `path_ordering` *with special symbols to differentiate them from nodes that are intermediate along the path.*

**Example 1.** `path_ordering`: *Let $k > 0$ and let $G(V, E)$ be a $k$-path-disjoint graph with $|V| > k$. An example of a* `path_ordering` *for each distinct pair $(a, b)$ of nodes in $V$ is given by the following polynomial-time algorithm that operates according to a lexicographic rule:* `step 1`: *on input $(G, a, b)$, set* `path_ordering` $= \varnothing$; `step 2`: *given $a, b$ list down all paths (not necessarily disjoint) in $G$ from $a$ to $b$;* `step 3`: *obtain the lexicographically first path from $a$ to $b$ in the list and include it in* `paths`, *then remove all nodes crossed by the path from $G$ to arrive at a residual graph $G'$; using $G'$, repeat* `step 2`–`step 3` *until $k$ disjoint paths from $a$ to $b$ are in* `path_ordering`.

**Example 2.** `transmission_ordering`: *Let $k > 0$, and let $G(V, E)$ be a $k$-path-disjoint graph with $|V| > k$. Let* `path_ordering` *be the same as in the prior example. Let $\mathcal{G}$ be an $A_{GN}$ game with $|V| = n + 1$ participants, such that the nodes $V = \{a_0, a_1, a_2, \ldots, a_{|V|}\}$ of $G$ are assigned as follows: $a_0 = d$ (the dealer), $a_1 = p_1$ (player 1), $a_2 = p_2$ (player 2), etc. An example of a* `transmission_ordering` *for $\mathcal{G}$ is given by the following polynomial-time algorithm:* `step 1`: *On input $G$, set* `transmission_ordering` $= \varnothing$. `step 2`: *construct the set* `pairings` *as follows, set the first pair in* `pairings` *as $(a_0, a_1)$, followed by a second pair $(a_0, a_2)$, etc., up to the nth pair $(a_0, a_n)$. After the nth pair, set the $n + 1$th pair as $(a_1, a_2)$, then the $n + 2$th pair as $(a_1, a_3)$, etc., up to $(a_1, a_n)$. Afterwards, the next pair is $(a_2, a_1)$ followed by $(a_2, a_3)$, etc., and so on and so forth so that $a_0$ (at the left of a pair) is paired with $n$ other nodes (at the right of a pair), and each player node (at the left of a pair) is paired with $n - 1$ other player nodes (at the right of a pair).* `step 3`: *for each pair in* `pairings`, *compute* `path_ordering` *using the algorithm in the example above and include* `path_ordering` *in* `transmission_ordering`, *where the origin-node and end-node of each path in* `path_ordering` *are assigned special symbols.*

Given common knowledge on the structure of $G(V, E)$ and the rules (i.e., polynomial-time algorithms) for constructing `transmission_ordering`, each player in the game can construct `transmission_ordering` in polynomial-time on his own at the start of the game. In the protocol $\Pi_2$ below, only one participant is meant to send a transmission for each round. The participant to send a transmission is the origin-node in the paths of `transmission_ordering`, and the protocol prescribes participants to follow the transmission ordering contained in `transmission_ordering` according to the edges listed in its paths, where each edge in a path corresponds to one round of transmission. With this rule, each participant in the game knows whose turn it is to send or receive a transmission given a certain round. It follows that a participant can verify if it received or sent information according to the protocol or not. Given this, we now proceed to describe $\Pi_2$. Given a security parameter $\kappa \in \mathbb{N}$, denote by $\nu := \nu(\kappa)$ the value of a polynomial in $\kappa$. Let $(V_G, V_E, V_P, V_V)$ correspond to polynomial-time algorithms that give a secure Verifiable Random Function scheme, where $V_G : 1^* \to \{0, 1\}^\nu \times \{0, 1\}^\nu$, $V_E : \{0, 1\}^\nu \times \{0, 1\}^\nu \to 0, 1^\nu$, $V_P : \{0, 1\}^\nu \times \{0, 1\}^\nu \to \{0, 1\}^\nu$, and $V_V : \{0, 1\}^\nu \times \{0, 1\}^\nu \times \{0, 1\}^\nu \times \{0, 1\}^\nu \to \{0, 1\}$. Let $\beta$ be a parameter of a geometric distribution that is independent of $\kappa$. Let $\mathcal{G}$ be an $A_{GN}$ rational secret sharing game associated with a $k$-path-disjoint graph $G(V, E)$ and domain

$\mathcal{S} := \{0,1\}^\nu$, with $n+1$ participants consisting of a dealer $d$ and $n$ players $\{p_i\}_{i \in [n]} := N$. The second protocol proposed in this paper, $\Pi_2(n,k)$ is described as follows.

**Protocol.** $\Pi_2(n,k)$.

**0. Initialization Phase**. The dealer performs the following to share a secret $s \in \{0,1\}^\nu$:

1. Choose $r^* \in \mathbb{N}$ according to a geometric distribution with parameter $\beta$;
2. Generate public and secret key pairs $(pk_1, sk_1), (pk_2, sk_2), \ldots, (pk_n, sk_n) \leftarrow V_G(1^\kappa)$;
3. Generate public and secret key pairs $(pk'_1, sk'_1), (pk'_2, sk'_2), \ldots, (pk'_n, sk'_n) \leftarrow V_G(1^\kappa)$;
4. Choose random $(n-1)$-degree polynomials $G \in \mathbb{F}_{2^\nu}[x]$ and $H \in \mathbb{F}_{2^\nu}[x]$ such that $G(0) = s$ and $H(0) = 0$;
5. Compute $\{g_i^* := G(i) \oplus V_E(sk_i, r^*)\}_{i \in [n]}$ and $\{h_i^* := H(i) \oplus V_E(sk_i, r^* + 1)\}_{i \in [n]}$;
6. Construct `transmission_ordering_a` by listing down $k$ disjoint paths from $d$ to $p_1$ according to `path_ordering` followed by $d$ to $p_2$, then $d$ to $p_3$, etc., up to $d$ to $p_n$, such that in each path in `transmission_ordering_a` the origin-node $d$ is marked with a special symbol `start` and the end-node of each path is marked with a special symbol `end`;
7. Construct `transmission_ordering_b` by listing down one arbitrarily chosen path for each pair of players starting with a path from $p_1$ to $p_2$, followed by a path from $p_1$ to $p_3$, etc., up to $p_1$ to $p_n$. Afterwards, list down a path from $p_2$ to $p_1$, followed by a path from $p_2$ to $p_3$, etc. (The algorithm for `path_ordering` is not needed for `transmission_ordering_b`.) In each path in `transmission_ordering_b`, the origin-node is marked with a special symbol `start`, and the end-node of each path is marked with a special symbol `end`;
8. Define the tuple of public information as:

$$\Psi = (\{pk_i\}_{i \in [n]}, \{pk'_i\}_{i \in [n]}, \{g_i\}_{i \in [n]}, \{h_i\}_{i \in [n]}, \quad \texttt{transmission\_ordering\_a},$$
$$\texttt{transmission\_ordering\_b}).$$

**1. Public Information dissemination Phase**. Let $s^0 \in \{0,1\}^\nu$ be a uniformly drawn number for each player $p_i \in N$:

1. For $i \in [n]$ and for $j \in [k]$, $d$ sends $\Psi$ to $p_i$ according to `transmission_ordering_a`.
2. For $i \in [n]$, if $p_i$ does not yet have $\Psi$ and receives it for the first time, it checks if it is meant to receive $\Psi$ according to `transmission_ordering_a` $\in \Psi$. If not, it outputs $s^0$ then aborts. Otherwise, it keeps the information if it is its turn to receive it (i.e., its own node is marked with `end`), or sends the transmission to the respective node dictated by `transmission_ordering_a`.
3. For $i \in [n]$, if $p_i$ has a prior copy of $\Psi$ (received from some previous round), it checks if it is meant to receive (or not receive) a transmission from some other node according to `transmission_ordering_a` in terms of the current round. If there is a violation, it outputs $s^0$ then aborts. Otherwise, if it received information, $p_i$ verifies if all of its copies of $\Psi$ are so far equal. If not, it outputs $s^0$ then aborts. Otherwise, it keeps $\Psi$ if it is its turn to receive it (i.e., its own node is marked with `end`), or sends the transmission to the respective node dictated by `transmission_ordering_a`.
4. For $i \in [n]$, if $p_i$ still does not receive $k$ copies of $\Psi$ as dictated by `transmission_order -ing_a` within `max_1` $\times n \times k$ rounds, it outputs $s^0$ then aborts. Otherwise, it verifies that all $k$ copies of $\Psi$ it received are equal. If not, it outputs $s^0$, then aborts.
5. After `max_1` $\times n \times k$ rounds, if all checks above do not fail for any participant, all participants move on to phase 2.

**2. Secret Key dissemination Phase**.

1. For $i \in [n]$, the dealer computes $\{s_{i,1}, s_{i,2}, \ldots, s_{i,k}\} = S_G(sk_i)$ and $\{s'_{i,1}, s'_{i,2}, \ldots, s'_{i,k}\} = S_G(sk'_i)$.

2. For $i \in [n]$ and for $j \in [k]$, $d$ sends $\{s_{i,j}, s'_{i,j}\}$ to the end-receiver $p_i$ according to `transmission_ordering_a`.

3. For $i \in [n]$, if $p_i$ receives or does not receive a transmission from some other node in violation of `transmission_ordering_a` in terms of the current round, it outputs $s^0$ then aborts. Otherwise, it keeps the information if it is its turn to receive it (i.e., its own node is marked with end) or sends the transmission to the respective node as dictated by `transmission_ordering_a`.

4. For $i \in [n]$, if $p_i$ still does not receive $k$ sets of information (following the transmissions dictated by `transmission_ordering_a`) within $\texttt{max\_l} \times n \times k$ rounds, it outputs $s^0$ then aborts. Otherwise, given $\{s_{i,j}\}_{j \in [k]}$ and $\{s'_{i,j}\}_{j \in [k]}$, it reconstructs $sk_i = S_R(s_{i,1}, s_{i,2}, \ldots, s_{i,k})$ and $sk'_i = S_R(s'_{i,1}, s'_{i,2}, \ldots, s'_{i,k})$.

5. After $\texttt{max\_l} \times n \times k$ rounds, if all checks above do not fail for any participant, all participants move on to phase 3.

**3. Reconstruction Phase**.

1. Given `transmission_ordering_b`, for $i \in [n]$, if it is $p_i$'s turn to transmit as the origin-node for the first time (i.e., its node is marked with start for the first time), $p_i$ computes the following:

$$y_i^r = V_E(sk_i, r), z_i^r = V_E(sk'_i, r)$$

$$\pi_i^r = V_P(sk_i, r), \psi_i^r = V_P(sk'_i, r)$$

Afterwards, $p_i$ sends $(g_i^r, h_i^r)$ to all other players $\{p_{i'}\}_{i' \in [n] \setminus i}$ according to the transmissions dictated in `transmission_ordering_b`.

2. For $i \in [n]$, if $p_i$ receives or does not receive a transmission from some other node in violation of `transmission_ordering_b` in terms of the current round, it outputs $s^{r-1}$ then aborts. Otherwise, if its node is not marked with end (following `transmission_ordering_b`), it sends the transmission to the respective receiver node as dictated by `transmission_ordering_b`. However, if it is $p_i$'s turn to receive information (i.e., its node is marked with end), it sets source as the index of the origin-node of the transmission, i.e., the transmission originates from player $p_{\texttt{source}}$. Afterwards, it performs the following:

   (a) Check if the information received is of the form $(y^r, z^r, \pi^r, \psi^r)$. If not true, output $s^{r-1}$ and abort.

   (b) Verify that both $V_V(pk_{\texttt{source}}, r, y^r, \pi^r)$ and $V_V(pk_{\texttt{source}}, r, z^r, \psi^r)$ are true. If any of these are false, abort.

   (c) Check if $n$ tuples of the form $(y_{i'}^r, z_{i'}^r, \pi_{i'}^r, \psi_{i'}^r)$ for indices $i' \in [n]$ have so far been acquired. If true, let $I$ denote the player indices corresponding to such tuples. Compute $h_{i'}^r := h_{i'} \oplus z_{i'}^r$ for all $i' \in I$, and interpolate a $(n-1)$-degree polynomial $H^r$ using $\{h_{i'}^r\}_{i' \in I}$. If $H^r(0) = 0$, output $s^{r-1}$ immediately as the computed secret and abort.

   (d) Otherwise, if $H^r(0) \neq 0$ in the above item, compute $s_i^r$ as follows: set $g_{i'}^r := g_{i'} \oplus y_{i'}^r$ for all $i' \in I$. Interpolate a $(n-1)$-degree polynomial $G^r$ through $\{g_r^{i'}\}_{i' \in I}$ and set $s_i^r := G^r(0)$.

3. For $i, i' \in [n]$, if $p_i$: (a) did not receive any transmission from some other *origin-node* $p_{i'}$ ($i' \neq i$) according to `transmission_ordering_b` within $\texttt{max\_l} \times n^2 \times k$ rounds, , it outputs $s^{r-1}$ then aborts.

4. After $\texttt{max\_l} \times n^2 \times k$ rounds, if all checks above do not fail for any participant, all participants move on to the next iteration in phase 3.

Phases 1–2 of $\Pi_2$ follow the same principle as that of phase 1 in $\Pi_1$, whereby, given that $G$ is $k$-path-disjoint, participants take advantage of the $k$ disjoint paths for each pair

of nodes in $G$ in order to transmit redundant information. With this, players can check the correctness of the transmitted data by comparing the $k$ copies to each other. In phase 3 of $\Pi_2$, however, instead of using $k$ disjoint paths to transmit information, they use the properties of the VRF to verify that received data are correct. The absence of redundancy in phase 3 of $\Pi_2$ enables $\Pi_2$ to have less communication complexity than $\Pi_1$. The following results regarding $\Pi_2$ are arrived at, whose proofs are in the Appendix.

**Theorem 4.** *Given $\kappa \in \mathbb{N}$, let $\nu := \nu(\kappa)$ denote the value of a polynomial in $\kappa$. Let $\mathcal{G}$ be an $A_{GN}$ game with $n + 1$ participants associated with a k-path-disjoint graph $G(V, E)$ for $k \leq n$, and domain $\mathcal{S} := \{0, 1\}^\nu$. The protocol $\Pi_2(n, k)$ is a computational Nash equilibrium, and is also a $(n - 1)$-key leakage-tolerant equilibrium provided that $[(\beta \times U^+) + (1 - \beta) \times U_{\texttt{rand}} - U] < 0$, where $\beta$ is the parameter of a geometric distribution. The average round complexity of $\Pi_2(n, k)$ is $[2 \times \texttt{max\_l} \times n \times k] + [(1 + 1/\beta) \times \texttt{max\_l} \times n^2]$, and the communication complexity per round is at most $O(6n\nu)$.*

**Theorem 5.** *Given $\kappa \in \mathbb{N}$, let $\nu := \nu(\kappa)$ denote the value of a polynomial in $\kappa$. Let $\mathcal{G}$ be an $A_{GN}$ game with $n + 1$ participants associated with a k-path-disjoint graph $G(V, E)$ for $k \leq n$ and domain $\mathcal{S} := \{0, 1\}^\nu$. The protocol $\Pi_2(n, k)$ is a computationally strict Nash equilibrium provided that $[(\beta \times U^+) + (1 - \beta) \times U_{\texttt{rand}} - U] < 0$.*

**Theorem 6.** *Given $\kappa \in \mathbb{N}$, let $\nu := \nu(\kappa)$ denote the value of a polynomial in $\kappa$. Let $\mathcal{G}$ be an $A_{GN}$ game with $n + 1$ participants associated with a k-path-disjoint graph $G(V, E)$ for $k \leq n$ and domain $\mathcal{S} := \{0, 1\}^\nu$. Suppose that no player can acquire other secret keys unless information related to it is shared by another player through a transmission. The protocol $\Pi_2(n, k)$ is a $(k - 1)$-resilient computational Nash equilibrium provided that $[(\beta \times U^+) + (1 - \beta) \times U_{\texttt{rand}} - U] < 0$.*

Proposed Protocol $\Pi_{2.1}(n, k)$: With Dealer Connected Directly to Each Player

The last protocol of this paper $\Pi_{2.1}$ induces a $\Phi$-resilient computational Nash equilibrium, where $\Phi$ is the condition that a subset of nodes be 1-disconnected. The idea behind this protocol is to provide some equilibrium notions that allow for certain large-sized coalitions to be formed, contrary to the usual equilibrium notion where all coalitions are bounded by $k$. However, unlike $\Pi_2$, the dealer is assumed to be directly connected to each player in $\Pi_{2.1}$ so that it can transmit shares and keys in one simultaneous move. Given this advantage, protocol $\Pi_{2.1}$ performs additional checks, whereby any transmission received by a node is checked for correctness. Given that any coalition is 1-disconnected, any transmission among members of the coalition have to pass through at least one player not belonging to the coalition, such that any deviations from the protocol will be checked. This prevents members of the coalition to share information outside of $\Pi_{2.1}$ to each other—in particular, secret keys.

**Protocol.** $\Pi_{2.1}(n, k)$.

**0. Secret Generation and Key dissemination Phase**. The dealer performs the following to share a secret $s \in \{0, 1\}^\nu$:

1. Choose $r^* \in \mathbb{N}$ according to a geometric distribution with parameter $\beta$;
2. Generate public and secret key pairs $(pk_1, sk_1), (pk_2, sk_2), \ldots, (pk_n, sk_n) \leftarrow V_G(1^\kappa)$;
3. Generate public and secret key pairs $(pk_1', sk_1'), (pk_2', sk_2'), \ldots, (pk_n', sk_n') \leftarrow V_G(1^\kappa)$;
4. Choose random $(n - 1)$-degree polynomials $G \in \mathbb{F}_{2^\nu}[x]$ and $H \in \mathbb{F}_{2^\nu}[x]$ such that $G(0) = s$ and $H(0) = 0$;
5. Compute $\{g_i^* := G(i) \oplus V_E(sk_i, r^*)\}_{i \in [n]}$ and $\{h_i^* := H(i) \oplus V_E(sk_i, r^* + 1)\}_{i \in [n]}$;
6. Construct `transmission_ordering_b` by listing down one arbitrarily chosen path for each pair of players starting with a path from $p_1$ to $p_2$, followed by a path from $p_1$ to $p_3$, etc., up to $p_1$ to $p_n$. Afterwards, list down a path from $p_2$ to $p_1$, followed by a path from $p_2$ to $p_3$, etc. (The algorithm for `path_ordering` is not needed for

transmission_ordering_b.) In each path in transmission_ordering_b, the origin-node is marked with a special symbol start, and the end-node of each path is marked with a special symbol end;

7.  Define the tuple of public information as:

$$\Psi = (\{pk_i\}_{i\in[n]}, \{pk_i'\}_{i\in[n]}, \{g_i\}_{i\in[n]}, \{h_i\}_{i\in[n]},$$

$$\texttt{transmission\_ordering\_a}$$

$$\texttt{transmission\_ordering\_b});$$

8.  For $i \in [n]$, send $((sk_i, sk_i'), \Psi)$ to $p_i$.

**1. Reconstruction Phase**.

1.  Given transmission_ordering_b, for $i \in [n]$, if it is $p_i$'s turn to transmit as the origin-node for the first time (i.e., its node is marked with start for the first time), $p_i$ computes the following:

$$y_i^r = V_E(sk_i, r), z_i^r = V_E(sk_i', r)$$

$$\pi_i^r = V_P(sk_i, r), \psi_i^r = V_P(sk_i', r)$$

Afterwards, $p_i$ sends $(g_i^r, h_i^r)$ to $\{p_{i'}\}_{i'\in[n]\setminus i}$ as per transmission_ordering_b.

2.  For $i \in [n]$, if $p_i$ receives or does not receive a transmission from some other node in violation of transmission_ordering_b in terms of the current round, it outputs $s^{r-1}$ then aborts. Otherwise, it checks transmission_ordering_b to determine the source of the transmission which is $p_{\texttt{source}}$ for some source $\in [n]$. Afterwards, given $r$ and $\{r, y^r, \pi^r, z^r, \psi^r\}$ in the transmission, $p_i$ checks that both $V_V(pk_{\texttt{source}}, r, y^r, \pi^r)$ and $V_V(pk_{\texttt{source}}, r, z^r, \psi^r)$ are true. If any of these are false, $p_i$ aborts.

    Otherwise, if $p_i$'s node is not marked with end as per transmission_ordering_b, it sends the transmission to the respective receiver node as per transmission_ordering_b. However, if it is $p_i$'s turn to receive information (i.e., its node is marked with end), it sets source as the index of the origin-node of the transmission, i.e., the transmission originates from player $p_{\texttt{source}}$. Afterwards, it performs the following:

    (a)  Check if the information received is of the form $(y^r, z^r, \pi^r, \psi^r)$. If not true, output $s^{r-1}$ and abort.
    (b)  Check if $n$ tuples of the form $(y_{i'}^r, z_{i'}^r, \pi_{i'}^r, \psi_{i'}^r)$ for indices $i' \in [n]$ have so far been acquired. If true, let $I$ denote the player indices corresponding to such tuples. Compute $h_{i'}^r := h_{i'} \oplus z_{i'}^r$ for all $i' \in I$, and interpolate an $(n-1)$-degree polynomial $H^r$ using $\{h_{i'}^r\}_{i'\in I}$. If $H^r(0) = 0$, output $s^{r-1}$ immediately as the computed secret and abort.
    (c)  Otherwise, if $H^r(0) \neq 0$ in the above item, compute $s_i^r$ as follows: set $g_{i'}^r := g_{i'} \oplus y_{i'}^r$ for all $i' \in I$. Interpolate an $(n-1)$-degree polynomial $G^r$ through $\{g_r^{i'}\}_{i'\in I}$ and set $s_i^r := G^r(0)$.

3.  For $i, i' \in [n]$, if $p_i$: (a) did not receive any transmission from some other *origin-node* $p_{i'}$ ($i' \neq i$) according to transmission_ordering_b, it outputs $s^{r-1}$ then aborts.

Equilibrium properties of $\Pi_{2.1}$ are stated in Theorem 7, which says that $\Pi_{2.1}$ guarantees a computational Nash equilibrium. Proof for Theorem 7 is in the Appendix. The more interesting result, however, for $\Pi_{2.1}$ is in Corollary 1, which states that $\Pi_{2.1}$ can accommodate coalitions of a size larger than $k$, as long as these coalitions are 1-disconnected. An example instance for which Corollary 1 applies is shown in Figure 4.

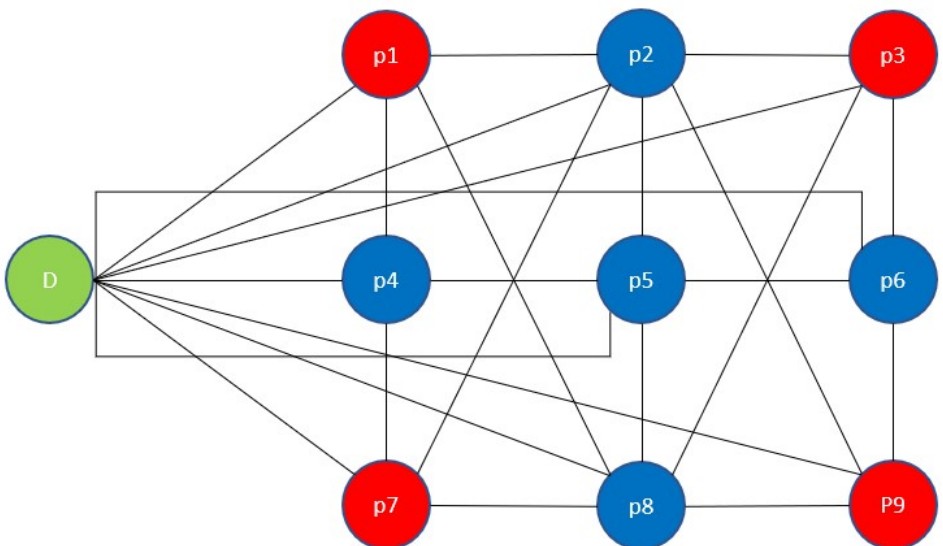

**Figure 4.** Example of a graph in the $A_{GN}$ game with a 4-member coalition (red-colored nodes). The coalition is 1-disconnected, since no member of the coalition is directly connected to every other member of the coalition. By Corollary 1, this set-up is allowed under $\Pi_{2.1}$ and results in a computational Nash equilibrium even if there is a coalition of size greater than $k = 3$.

**Theorem 7.** *Given $\kappa \in \mathbb{N}$, let $v := v(\kappa)$ denote the value of a polynomial in $\kappa$. Let $\mathcal{G}$ be an $A_{GN}$ game with $n + 1$ participants associated with a $G(V, E)$ and domain $\mathcal{S} := \{0, 1\}^v$ such that the $E$ has edges from the dealer node to each of the player nodes. Let $\Phi$ denote the set of conditions $\Phi := \{1\text{-disconnected}\}$. The protocol $\Pi_{2.1}(n, k)$ is a $\Phi$-resilient computational Nash equilibrium provided that $[(\beta \times U^+) + (1 - \beta) \times U_{\mathtt{rand}} - U] < 0$, where $\beta$ is the parameter of a geometric distribution.*

**Corollary 1.** *Given $\kappa \in \mathbb{N}$, let $v := v(\kappa)$ denote the value of a polynomial in $\kappa$. Let $\mathcal{G}$ be an $A_{GN}$ game with $n + 1$ participants associated with a $G(V, E)$ and domain $\mathcal{S} := \{0, 1\}^v$ such that the $E$ has edges from the dealer node to each of the player nodes. Let $\Phi$ denote the set of conditions $\Phi := \{1\text{-disconnected}\}$. If $\Pi_{2.1}(n, k)$ is a $\Phi$-resilient computational Nash equilibrium, then $\Pi_{2.1}(n, k)$ is resilient against some coalitions of size larger than $k$.*

**Proof.** By the definition of a $\Phi$-resilient computational Nash equilibrium, if a protocol is $\Phi$-resilient, then it is secure against any coalition that satisfies the requirements of $\Phi$ regardless of their size. The corollary thus follows. □

### 5. Possible Directions for Future Work

Some possible directions for future work are as follows:

1.  Our paper showed the existence of protocols that guarantee equilibria in an $A_{GN}$ secret sharing game given very specific graph-theoretical properties. Natural extensions over these results would be to investigate if there are certain protocols that induce equilibria over more general graph-theoretical properties. On the other hand, one could also investigate if there are other graph-theoretical properties that allow either computationally strict Nash equilibria or $\Phi$-equilibria. For instance, aside from 1-disconnected, could other properties also be included in $\Phi$ in order to tolerate larger coalitions?
2.  Our protocols could be further simplified or optimized in terms of their round and communication complexity. For instance, there may be more computationally efficient secret sharing schemes aside from Shamir Secret Sharing that allow the protocol to induce the same types of equilibria. It is also possible to further improve the

complexity of the $(n, k)$ Shamir Secret Sharing used in securely distributing the secret along $k$-disjoint paths.

## 6. Conclusions

In this paper, we address the problem of designing secret sharing protocols over a general network with rational players, such that these protocols induce the desirable equilibrium outcome whereby it is advantageous for each player to stick to the protocol and let all players correctly reconstruct the secret in the process. We present three protocols, whereby our first protocol uses the pseudorandom cryptographic primitive along with a standard Shamir Secret Sharing scheme in the presence of an online dealer. The second protocol uses a more sophisticated cryptographic primitive, namely, VRFs in order to reduce communication complexity from the first protocol and requires only a semi-online dealer. Our third protocol is similar to the second protocol, but requires a special type of general network whereby the dealer is directly connected to each player.

To formally express the game-theoretic behaviour of our protocols in the context of computational complexity, we utilize existing notions of computational Nash equilibrium and also present novel notions of computational equilibria—namely, $(n-1)$-key leakage-tolerant equilibrium and $\Phi$-resilient computational Nash equilibrium. Our results and proofs show that our first and second protocols, $\Pi_1$ and $\Pi_2$, respectively, both induce an $(n, k)$ strict computational Nash equilibrium, a $(n-1)$-key leakage-tolerant equilibrium, and a $(k-1)$-resilient computational Nash equilibrium relative to certain values of the geometric distribution parameter $\beta$ and the values of the players' utilities $U^+, U, U^-$. The communication complexity of $\Pi_2$ per round is less than $\Pi_1$, but $\Pi_2$ has much higher round complexity. Finally, for the third protocol, $\Pi_{2.1}$, we show that it induces a $\Phi$-resilient computational Nash equilibrium, where $\Phi$ contains the graphical property of being 1-disconnected. This implies that under $\Pi_{2.1}$, certain coalitions of size larger than $k$ can be tolerated by the protocol as long as the location of the members of the coalition in the network's graph satisfy the 1-disconnected property.

**Author Contributions:** Conceptualization, A.L.; Formal analysis, A.L. and H.A.; Supervision, H.A. All authors have read and agreed to the published version of the manuscript.

**Funding:** This study was funded by the Engineering Research and Development for Technology (ERDT) program of the Department of Science and Technology (DOST), Philippines.

**Data Availability Statement:** Not applicable.

**Conflicts of Interest:** The authors declare no conflict of interest.

## Appendix A. Coalition Equilibrium Notions

**Definition A1.** *Let $\mathtt{script}_d$ be as in Definition 15. Given a coalition $\mathcal{C}$, we define $\mathtt{view}_{-\mathcal{C}}^\Pi$ as follows. Let $\mathtt{script}_\mathcal{C}$ denote the transmissions of members of $\mathcal{C}$ to adjacent player nodes that are not members of $\mathcal{C}$ over the course of the game. $\mathtt{script}_\mathcal{C}$ does not include any transmissions of members of $\mathcal{C}$, once a member of $\mathcal{C}$ outputs a guess of the secret $s$. Let $\mathtt{script}_{-\mathcal{C}}$ denote the set of transmissions of $p_{i'}$ for $i' \in [n]$ with $i' \notin \mathcal{C}$ to its adjacent nodes over the course of the game. Let all participants follow the strategies prescribed by $\Pi$. $\mathtt{view}_{-i}^\Pi$ is defined as information which includes $\mathtt{script}_d$, $\mathtt{script}_\mathcal{C}$, and $\mathtt{script}_{-\mathcal{C}}$, plus all randomness involved in the computations of $p_{i'}$ for $i' \in [n]$ with $i' \notin \mathcal{C}$ across all rounds.*

**Definition A2.** *Let $\rho_\mathcal{C}$ be a set of strategies of members of $\mathcal{C}$ such that $\rho_\mathcal{C} \neq \sigma_\mathcal{C}$. Let all participants (except those in $\mathcal{C}$) follow the strategies prescribed by $\Pi$, while members of $\mathcal{C}$ follow $\rho_\mathcal{C}$. Given this set of strategies, let $\mathtt{script}_d$, $\mathtt{script}_\mathcal{C}$, $\mathtt{script}_{-\mathcal{C}}$ be as in Definition A1. Let $T$ be some polynomial-time algorithm that knows the entire view of members of $\mathcal{C}$ as they follow $\rho_\mathcal{C}$, and which outputs a truncation $\mathtt{script}'_\mathcal{C}$ of $\mathtt{script}_\mathcal{C}$. We define $\mathtt{view}_{-\mathcal{C}}^{T, \rho_\mathcal{C}, \Pi}$ as information which includes $\mathtt{script}_d$, $\mathtt{script}'_\mathcal{C}$, $\mathtt{script}_{-\mathcal{C}}$, plus all randomness involved in the computations of $p_{i'}$ for $i' \in [n]$*

with $i' \notin \mathcal{C}$ across all rounds. Similarly, define $\mathtt{view}_{-\mathcal{C}}^{\rho_{\mathcal{C}},\Pi}$ as the same information contained in $\mathtt{view}_{-\mathcal{C}}^{T,\rho_{\mathcal{C}},\Pi}$ but which excludes reference to T.

**Definition A3.** *Let $f$ denote a negligible function over $\kappa$. For a coalition $\mathcal{C}$, a strategy $\rho_{\mathcal{C}}$ is equivalent with respect to $\Pi$, or $\rho_{\mathcal{C}} \sim \Pi$ if there exists a polynomial-time algorithm $T$ such that for all polynomial-time distinguishers $D$, we have:*

$$\left| \Pr\left[D(1^{\kappa}, \mathtt{view}_{-\mathcal{C}}^{T,\rho_{\mathcal{C}},\Pi}) = 1\right] - \Pr\left[D(1^{\kappa}, \mathtt{view}_{-\mathcal{C}}^{\Pi}) = 1\right] \right| \leq f(\kappa).$$

**Definition A4.** $\Pi$ *induces a $(k-1)$-resilient computational strict Nash equilibrium if: (1) it induces a $(k-1)$-resilient computational Nash equilibrium and (2) for any coalition $\mathcal{C} \subseteq \boldsymbol{P}$ such that $|\mathcal{C}| < k$, and any polynomial-time strategy $\sigma_{\mathcal{C}}'$ such that $\sigma_{\mathcal{C}}' \nsim \Pi$, there is a $c > 0$ such that $u_{\mathcal{C}}(\sigma) \geq u_{\mathcal{C}}(\sigma_{\mathcal{C}}', \sigma_{-\mathcal{C}}) + 1/\kappa^c$ for infinitely many values of $\kappa$.*

## Appendix B. Security of the Shamir Secret Sharing Scheme

The security notion of an $(n,k)$ secret sharing scheme is stated formally in [2], whereby an $(n,k)$-secret sharing scheme $(S_G, S_R)$ over $\mathcal{S}$ is secure if, for every possible secret $s, s' \in \mathcal{S}$ and every subset $\{s_1, s_2, \ldots, s_{k-1}\} \subseteq \mathcal{S}^{k-1}$ of size $k-1$, the distribution of $S_G(s)$ is identical to the distribution of $S_G(s')$ such that given any set of shares of size $k-1$, one cannot tell if the secret is $s$ or $s'$ for all $s, s' \in \mathcal{S}$. For a specific instance of a secure $(n,k)$ secret sharing scheme, below is a non-rational $(n,k)$-Shamir Secret Sharing scheme based on Lagrange Interpolation from [1].

[Share Generation.] $S_G(s)$: on input secret $s$, let $\mathbb{Z}_p$ be a field for some prime $p$. Perform the following given $n$ and $k$:

1.  Sample $k-1$ random numbers $(r_i)_{i \in [k-1]}$, where $r_i \leftarrow \mathcal{Z}_p$;
2.  Define the polynomial $f(x) \in \mathbb{Z}_p[x]$ as $f(x) := r_{k-1}x^{k-1} + r_{k-2}x^{k-2} + \cdots + r_1 x + c$;
3.  Choose arbitrary $\{x_1, x_2, \ldots, x_n\} \in \mathbb{Z}_p$;
4.  Evaluate $y_i = f(x_i)$ and set $s_i := (x_i, y_i)$ for $i \in [n]$;
5.  Return $\boldsymbol{s} := (s_1, s_2, \ldots, s_k)$.

[Secret Reconstruction.] $S_R(\boldsymbol{s}')$: on input $\boldsymbol{s}'$ of size at least $k$, perform the following:

1.  Using any set of $k$ shares from $\boldsymbol{s}'$, i.e., $\{s_i := (x_i', y_i')\}_{i \in [j]}$, re-construct $f(x)$ using Lagrange interpolation by constructing $k$ polynomials of the form $L_i(x)$ below:

$$L_i(x) := \prod_{\substack{j=1 \\ j \neq i}}^{k} \frac{x - x_j'}{x_i' - x_j'} \in \mathbb{Z}_p[x] \text{ for } i \in [k];$$

2.  Form another polynomial $g(x) := L_1(x) \cdot y_1 + L_2(x) \cdot y_2 + \cdots + L_k(x) \cdot y_k \in \mathbb{Z}_p[x]$ and return $s' := g(0)$.

**Lemma A1.** *From [2], the scheme above is a secure $(n,k)$-secret sharing scheme.*

The following Lemma is a standard result using Lagrange Interpolation.

**Lemma A2.** *Let $\{(x_1, y_1), (x_2, y_2), \ldots, (x_{n-1}, y_{n-1})\}$, where $x_i, y_i \in \mathbb{Z}_p$ for some prime $p > 0$ be coordinates of an $n-1$-degree polynomial that is not known. Given the tuple*

$$\{(x_1, y_1), (x_2, y_2), \ldots, (x_{n-1}, y_{n-1})\}$$

*and $x_n$, the distribution of possible values of $y_n$ is uniform.*

## Appendix C. Pseudorandom Functions

**Definition A5.** *A pseudorandom function $\Lambda : \mathcal{SK} \times \mathcal{S} \to \mathcal{Y}$, where $\mathcal{SK}$ is a key space and $\mathcal{S}$ is an input data block, is a deterministic algorithm that behaves like a truly random function [2]. A pseudorandom function has the following properties:*

1. *Pseudorandomness: The pseudorandom security of a pseudorandom function $\Lambda$ is defined in terms of an Attack Game between a challenger and an adversary. Given $\kappa$, at the start of the game, the challenger randomly draws $b \in \{0, 1\}$ and selects a random function $f$ from $\mathcal{S}$ to $\mathcal{Y}$. The adversary submits a sequence of queries to the challenger, where each query consists of an element $s \in \mathcal{S}$. If $b = 0$, the challenger draws $sk \leftarrow \mathcal{SK}$ and submits $\Lambda(sk, s)$ to the adversary. If $b = 1$, the challenger submits $f(s)$ to the adversary. The game ends once the adversary submits a guess $b' \in \{0, 1\}$ who wins if $b' = b$. The advantage of the adversary in this game is defined as $|\Pr[b' = b] - 1/2|$. The pseudorandom function $P$ is a secure PRF if the advantage of any polynomial time adversary in this game is negligible in $\kappa$. It follows that the distribution of the output of $\Lambda$ is indistinguishable from uniform.*

2. *Secure key recovery: Let $\Lambda : \mathcal{SK} \times \mathcal{S} \to \mathcal{Y}$ be a pseudorandom function. Given $s \in \mathcal{S}$ and $y \in \mathcal{Y}$, it is computationally difficult to compute $sk \in \mathcal{SK}$ such that $\Lambda(sk, s) = y$.*

We note that while secure key recovery is not normally included among the properties of a pseudorandom function in the literature [2], given that pseudorandomness is a stronger property than secure key recovery, we explicitly include it here for reference in the proofs.

## Appendix D. Verifiable Random Functions

**Definition A6.** *A verifiable random function (VRF) scheme with range $\mathcal{R} = \{\mathcal{R}\}_\kappa$ is a tuple of probabilistic polynomial-time algorithms $(V_G, V_E, V_P, V_V)$, where $V_G$ is a key generation algorithm, $V_E$ is an evaluation algorithm, $V_P$ is a proof generation algorithm, and $V_V$ is a proof verification algorithm. The following properties are required of a VRF following [8,24]:*

1. *Correctness: given $\kappa$, let $(pk, sk) \leftarrow V_G(1^\kappa)$. Let $y \leftarrow V_E(sk, x)$ and $\pi \leftarrow V_P(sk, x)$ for some $\kappa$-bit input $x$. We have $V_V(pk, x, V_E(sk, x), V_P(sk, x)) = 1$ with probability 1.*

2. *Verifiability: given $\kappa$, for all possible $(pk, sk) \leftarrow V_G(1^\kappa)$, there does not exist a tuple $(x, y, y', \pi, \pi')$ with $y \neq y'$ such that $V_V(pk, x, y, \pi) = 1 = V_V(pk, x, y', \pi')$.*

3. *Uniqueness of proofs: given $\kappa$, for all possible $(pk, sk) \leftarrow V_G(1^\kappa)$, there does not exist a tuple $(x, y, \pi, \pi')$ with $\pi \neq \pi'$ such that $V_V(pk, x, y, \pi) = 1 = V_V(pk, x, y, \pi')$.*

4. *Pseudorandomness: the security notion for pseudorandomness of a VRF is defined in terms of an Attack Game between a challenger and an adversary. Given $\kappa$, at the start of the game, the challenger samples $b \in \{0, 1\}$, and $(pk, sk) \leftarrow V_G(1^\kappa)$ then gives $pk$ to the adversary. The adversary adaptively sends a finite number of queries $x_i \in \mathcal{R}_\kappa$ to the challenger, for which the challenger returns $(y_i, \pi_i) = (V_E(sk, x_i), V_P(sk, x_i))$. At some point, the adversary performs a challenge query, whereby it sends the challenge query input $x^*$ to the challenger (subject to the restriction that $x^*$ is not in any previous query). Once the challenger receives $x^*$, if $b = 0$, the challenger returns the challenge ciphertext $y^* = V_E(sk, x^*)$ to the adversary. However, if $b = 1$, the challenger returns a uniformly sampled $y^* \leftarrow \mathcal{R}_\kappa$. After the challenge query, the adversary may proceed to query the challenger again for a finite number of times (subject to the restriction that no query is equal to $x^*$). The game ends once the adversary outputs a guess $b' \in \{0, 1\}$. The adversary wins if $b = b'$. Under this Attack Game, a VRF is pseudorandom if, for all polynomial-time adversaries, the advantage $|1/2 - \Pr[b = b']|$ is negligible in $\kappa$.*

## Appendix E. Protocol by Fuchsbauer et al.

The following protocol by [8] provides an exactly t-out-of-n secret sharing. Let $(V_G, V_E, V_P, V_V)$ correspond to polynomial-time algorithms that give a secure Verifiable Random Function Scheme. To share a secret $s \in \{0, 1\}^l$ to $n$ players $p_1, p_2, \ldots p_n$, [8]'s protocol has a sharing phase followed by a reconstruction phase, as follows:

**1. Secret Generation and Key dissemination Phase**.

1. Choose $r^* \in \mathbb{N}$ according to a geometric distribution with parameter $\beta$;
2. Generate keys $(pk_1, sk_1), \ldots, (pk_n, sk_n) \leftarrow V_G(1^\kappa)$ and $(pk'_1, sk'_1), \ldots, (pk'_n, sk'_n) \leftarrow V_G(1^\kappa)$;
3. Choose $(t-1)$ random polynomials $G$ and $H$ such that $G(0) = 0$ and $H(0) = 0$;
4. Send $(sk_i, sk'_i)$ to $p_i$;
5. Send to all parties the following:

   (a) $\{pk_j, pk'_j\}_{1 \leq j \leq n}$;

   (b) $\{g_j := G(j) \oplus V_E(sk_j, r^*)\}_{1 \leq j \leq n}$;

   (c) $\{h_j := H(j) \oplus V_E(sk_j, r^*+1)\}_{1 \leq j \leq n}$.

**2. Reconstruction Phase**.

1. Each player $p_i$ chooses $s_i^0$ uniformly, and in each iteration, each $p_i$ performs the following:

   (a) Send the following to all players:

   - $y_i^r = V_E(ski, r)$ and $z_i^r = V_E(sk'_i, r)$;
   - $V_P(sk_i, r)$ and $V_P(sk'_i, r)$.

   (b) If $p_i$ receives nothing or an incorrect proof from some other player $p_j$, $p_i$ terminates and outputs $s_i^{r-1}$ and aborts. Otherwise:

   - $p_i$ sets $h_j^r := h_j \oplus z_j^r$ for all other players, and interpolates a $(t-1)$ polynomial $H^r$ through these points. If $H^r(0) = 0$, $p_i$ outputs $s_i^{r-1}$ and aborts.
   - Otherwise, $p_i$ sets $g_j^r := g_j \oplus y_j^r$ and interpolates a $(t-1)$ polynomial $G^r$ through these points. It sets $s_i^r := G(0)$.

*Appendix E.1. Issues under a General Network with Combining SMT and [8]'s Protocol*

Suppose that in some $k$-path disjoint graph, the dealer $d$ is not directly connected to some player $p_l$, but there is a path from $d$ to $p_l$ passing through another player $p_i$. Suppose that the prior protocol by [8] is implemented together with SMT in a general network, whereby, under this protocol's strategy, the dealer $d$ sends $(sk_l, sk'_l)$ to $p_l$ securely using SMT. Following SMT, $(sk_l, sk'_l)$ is broken down to several sub-shares and is sent along $k$-disjoint paths to $p_l$, for which $p_l$ securely reconstructs $(sk_l, sk'_l)$. However, under the protocol of [8], the dealer $d$ also has to send the tuple of public information $\Psi :=$ $(\{pk_j, pk'_j\}_{j \in [n]}, \{g_j\}_{j \in [n]}, \{h_j\}_{j \in [n]}$ to player $p_l$ in phase 1. However, given that $\Psi$ is public, SMT is no longer applied to $\Psi$ under this protocol. Instead, $d$ sends $\Psi$ to $p_i$, under the assumption that $p_i$ merely has to send $\Psi$ to $p_l$ without any modifications. In addition, The strategy of this protocol for $p_i$, however, is dominated by another strategy. Namely, in this dominating strategy, $p_i$ mauls $\{h_j\}_{j \in [n]}$. As a result of this action by $p_i$, $p_l$ can no longer correctly compute $H(0) = r^* + 1$ in the secret reconstruction phase, and $p_l$ cannot determine if the definitive iteration has been reached. However, $p_i$ continues to receive the correct information from $p_l$ during the secret reconstruction phase given that the tuple $(y_l^r, z_l^r, V_P(sk_l, r), V_P(sk'_l, r))$ provided by $p_l$ is independent of $\{h_j\}_{j \in [n]}$. This implies that $p_i$ can still correctly compute $H(0) = r^* + 1$ and determine if the definitive iteration has been reached, while $p_l$ can no longer do so. Given the utility assumptions in Section 2.3, $p_i$ has higher utility under this strategy since it means that one less player gets to know about the secret. It could be seen that if $p_l$ were able to determine that the $\Psi$ it received from $p_i$ is mauled, then $p_l$ could avoid this situation by aborting. This is the idea behind the duplication checks in the proposed protocols of this paper.

## Appendix F. Technical Results for Protocol $\Pi_1$

**Lemma A3.** *Given an extensive game $\mathcal{G}$ with imperfect information, let $\mathbf{e}_0$ and $\mathbf{e}_1$ be two mutually exclusive events in the game, such that either $\mathbf{e}_0$ or $\mathbf{e}_1$ occurs with probability 1. For each player $p_i \in N$ in the game, let $\{\mathbf{e}_0, \mathbf{e}_1\} \notin \phi_i(I)$ for $I \in \mathcal{I}_i$, i.e., no player knows if either $\mathbf{e}_0$ or $\mathbf{e}_1$ occurred. Denote by $\mathrm{Pr}_0$ the game's probabilities conditional on $\mathbf{e}_0$ having occurred, and by $\mathrm{Pr}_1$, the game's*

*probabilities conditional on* $\mathbf{e}_1$ *having occurred. If* $\Pr_0[\phi_i(I)]$ *is indistinguishable from* $\Pr_1[\phi_i(I)]$ *for all* $I \in \mathcal{I}_i$ *and for each* $p_i \in N$*, then the distribution of histories of the game under* $\mathbf{e}_0$ *is indistinguishable from the distribution of histories of the game under* $\mathbf{e}_1$*.*

**Proof.** If $\Pr_0[\phi_i(I)]$ is indistinguishable from $\Pr_1[\phi_i(I)]$ for all $I \in \mathcal{I}_i$ and for each $p_i \in N$, the distribution of actions $\Pr_0[A_i(\phi_i(I))]$ is also indistinguishable from $\Pr_1[A_i(\phi_i(I))]$ for all $I \in \mathcal{I}_i$ and for each $p_i \in N$ given that $A_i$ is a function of $\phi_i$. The statement thus follows. □

**Definition A7.** *Let* $\mathcal{G}$ *be an* $A_{GN}$ *game with N as the set of n players. Given the protocol* $\Pi_1(n,k)$ *over* $\mathcal{G}$ *for some* $k \leq n$*, the following events are defined (relative to player* $p_i \in N$*):*

1. `short` *occurs if some player aborts before phase 1 ends.*
2. `abort` *occurs if phase 2 is reached and if some player aborts before iteration* $r^* + 1$*.*
3. `early` *is the event that* $r < r^*$ *and an* `abort` *occurs.*
4. `exact` *is the event that* $r = r^*$ *and an* `abort` *occurs.*
5. `late` *is the complement of* `abort`*, i.e., no player aborts before iteration* $r^* + 1$*.*
6. `maul` *is the event that* $p_i$ *modifies any share* $s_{l,j}$ *for some* $l \in [n], j \in [k]$ *received during phase 1.*
7. `true(i)` *is the event that* $p_i$ *outputs the correct secret s.*
8. `true(-i)` *is the event that all other players* $p_j$ $(i \neq j)$ *outputs the correct secret s*

From the above definition, we have `abort = early ∪ exact`

**Definition A8.** *From the description of* $\Pi_1(n,k)$*, each transmission from a player* $p_i$ *to some player* $p_l$ $(i \neq l)$ *contains a path encoding corresponding to* $\{\text{path}_{l,j}\}_{j \in [k]}$*. Another player* $p_m$ $(m \neq i \wedge m \neq l)$ *does not follow the path encoding contained in a received transmission, if for some* $\text{path} \in \{\text{path}_{l,j}\}_{j \in [k]}$*,* $p_m$ *is in the node sequence corresponding to* `path` *and either: (a)* $p_m$ *refuses to send the transmission to the next node listed in* `path`*, or (b)* $p_m$ *modifies* `path` *to another value and sends the transmission.*

**Lemma A4.** *Given* $\Pi_1(n,k)$*, denote by* $\sigma$ *the corresponding set of strategies prescribed by* $\Pi_1$*. Let* $p_i$ *follow some polynomial-time strategy* $\sigma_i'$ *and let all other players follow* $\sigma_{-i}$*. The event* `short` *occurs due to* $p_i$ *with non-negligible probability if: (1)* $p_i$ *follows* $\sigma_i'$ *such that it aborts during phase 1; or (2) if for some transmission meant for another player* $p_j$ $(i \neq j)$*,* $p_i$ *does not follow the path encoding according to Definition A8; or if (3)* $p_i$ *modifies a transmission meant for some other player* $p_j$*, where* $p_i$ *sends* $\{(\hat{g}_i, \hat{h}_i)\}_{i \in [n]}$ *such that* $\{(\hat{g}_i, \hat{h}_i)\}_{i \in [n]} \neq \{(g_i^*, h_i^*)\}_{i \in [n]}$*. If* `short` *occurs due to* $p_i$*, we have* $\sigma_i' \not\sim \Pi_1$*.*

**Proof.** In (1), if $p_i$ itself aborts in phase 1, `short` occurs by definition. For (2), let $p_i$ receive a transmission from the dealer whose end-receiver is another player $p_j$, $(i \neq j)$. From Definition A8, $p_i$ does not follow the path encoding if: (a) $p_i$ refuses to send the transmission to the next node in the original path encoding, or (b) $p_i$ modifies the original path encoding and sends the transmission. For (b), two cases are possible: (b.1): $p_j$ does not receive the modified transmission due to the change in the path encoding; (b.2): $p_j$ does receive the modified transmission. For (a) at some point, $p_j$ discovers that it has less than $k$ tuples of information from the dealer after `max_l` rounds and aborts as a result—making `short` occur. For (b.1), if $p_i$ modifies the path such that $p_j$ will not receive the transmission, $p_j$ will discover that it has less than $k$ tuples of information from $d$ and aborts—making `short` occur. For (b.2), if $p_i$ modifies the path encoding, $p_j$ will detect this change given that it has $k-1$ other copies of the path encoding for comparison—making `short` occur. Finally, for (3), if $p_i$ modifies $\{(g_i^*, h_i^*)\}_{i \in [n]}$ in a transmission meant for $p_j$, the player $p_j$ will notice this given that it has $k-1$ other copies of $\{(g_i^*, h_i^*)\}_{i \in [n]}$. Denote by $\text{view}^{\sigma_i', \Pi_1}$ the set of information following Definition 16. For the last statement of the Lemma, we have $\sigma_i' \not\sim \Pi$ if $\text{view}^{\sigma_i', \Pi} \neq \text{view}^{\Pi}$. If $p_i$ does not follow the path encoding or modifies the

transmission, then at least one other player $p_j$ aborts before iteration $r^* + 1$, which implies that $\texttt{view}^{\sigma'_i, \Pi_1} \neq \texttt{view}^{\Pi_1}$ and, therefore, $\sigma'_i \not\sim \Pi_1$.  □

**Lemma A5.** *Given $\Pi_1(n, k)$, denote by $\sigma$ the corresponding set of strategies prescribed by $\Pi_1$. Let $p_i$ follow some polynomial-time strategy $\sigma'_i$ and let all other players follow $\sigma_{-i}$. The event* $\texttt{abort}$ *occurs due to $p_i$ with non-negligible probability if for some iteration $r \leq r^*$, any of the following occurs: (1) $p_i$ aborts before iteration $r^* + 1$; (2) some path encoding in a transmission from either the dealer or some other player is not followed in phase 2 by $p_i$; (3) in some transmission, $p_i$ sends $h^\circ$ such that $h^\circ \neq h'$ (where $h'$ is from the dealer); (4) in some transmission from $p_l$ to $p_j$ that passes through $p_i$, $p_i$ sends $(\hat{g}_l, \hat{h}_l)$ such that $(\hat{g}_l, \hat{h}_l) \neq (g^r_l, h^r_l)$; (5) with $p_i$ as the origin-node, $p_i$ sends $(\hat{g}_i, \hat{h}_i)$ such that $(\hat{g}_i, \hat{h}_i) \neq (g^r_i, h^r_i)$; or (6)* $\texttt{maul}$ *occurs in phase 1 due to $p_i$. If* $\texttt{abort}$ *occurs due to $p_i$, we have $\sigma'_i \not\sim \Pi_1$.*

**Proof.** For (1), if $p_i$ itself aborts before iteration $r^* + 1$, then $\texttt{abort}$ occurs by definition. For (2), if $p_i$ does not follow some path encoding in a transmission from either the dealer or some other player (either by refusing to send or by modifying the path encoding), the same reasoning and cases as in the proof for Lemma A4 applies (changing the origin-node of the path encoding from the $d$ to some other player's node as the case may be). Therefore, $\texttt{abort}$ occurs in this case. For (3) if $p_i$ sends $h^\circ$ such that $h^\circ \neq h'$ (where $h'$ is from the dealer) to some other player $p_j$ ($i \neq j$), this change will be detected by $p_j$ given that it has $k - 1$ other copies of $h'$. In this case, $p_j$ aborts, and $\texttt{abort}$ occurs. The same reasoning applies for (4), whereby if $p_i$ sends $(\hat{g}_l, \hat{h}_l)$ such that $(\hat{g}_l, \hat{h}_l) \neq (g^r_l, h^r_l)$ to $p_j$ for some $j \in [n] \setminus \{i, l\}$, the player $p_j$ will detect this given that it has $k - 1$ other copies of $(g^r_l, h^r_l)$. In this case, $p_j$ aborts and $\texttt{abort}$ occurs.

For (5), if $p_i$ itself sends $(\hat{g}_i, \hat{h}_i)$ such that $(\hat{g}_i, \hat{h}_i) \neq (g^r_i, h^r_i)$, the other players would not detect this using the $k - 1$ other copies of $(\hat{g}_i, \hat{h}_i)$ since they are all equal. However, the players will detect the change given that $\oplus_{i \in [n]} h^r_i \neq h'$ with non-negligible probability, and $\texttt{abort}$ occurs. This also implies (6) since, if $p_i$ modified some share $s_{l,j}$ meant for $p_l$ ($i \neq l$) (i.e., $\texttt{maul}$ occurs due to $p_i$) along the $j$th path to $p_l$, the player $p_l$ computes a secret key $sk'_l$ such that $sk_l \neq sk'_l$. It follows that all computations of $p_l$ involving $\Lambda$ are affected by this change from $sk_l$ to $sk'_l$. In particular, $p_l$ computes $\hat{h}^r_l = \Lambda(sk'_l, r)$ such that $\hat{h}^r_l \neq h^r_l$ with non-negligible probability. It follows that $\oplus_{j \in [n] \setminus l} h^r_j \oplus \hat{h}^r_l \neq h'$ with non-negligible probability, and $\texttt{abort}$ occurs. The same applies if $p_i$ for some reason modified $s_{i,j}$ for some $j \in [k]$ (i.e., a share that is meant for $p_i$ as end-receiver).

Denote by $\texttt{view}^{\sigma'_i, \Pi_1}$ the set of information following Definition 16. For the last statement of the Lemma, we have $\sigma'_i \not\sim \Pi$ if $\texttt{view}^{\sigma'_i, \Pi_1} \neq \texttt{view}^{\Pi_1}$. If $p_i$ performs any of (1)–(6) under $\sigma'_i$, then at least one other player $p_j$ notices this and $\texttt{abort}$ occurs as shown above, which implies that $\texttt{view}^{\sigma'_i, \Pi_1} \neq \texttt{view}^{\Pi_1}$ and therefore $\sigma'_i \not\sim \Pi_1$.  □

**Definition A9.** *Let $\phi_i$ denote the relevant information from $p_i$'s point of view for achieving utilities $U^+$ or $U$ at any information set in either phase 1 or 2 of $\Pi_1(n, k)$. It follows that we have $\phi_i := \{sk_i, \{g^*_i\}_{i \in [n]}, \{h^*_i\}_{i \in [n]}\}$ in phase 1, and for iteration $r$ in phase 2, we have $\phi_i := \{sk_i, \{g^*_i\}_{i \in [n]}, \{h^*_i\}_{i \in [n]}, \{\{g^\tau_i\}_{i \in [n]}, \{h^\tau_i\}_{i \in [n]}\}_{\tau \leq r}\}$.*

**Lemma A6.** *Under $\Pi_1(n, k)$, suppose that $p_i$ deviated and acquired $n - 1$ secret keys. Given $\phi_i$ from any information set $I$ in either phase 1 or any iteration $r \leq r^*$ in phase 2, the distributions of $\{h^*_i\}_{i \in [n]}$, $\{g^*_i\}_{i \in [n]}$, and the polynomials $H$ and $G$ are all indistinguishable from random. In addition, the probability of guessing $r^*$ is $\beta$.*

**Proof.** Without loss of generality, let $p_i$ acquire $n - 1$ secret keys except the last one, $sk_n$, which is owned by $p_n$. We first show that the above Lemma does not hold if $p_i$ has $n$ pairs of secret keys at its disposal. Suppose that $p_i$ knows $sk_n$ as well. A strategy for $p_i$ to compute $r^*$ is to evaluate $h^r_i = h^*_i \oplus \Lambda(sk_i, r)$ for $i \in [n]$ and for $r < 2^\kappa - 1$ in one round

(internally). For $r < 2^\kappa - 1$, $p_i$ checks if the interpolated polynomial $H^r$ from $\{h_i^r\}_{i \in [n]}$ satisfies $H^r(0) = 0$. If $H^r(0) = 0$, then $p_i$ sets $r - 1 = r^*$. Thus, $r^*$ is learned with probability greater than $\beta$ since sampling $r^* \geq 2^\kappa - 1$ is negligible.

So suppose that $p_i$ does not know $sk_n$. The other keys $sk_j$ for $j \in [n] - 1$ do not provide information on $sk_n$ since they are sampled independently. This leaves us with $\phi_i$. Since $\phi_i$ from phase 1 is a subset of $\phi_i$ from any iteration $r \leq r^*$ in phase 2, we need only consider $\phi_i$ from iteration $r \leq r^*$ in phase 2. First, we note that for any $\tau \leq r$, the set $\{\{g_i^\tau\}_{i \in [n]}, \{h_i^\tau\}_{i \in [n]}\}_{\tau \leq r}\} \in \phi_i$ does not provide information on $sk_n$ by the secure key recovery property of $\Lambda$ in Definition A5. Given this, we consider additional information in $\phi_i$. The $n - 1$ secret keys and $\phi_i$, give information on the values $g_c'(i) = g_i^* \oplus \Lambda(sk_i, r)$ and $h_c^p(i) = h_i^* \oplus \Lambda(sk_i, r)$ for $r > 0$ and $i \in [n-1]$. This leads to several coordinate tuples $[(1, g_c'(1)), (2, g_c'(2)), \ldots, (n-1, g_c'(n-1))]$ and $[(1, h_c^p(1)), (2, h_c^p(2)), \ldots, (n-1, h_c^p(n-1))]$ for $r > 0$. The first tuple in the prior statement can be combined with the coordinate $(0, s)$ to interpolate a candidate polynomial $G_c^r$ such that $G_c^r(0) = s$ and acquire information on $G_c^r(n)$. It follows that at iteration $r > 0$, the value $g_n^r$ received from $p_n$ equals $G_c^r(n) \oplus g_n^*$. However, information about $G_c^r(n)$ is not useful at any iteration $r \leq r^*$ given that $s$ is sampled randomly and is unknown for all iterations $r \leq r^*$, and the polynomial condition $G_c^r(0) = s$ cannot be performed. This leaves us with the second coordinate tuple $[(1, h_c^p(1)), (2, h_c^p(2)), \ldots, (n-1, h_c^p(n-1))]$. For each $r > 0$, this tuple can be combined with the known coordinate $(0, 0)$ to interpolate a candidate polynomial $H_c^r$. This results in a target coordinate $\hat{y} = H_c^r(n)$ and a target value $\hat{h}_n^r = \hat{y} \oplus h_n^*$. It follows that at iteration $r = r^* + 1$, the value of $h_n^r$ received from player $p_n$ is equal to $\hat{h}_n^r$. Using $(n, h_n^* \oplus h_n^r)$ and the known $n - 1$ other coordinates, information about the true polynomial $H$ is arrived at—followed by information about $G$ and $s$ (i.e., by following the reconstruction of $G$ and $H$ in $\Pi_1$). However, for iterations $r \leq r^* + 1$, given that $\Lambda$ is pseudorandom and $sk_n$ is unknown, the distribution of possible values of $h_n^r = \Lambda(sk_n, r)$ is indistinguishable from random, so that the distribution of coordinate $(n, h_n^* \oplus h_n^r)$ combined with other $n - 1$ coordinates at iteration $r$ do not provide much information about the distribution of $H$ (i.e., since $H^r(0) \neq 0$ with non-negligible probability). The same holds for the distribution of $G$ so that both $G$ and $H$ are unknown and their distributions are indistinguishable from random. It follows that for player $p_i$, given information $\phi_i$ from any information set $I$ in iteration $r \leq r^*$, it is computationally hard to determine if $h_n^r = \hat{h}_n^r$ for some future $r$ so that $r^*$ could only be guessed with probability $\beta$. Given that $H$ and $G$ are sampled randomly and are unknown, it follows that the distribution of $h_i^* = H(i) \oplus h_n^r$ and $g_i^* = G(i) \oplus g_n^r$ is also indistinguishable from random. The Lemma thus follows. $\square$

**Lemma A7.** *Suppose that no player can acquire other secret keys unless information related to it is shared by another player through a transmission. For any coalition $\mathcal{C} \subseteq N$ of size at most $k - 1$, given $\phi_i$ from any information set $I$ in either phase 1 or any iteration $r \leq r^*$ in phase 2 of $\Pi_1(n, k)$, the distributions of $\{h_i^*\}_{i \in [n]}$, $\{g_i^*\}_{i \in [n]}$ and the polynomials $H$ and $G$ are all indistinguishable from random. In addition, the probability of guessing $r^*$ is $\beta$.*

**Proof.** This is a corollary of Lemma A6. Given that in a coalition $\mathcal{C}$ of size $k - 1$, the members can share up to $k - 1$ secret keys, the results of Lemma A6 can be applied to each member of $\mathcal{C}$, which assumes a stronger condition of up to $n - 1$ secret keys. $\square$

**Lemma A8.** *Under $\Pi_1(n, k)$, suppose that $p_i$ deviated and acquired $n - 1$ secret keys. If* `maul` *occured in phase 1 due to $p_i$, the probability of* `true(i)` *and* `true(-i)` *is negligible at any phase.*

**Proof.** Without loss of generality, let $p_i$ acquire $n - 1$ secret keys except the last one $sk_n$, which is owned by $p_n$, where the $n - 1$ secret keys in $p_i$'s possession are correct and not modified due to `maul` on $p_i$'s part. The Lemma does not hold if $p_i$ has $n$ pairs of correct secret keys at its disposal using a similar strategy as in the proof of Lemma A6. So suppose that $p_i$ does not know the correct $sk_n$ but knows the correct keys $sk_i$ for $i \in [n-1]$. Let the event

maul modify $sk_j$ for $j \in [n]$ to $\hat{sk}_j$ such that $\hat{sk}_j \neq sk_j$. Information from phase 1 received by $p_i$ is independent of the value of the modified $\hat{sk}_j$ due to maul. Hence, the situation of $p_i$ in phase 1 is similar to its situation if maul did not occur. Using Lemma A6, we arrive at the statement of Lemma A6 for phase 1. It follows that without information on $H$ and $G$, $p_i$'s guess of $s$ (so that true(i) occurs) is as good as random. Since this holds for any player, the Lemma is proven for phase 1. For phase 2, by the pseudorandomness of $\Lambda$, it follows that with non-negligible probability, we have $\Lambda(\hat{sk}_j, r) \neq \Lambda(sk_j, r)$ for all $r > 0$. In particular, at iteration $r = 1$, we have $\hat{h}_j^1 = \Lambda(\hat{sk}_j, 1) \neq \Lambda(sk_j, 1) = h_j^1$ so that $[\oplus_{l \in [n] \setminus j} h_l^1 \oplus \hat{h}_j^1] \neq [\oplus_{i \in [n]} h_i^1] = h'$ with non-negligible probability. Thus, the check in $\Pi_1$ fails at iteration 1 of phase 2 with non-negligible probability, and all players are forced to guess $s$ from the uniform distribution. This proves the Lemma. $\square$

**Lemma A9.** *Suppose that no player can acquire other secret keys unless information related to it is shared by another player through a transmission. For any coalition $\mathcal{C} \subseteq N$ of size at most $k - 1$, suppose that maul occured in phase 1 due to some deviation of $p_i \in \mathcal{C}$ from $\Pi_1(n, k)$. The probabilities of true(i) and true(-i) are8 negligible at any phase.*

**Proof.** This is a corollary of Lemma A8. Given that in a coalition $\mathcal{C}$ of size $k - 1$, the members can share up to $k - 1$ secret keys, the results of Lemma A8 can be applied to each member of $\mathcal{C}$, which assumes a stronger condition of up to $n - 1$ secret keys. $\square$

**Lemma A10.** *Given $\Pi_1(n, k)$, let $p_i$ follow any polynomial-time strategy $\sigma_i'$, and let the rest of the players follow strategies $\sigma_{-i}$ prescribed by $\Pi_1$. We have the following, where $\mathcal{S}$ is the domain of the secret, and where $f$ is some negligible function in $\kappa$. This result holds even if $\sigma_i'$ led $p_i$ to acquire less than n secret keys.*

1. $\Pr[\text{true(i)}|\text{short}, \overline{\text{maul}}] = 1/|\mathcal{S}| + f(\kappa)$
2. $\Pr[\text{true(i)}|\text{early}, \overline{\text{maul}}] = 1/|\mathcal{S}| + f(\kappa)$
3. $\Pr[\text{true(i)}|\text{exact}, \overline{\text{maul}}] = 1/|\mathcal{S}| + f(\kappa)$
4. $\Pr[\overline{\text{true}(-i)}|\text{late}, \overline{\text{maul}}] = 0$
5. $\Pr[\text{true(i)}|\text{maul}] = 1/|\mathcal{S}| + f(\kappa)$

**Proof.** Let $\phi_i$ be defined as in Definition A9. To determine $s$ so that true(i) occurs, $p_i$ needs to determine $G$ and $H$ so that $G(0) = s$ and $H(0) = 0$. But as per Lemma A6, $G$ and $H$ are unknown in phase 1 and for any iteration $r \leq r^*$ in phase 2, and their distribution is indistinguishable from random. With $G$ and $H$ unknown, the probability of guessing $s$ is uniform, i.e., $1/|\mathcal{S}|$. This proves statements (1)–(3). For statement (4), the event $[\overline{\text{true}(-i)}|\text{late}]$ occurs if at some iteration $r > r^*$ all other players do not output $s$ correctly. If iteration $r^* + 1$ is reached, this implies that the strategy followed by $p_i$ follows the protocol $\Pi_1$ up to iteration $r^* + 1$ (otherwise, short or abort occurs). If $p_i$ follows $\Pi_1$ at iteration $r^* + 1$, then all other players will also learn about $s$, and $\overline{\text{true}(-i)}$ does not take place. If $p_i$ does not follow $\Pi_1$ at iteration $r = r^* + 1$ such that other players notice, then all other players will output $s^{r-1} = s$, and $\overline{\text{true}(-i)}$ will not take place as well. Statement (5) follows from Lemma A8. $\square$

**Definition A10.** *We now define the following experiments. Let $\sigma_i'$ denote any arbitrary polynomial-time strategy of $p_i$. Define $\Pr_0$ as the probabilities in Exp 0, by $\Pr_1$ the probabilities in Exp 1, and by $\Pr_2$ the probabilities in Exp 2, where Exp 0, 1, and 2 are as follows:*

***Exp 0**: This experiment runs $\Pi_1$ but with $p_i$ following $\sigma_i'$ and the rest following the prescribed strategies $\Pi$. In addition $p_i$ acquires $n - 1$ secret keys $(sk_{i_l})_{l \in [n] - 1}$ (through some leakage attacks).*

***Exp 1**: This experiment is the same as Exp 0, except that in the initialization phase (i.e., phase 0), the dealer computes $g_i^* = G(i) \oplus v_1$ and $h_i^* = H(i) \oplus v_2$, where $v_1$ and $v_2$ are uniformly sampled from the range of $V_E$.*

> ***Exp 2***: *This experiment is the same as Exp 1, except that given the k disjoint paths in phase 1 where $p_i$ receives (from the dealer) the set of shares $\{s_{i,1}, s_{i,2}, \ldots, s_{i,k}\}$ to reconstruct $sk_i$, one share $(s_{i,j})$ for some $j \in [k]$ is replaced by the dealer in phase 0 with a uniformly sampled number $\hat{s}_{i,j}$ in the range of $S_G$. Afterwards, the dealer reconstructs a different secret key for $p_i$, i.e., $\overline{sk}_i$, where $\overline{sk}_i$ is computed by the dealer using $S_R$ on input $(\{s_{i,j'}\}_{j' \in [k] \setminus j} \cup \hat{s}_{i,j})$. Afterwards, the dealer uses $\overline{sk}_i$ in computing for h' in phase 2.*

**Lemma A11.** *Given $\Pi_1(n, k)$, for any polynomial-time strategy $\sigma_i'$ adopted by $p_i$, there exists a negligible function f in $\kappa$ such that we have the following, given a fixed* stat $\in \{\texttt{maul}, \overline{\texttt{maul}}\}$ *for each statement. This result holds even if $\sigma_i'$ led $p_i$ to acquire less than n secret keys.*

1.  $|\Pr_0[\texttt{short}|\texttt{stat}] - \Pr_1[\texttt{short}|\texttt{stat}] \leq f(\kappa)$
2.  $|\Pr_0[\texttt{exact} \wedge \texttt{true(i)}|\texttt{stat}] - \Pr_1[\texttt{exact} \wedge \overline{\texttt{true(i)}}|\texttt{stat}]| \leq f(\kappa)$
3.  $|\Pr_0[\texttt{exact} \wedge \overline{\texttt{true(i)}}|\texttt{stat}] - \Pr_1[\texttt{exact} \wedge \overline{\texttt{true(i)}}|\texttt{stat}]| \leq f(\kappa)$
4.  $|\Pr_0[\texttt{late}|\texttt{stat}] - \Pr_1[\texttt{late}|\texttt{stat}]| \leq f(\kappa)$
5.  $|\Pr_0[\texttt{early} \wedge \texttt{true(i)}|\texttt{stat}] - \Pr_1[\texttt{early} \wedge \texttt{true(i)}|\texttt{stat}]| \leq f(\kappa)$
6.  $|\Pr_0[\texttt{early} \wedge \overline{\texttt{true(i)}}|\texttt{stat}] - \Pr_1[\texttt{early} \wedge \overline{\texttt{true(i)}}|\texttt{stat}]| \leq f(\kappa)$

**Proof.** From Definition A10, the only difference between Exp 0 and Exp 1 is in the computation of $g_i^*$ and $h_i^*$ by the dealer in phase 0. Following the notations in Lemma A3, let $\mathbf{e}_0$ denote the event that $g_i^*$ and $h_i^*$ are computed following $\Pi_1$ (i.e., Exp 0), and let $\mathbf{e}_1$ denote the event that $g_i^*$ and $h_i^*$ are sampled uniformly (i.e., Exp 1). Let $\phi_i$ be as defined in Definition A9, so that $\{\mathbf{e}_0, \mathbf{e}_1\} \notin \phi_i(I)$ for any information set $I$ in either phase 1 or phase 2. Suppose first that stat $= \overline{\texttt{maul}}$. A difference in player actions between $\mathbf{e}_0$ and $\mathbf{e}_1$ is sure to occur at iteration $r^* + 1$ given that under $\mathbf{e}_0$, both $r^*$ and $s$ will be learned by all players, while under $\mathbf{e}_1$, it is not clear if $r^*$ or $s$ will be learned by any player since $g_i^*$ and $h_i^*$ are sampled randomly. For phase 1 and at any iteration $r \leq r^*$ in phase 2, possible differences between the distribution of $\phi_i(I)$ under $\mathbf{e}_0$ and the distribution of $\phi_i(I)$ under $\mathbf{e}_1$ depend on differences in distribution of $\{g_i^*, h_i^*\}$ under $\mathbf{e}_0$ and its distribution under $\mathbf{e}_1$. We consider two cases that may arise here, affecting phase 2:

Case 1: In phase 2, for some iteration $r \leq r^*$, we have $H^r(0) = 0$ under $\mathbf{e}_1$.

Case 2: In phase 2, for all iterations $r \leq r^*$, we have $H^r(0) \neq 0$ under $\mathbf{e}_1$.

The probability of case 1 is negligible by the pseudorandomness of $\Lambda$. Note that under $\mathbf{e}_0$, at iteration $r = r^* + 1$, we have $H^r(0) = 0$, but for iterations $r \leq r^*$, we have $H^r(0) \neq 0$. Under $\mathbf{e}_1$, for iterations $r \leq r^*$, there is a possibility that $H^r(0) = 0$ given that $h_i^p = h_i^* \oplus \Lambda(sk_i, r)$, and $h_i^* \neq H(i) \oplus \Lambda(sk_i, r)$ with non-negligible probability. We specify the conditions that are needed for $H^r(0) = 0$ to occur at iteration $r \leq r^*$ under $\mathbf{e}_1$. Let $[(1, h_c^p(1)), (2, h_c^p(2)), \ldots, (n-1, h_c^p(n-1))]$ represent the tuple of coordinates given information in $\phi_i(I)$ at iteration $r \leq r^*$ in phase 2, where $h_c^p(j) = h_j^* \oplus \Lambda(sk_j, r)$ for $j \in [n-1]$ (and $h_i^*$ is randomly sampled). Combining this tuple with the coordinate $(0, 0)$, results in an interpolated candidate polynomial $H_c^r$ such that $H_c^r(0) = 0$. This gives a target value $H_c^r(n) = \hat{y}$. It follows that $H^r(0) = 0$ if and only if $h_n^* \oplus \Lambda(sk_n, r) = \hat{y}$, or $\hat{y} \oplus h_n^* = \Lambda(sk_n, r)$. By the pseudorandomness of $\Lambda$, the probability that $\hat{y} \oplus h_n^* = \Lambda(sk_n, r)$ is close to uniform. Thus, the probability of case 1 is negligible. As for case 2, given that $H^r(0) \neq 0$ for $r \leq r^*$, the situation of players under $\mathbf{e}_1$ is no different from their situation under $\mathbf{e}_0$. Since case 1 is negligible, this implies that its complement, i.e., case 2, is non-negligible in probability. Given this fact, we note that since $sk_n$ is unknown, we can apply Lemma A6, where the distribution of $g_i^*$ and $h_i^*$ is indistinguishable from random in phases 1–2. Hence, sampling $g_i^*$ and $h_i^*$ uniformly as in $\mathbf{e}_1$ is not noticeable, and the distribution of $\phi_i(I)$ under $\mathbf{e}_0$ is no different from the distribution of $\phi_i(I)$ under $\mathbf{e}_1$ for all information sets $I$ in phase 1 and for all information sets $I$ in iteration $r \leq r^*$ in phase 2, i.e., $\Pr_0[\phi_i(I)]$ is indistinguishable from $\Pr_1[\phi_i(I)]$ for all information sets $I$ in phase 1 and

for all information sets $I$ in iteration $r \leq r^*$ in phase 2. By Lemma A3, statements (1)–(6) follow under $\mathtt{stat} = \overline{\mathtt{maul}}$.

Suppose now that $\mathtt{stat} = \mathtt{maul}$, where $p_i$ modified a share in phase 1. As per $\Pi_1$, for players $p_j \neq p_i$, no abort is performed in phase 1 due to a share's value. It follows that for $p_j \neq p_i$, their actions in phase 1 are independent of $\mathtt{maul}$ or $\overline{\mathtt{maul}}$. For $p_i$, following the above paragraph, we have that the distribution of $\phi_i$ under $\mathbf{e}_1$ in phase 1 is indistinguishable from the distribution of $\phi_i$ in phase 1 under $\mathbf{e}_2$. Since this holds even if $\mathtt{maul}$ occurs, statement (1) follows under phase 1. For phase 2, as shown in the proof of Lemma A8, with non-negligible probability, all players abort at iteration 1 and are forced to output a random guess for $s$ due to $\mathtt{maul}$. Thus, under both $\mathbf{e}_0$ and $\mathbf{e}_1$, the probability of the event $\mathtt{early} \wedge \overline{\mathtt{true(i)}}|\mathtt{maul}$ in statement (6) holds with non-negligible probability. All other events in statements (2)–(5) are negligible, and the Lemma follows under phase 2. □

**Lemma A12.** *Under $\Pi_1(n,k)$, for any polynomial-time strategy $\sigma'_i$ adopted by $p_i$, there exists a negligible function $f$ in $\kappa$ such that we have the following, given a fixed $\mathtt{stat} \in \{\mathtt{maul}, \overline{\mathtt{maul}}\}$ for each statement. This result holds even if $\sigma'_i$ led $p_i$ to acquire less than $n$ secret keys:*

1. $|\mathrm{Pr}_1[\mathtt{short} \wedge \mathtt{true(i)}|\mathtt{stat}] - \mathrm{Pr}_2[\mathtt{short} \wedge \mathtt{true(i)}|\mathtt{stat}]| \leq f(\kappa)$
2. $|\mathrm{Pr}_1[\mathtt{short} \wedge \overline{\mathtt{true(i)}}|\mathtt{stat}] - \mathrm{Pr}_2[\mathtt{short} \wedge \overline{\mathtt{true(i)}}|\mathtt{stat}]| \leq f(\kappa)$
3. $|\mathrm{Pr}_1[\mathtt{exact} \wedge \mathtt{true(i)}] - \mathrm{Pr}_2[\mathtt{exact} \wedge \mathtt{true(i)}]| \leq f(\kappa)$
4. $|\mathrm{Pr}_1[\mathtt{exact} \wedge \overline{\mathtt{true(i)}}] - \mathrm{Pr}_2[\mathtt{exact} \wedge \overline{\mathtt{true(i)}}]| \leq f(\kappa)$
5. $|\mathrm{Pr}_1[\mathtt{late}|\mathtt{stat}] - \mathrm{Pr}_2[\mathtt{late}|\mathtt{stat}]| \leq f(\kappa)$
6. $|\mathrm{Pr}_1[\mathtt{early} \wedge \mathtt{true(i)}|\mathtt{stat}] - \mathrm{Pr}_2[\mathtt{early} \wedge \mathtt{true(i)}|\mathtt{stat}]| \leq f(\kappa)$
7. $|\mathrm{Pr}_1[\mathtt{early} \wedge \overline{\mathtt{true(i)}}|\mathtt{stat}] - \mathrm{Pr}_2[\mathtt{early} \wedge \overline{\mathtt{true(i)}}|\mathtt{stat}]| \leq f(\kappa)$

**Proof.** From Definition A10, the only difference between $\mathtt{Exp\ 1}$ and $\mathtt{Exp\ 2}$ is that some share $\hat{s}_{i,j}$ ($j \in [k]$) transmitted by the dealer to $p_i$ in phase 1 is uniformly sampled in $\mathtt{Exp\ 2}$. Without loss of generality, let this uniformly sampled share be $\hat{s}_{i,k}$, i.e., the share transmitted along the $k$th path from the dealer to $p_i$. Following the notations in Lemma A3, let $\mathbf{e}_1$ denote the event that $s_{i,k}$ is computed using $S_G$ but $h^*_i$ and $g^*_i$ are sampled uniformly (i.e., $\mathtt{Exp\ 1}$), and let $\mathbf{e}_2$ denote the event that $\hat{s}_{i,k}$, $h^*_i$ and $g^*_i$ are sampled uniformly (i.e., $\mathtt{Exp\ 2}$). Let $\phi_i$ be as defined in Definition A9, so that $\{\mathbf{e}_1, \mathbf{e}_2\} \notin \phi_i(I)$ for any information set in either phase 1 or phase 2. Suppose first that $\mathtt{stat} = \overline{\mathtt{maul}}$. We consider three cases brought about by the change in $\mathbf{e}_2$:

Case 1: The distribution of $\hat{s}_{i,k}$ is distinguishable from the distribution of $s_{i,k}$ conditional on $\{s_{i,j}\}_{j \in [k-1]}$ in phase 1.

Case 2: In phase 2, for some iteration $r \leq r^*$, we have $H^r(0) = 0$.

Case 3: In phase 2, for all iterations $r \leq r^*$, we have $H^r(0) \neq 0$.

The probability of case 1 is negligible by the security of the secret sharing scheme. From Lemma A1, given $k-1$ shares, one cannot tell the true value of $sk_i$. Hence, conditional on $\{s_{i,j}\}_{j \in [k-1]}$, from the point of view of $p_i$, the distribution of possible values of $sk_i$ under $\mathbf{e}_1$ is indistinguishable from the distribution of possible values of $\hat{sk}_i$ in $\mathbf{e}_2$. This implies that from the point of view of $p_i$, the distribution of possible values of the $k$th share such that the secret $sk_i$ is reconstructed is indistinguishable from the distribution of possible values of the $k$th share such that the secret $\hat{sk}_i$ is reconstructed. Hence, the distribution of $\phi_i(I)$ (with $sk_i \in \phi_i(I)$) for any information set $I$ in phase 1 is indistinguishable from the distribution of $\phi_i(I)$ (with $\hat{sk}_i \in \phi_i(I)$) for any information set $I$ in phase 1.

The probability of case 2 is likewise negligible by the pseudorandomness of $\Lambda$. First, we note that under $\mathbf{e}_1$, the probability that $H^r(0) = 0$ is negligible for any iteration $r \leq r^*$, as shown in the proof of Lemma A11. Given this, assume that $H^r(0) = 0$ for any iteration $r \leq r^*$ under $\mathbf{e}_1$. Under $\mathbf{e}_2$, there is a possibility that $H^r(0) = 0$ for some iteration $r \leq r^*$ due to the change from $sk_i$ to $\hat{sk}_i$. We consider the conditions that are needed for $H^r(0) = 0$ to occur at iteration $r \leq r^*$ under $\mathbf{e}_2$. Let $[(1, h^p_c(1)), (2, h^p_c(2)), \ldots, (n-1, h^p_c(n-1))]$

represent the tuple of coordinates formed from $\phi_i(I)$ at iteration $r \le r^*$ in phase 2, where $h_c^p(j) = h_j^* \oplus \Lambda(sk_j, r)$ for $j \in [n-1]$ (and where $\hat{sk}_i \ne sk_i$ under $\mathbf{e}_2$, and $h_i^*$ is randomly sampled in both $\mathbf{e}_1$ and $\mathbf{e}_2$). Combining this tuple with the coordinate $(0,0)$ results in an interpolated candidate polynomial $H_c^r$ such that $H_c^r(0) = 0$. This gives a target value $H_c^r(n) = \hat{y}$. It follows that $H^r(0) = 0$ if and only if $h_n^* \oplus \Lambda(sk_n, r) = \hat{y}$ or $\hat{y} \oplus h_n^* = \Lambda(sk_n, r)$. By the pseudorandomness of $\Lambda$, the probability that $\hat{y} \oplus h_n^* = \Lambda(sk_n, r)$ is close to uniform. Thus, the probability of case 2 is negligible. This in turn implies that the complement of case 2 in phase 2, i.e., case 3, is non-negligible. However, given case 3, the situation of players under $\mathbf{e}_2$ is no different from their situation under $\mathbf{e}_1$ and $\mathbf{e}_0$. Moreover, by the pseudorandomness of $\Lambda$, from the point of view of $p_i$, the distribution of $\Lambda(sk_i, r)$ is indistinguishable from the distribution of $\Lambda(\hat{sk}_i, r)$ for $r > 0$. It follows that the distribution of $\phi_i(I)$ under $\mathbf{e}_1$ is no different from the distribution of $\phi_i(I)$ under $\mathbf{e}_2$ for all information sets $I$ in phase 1 and for all information sets $I$ in iteration $r \le r^*$ in phase 2. By Lemma A3, statements (1)–(7) follow under $\mathtt{stat} = \overline{\mathtt{maul}}$.

Suppose now that $\mathtt{stat=maul}$, where $p_i$ modified a share in phase 1. As per $\Pi_1$, for players $p_j \ne p_i$, no abort is performed in phase 1 due to a share's value. It follows that for $p_j \ne p_i$, their actions in phase 1 are independent of $\mathtt{maul}$ or $\overline{\mathtt{maul}}$ regardless of the change from $sk_i$ to $\hat{sk}_i$. For $p_i$, following the above paragraph, we have that the distribution of $sk_i$ is indistinguishable from the distribution of $\hat{sk}_i$ conditional on $k-1$ other shares. Since, this holds even if $\mathtt{maul}$ occurs, given Lemma A8, statement (1)–(2) follows under phase 1. For phase 2, as shown in the proof of Lemma A8, with non-negligible probability, all players already abort at iteration 1 under $\mathbf{e}_1$ and are forced to output a random guess for $s$ due to $\mathtt{maul}$. The reasoning of Lemma A8 holds even if $sk_i$ is changed to $\hat{sk}_i$. Thus, under both $\mathbf{e}_1$ and $\mathbf{e}_2$, the probability of the event $\mathtt{early} \wedge \mathtt{true(i)}|\mathtt{maul}$ in statement (7) holds with non-negligible probability. All other events in statements (2)–(6) are negligible and the Lemma follows under phase 2. $\square$

**Proof of Theorem 1.** The proof for this theorem follows the flow in the proof of [8]. Let $\mathtt{Exp}$ 0, $\mathtt{Exp}$ 1 and $\mathtt{Exp}$ 2 be defined as in Definition A10. Denote by $(\sigma_i', \sigma_{-i})$ a polynomial-time strategy where $p_i$ follows some polynomial-time strategy $\sigma_i'$, and all other players following strategies $\sigma_{-i}$ prescribed by $\Pi$. For correctness of $\Pi_1$, in phase 2, if all active $n$ parties run $\Pi$ honestly, the correct secret is reconstructed by Lagrange Interpolation unless: (1) $r^* \ge 2^\kappa - 1$ or (2) if for some $r < r^* + 1$ and $i \in [n]$, we have $H(i) = h_i^r = \Lambda(sk_i, r)$. Sampling $r^*$ such that $r^* \ge 2^\kappa$ as in (1) occurs with negligible probability and the pseudorandomness of $\Lambda$ implies that (2) occurs with negligible probability as well. Thus, the correctness of $\Pi$ with overwhelming probability is shown. Denote by $u_i(\sigma_i', \sigma_{-i})$ the expected utility of player $p_i$ across phases 1 and 2 if $(\sigma_i', \sigma_{-i})$ is followed. Denote by $u_i^2(\sigma_i', \sigma_{-i})$ the expected utility of player $p_i$ achieved during phase 2 (conditional on the event that it has reached phase 2 under $\sigma_i'$). Note that $u_i^2(\sigma_i', \sigma_{-i}) > 0$ if and only if $\mathtt{short}$ has not occurred. We first consider the differences in utilities under the experiments in phase 2, followed by a combination of the differences in utilities under the experiments in both phase 1 and 2—similar to a backward-induction process. Combining all possibilities of events described in Definition A7 that apply to phase 2, we have the following expression for $u_i^2(\sigma_i', \sigma_{-i})$:

$$
\begin{aligned}
u_i^2(\sigma_i', \sigma_{-i}) \le \; &(\mathrm{Pr}_0[\mathtt{maul}] \times \mathtt{util}_i(\mathrm{Pr}_0, \mathtt{maul})) \\
&+ (\mathrm{Pr}_0[\overline{\mathtt{maul}}] \times \mathtt{util}_i(\mathrm{Pr}_0, \overline{\mathtt{maul}}))
\end{aligned}
$$

where $\mathtt{util_i} : \{\mathrm{Pr}_0, \mathrm{Pr}_1\} \times \{\mathtt{maul}, \overline{\mathtt{maul}}\} \to \mathbb{R}$ is a function defined as follows, where $\mathtt{stat} \in \{\mathtt{maul}, \overline{\mathtt{maul}}\}$:

$$\mathtt{util}_i(\mathrm{Pr}, \mathtt{stat})$$

$$
\begin{aligned}
:= {} & (U^+ \times \mathrm{Pr}[\mathtt{exact} \wedge \mathtt{true(i)} \wedge \overline{\mathtt{true(-i)}} | \mathtt{stat}]) \\
& + (U^- \times \mathrm{Pr}[\mathtt{exact} \wedge \overline{\mathtt{true(i)}} \wedge \overline{\mathtt{true(-i)}} | \mathtt{stat}]) \\
& + (U \times \mathrm{Pr}[\mathtt{exact} \wedge \mathtt{true(i)} \wedge \mathtt{true(-i)} | \mathtt{stat}]) \\
& + (U^- \times \mathrm{Pr}[\mathtt{exact} \wedge \overline{\mathtt{true(i)}} \wedge \mathtt{true(-i)} | \mathtt{stat}]) \\
& + (U^+ \times \mathrm{Pr}[\mathtt{early} \wedge \mathtt{true(i)} \wedge \overline{\mathtt{true(-i)}} | \mathtt{stat}]) \\
& + (U^- \times \mathrm{Pr}[\mathtt{early} \wedge \overline{\mathtt{true(i)}} \wedge \overline{\mathtt{true(-i)}} | \mathtt{stat}]) \\
& + (U \times \mathrm{Pr}[\mathtt{early} \wedge \mathtt{true(i)} \wedge \mathtt{true(-i)} | \mathtt{stat}]) \\
& + (U^- \times \mathrm{Pr}[\mathtt{early} \wedge \overline{\mathtt{true(i)}} \wedge \mathtt{true(-i)} | \mathtt{stat}]) \\
& + (U^+ \times \mathrm{Pr}[\mathtt{late} \wedge \mathtt{true(i)} \wedge \overline{\mathtt{true(-i)}} | \mathtt{stat}]) \\
& + (U^- \times \mathrm{Pr}[\mathtt{late} \wedge \overline{\mathtt{true(i)}} \wedge \overline{\mathtt{true(-i)}} | \mathtt{stat}]) \\
& + (U \times \mathrm{Pr}[\mathtt{late} \wedge \mathtt{true(i)} \wedge \mathtt{true(-i)} | \mathtt{stat}]) \\
& + (U^- \times \mathrm{Pr}[\mathtt{late} \wedge \overline{\mathtt{true(i)}} \wedge \mathtt{true(-i)} | \mathtt{stat}])
\end{aligned}
$$

Let $\overline{u}_i^2(\sigma_i', \sigma_{-i})$ represent some upper bound for $u_i^2(\sigma_i', \sigma_{-i})$. To come up with an expression for $\overline{u}_i^2(\sigma_i', \sigma_{-i})$, we modify some terms in $\mathtt{util}_i(\mathrm{Pr}, \mathtt{stat})$. All probabilities that involve events with $\mathtt{exact} \wedge \overline{\mathtt{true(i)}}$ can be ruled out since there exists a polynomial-time strategy for which this event occurs with probability 0. For instance, take the strategy, form a guess for $r = r^*$, then output $s^r$ at iteration $r$. It follows that if $\mathtt{exact}$ occurs, $\mathtt{true(i)}$ automatically occurs as well. The probability $\mathrm{Pr}[\mathtt{exact} \wedge \mathtt{true(-i)}]$ can be replaced with some negligible function (say $1/|\mathcal{S}|$) given that if $\mathtt{exact}$ occurs at iteration $r$ since $p_i$ aborts, other players will output $s^{r-1}$, which is not equal to the secret $s$ with non-negligible probability. The same applies to $\mathrm{Pr}[\mathtt{early} \wedge \mathtt{true(-i)}]$. We also note that $\mathrm{Pr}[\overline{\mathtt{true(-i)}} | \mathtt{late}, \mathtt{stat}] = 0$ if $\mathtt{stat} = \overline{\mathtt{maul}}$ as per Lemma A10, since at iteration $r = r^* + 1$, all other players will output $s^{r-1} = s$ regardless of the actions of $p_i$. Moreover, any strategy such that $\mathrm{Pr}[\overline{\mathtt{true(i)}} | \mathtt{late}, \overline{\mathtt{maul}}]$ occurs with positive probability is strictly dominated by a strategy that sets the probability of this event to 0, i.e., since $p_i$ reached $\mathtt{late}$, this means that it followed strategies equivalent to $\Pi_1$ up to iteration $r^* + 1$. At iteration $r^* + 1$, all players can learn both $r^*$ and $s$. Under $\Pi_1$, all other players will output $s$ regardless of the actions of $p_i$ at iteration $r^* + 1$, so $p_i$ will gain the most utility if it follows other players and output $s$ as well. From these statements, we denote the upper bound for $u_i^2(\sigma_i', \sigma_{-i})$, as follows:

$$
\begin{aligned}
\overline{u}_i^2(\sigma_i', \sigma_{-i}) = {} & (\mathrm{Pr}_0[\mathtt{maul}] \times \overline{\mathtt{util}}_i(\mathrm{Pr}_0, \mathtt{maul})) \\
& + (\mathrm{Pr}_0[\overline{\mathtt{maul}}] \times \overline{\mathtt{util}}_i(\mathrm{Pr}_0, \overline{\mathtt{maul}}))
\end{aligned}
$$

where $\overline{\mathtt{util}}_i : \{\mathrm{Pr}_0, \mathrm{Pr}_1\} \times \{\mathtt{maul}, \overline{\mathtt{maul}}\} \to \mathbb{R}$ is a function defined below, making use of the following facts: (1) $U^+ > U > U^-$; (2) the sum of $\mathrm{Pr}[\mathtt{exact} \wedge \mathtt{true(i)} \wedge \overline{(\mathtt{true(i)})}]$ and $\mathrm{Pr}[\mathtt{exact} \wedge \mathtt{true(i)} \wedge \mathtt{true(i)}]$ is less than or equal to $\mathrm{Pr}[\mathtt{exact}]$; (3) the sum of $\mathrm{Pr}[\mathtt{early} \wedge \mathtt{true(i)} \wedge \mathtt{true(-i)}]$ and $\mathrm{Pr}[\mathtt{early} \wedge \mathtt{true(i)} \wedge \overline{\mathtt{true(-i)}}]$ is equal to $\mathrm{Pr}[\mathtt{early} \wedge \mathtt{true(i)}]$ (similarly for $\mathrm{Pr}[\mathtt{early} \wedge \overline{\mathtt{true(i)}}]$); and (4) if $\mathtt{maul}$ occurs, from Lemma A8, the probability of $\mathtt{true(i)}$ is equal to random so that an upper bound for $u_i^2(\sigma_i', \sigma_{-i})$ implies that $\overline{\mathtt{maul}}$ holds:

$$\overline{\texttt{util}}_i(\mathrm{Pr}, \overline{\texttt{maul}}) := (U^+ \times \mathrm{Pr}[\texttt{exact}|\overline{\texttt{maul}}])$$
$$+ (U^+ \times \mathrm{Pr}[\texttt{early} \wedge \texttt{true(i)}|\overline{\texttt{maul}}])$$
$$+ (U^- \times \mathrm{Pr}[\texttt{early} \wedge \overline{\texttt{true(i)}}|\overline{\texttt{maul}}])$$
$$+ (U \times \mathrm{Pr}[\texttt{late}|\overline{\texttt{maul}}])$$
$$\overline{\texttt{util}}_i(\mathrm{Pr}, \texttt{maul}) := (U^+ \times 1/|\mathcal{S}|) + (U^- \times (1 - 1/|\mathcal{S}|))$$

We now define $U_{\texttt{exp\_1}}$ as follows, which uses probabilities of the game under Exp 1:

$$U_{\texttt{exp\_1}} = (\mathrm{Pr}_1[\texttt{maul}] \times \overline{\texttt{util}}_i(\mathrm{Pr}_1, \texttt{maul}))$$
$$+ (\mathrm{Pr}_1[\overline{\texttt{maul}}] \times \overline{\texttt{util}}_i(\mathrm{Pr}_1, \overline{\texttt{maul}}))$$

From Lemma A11, we have $|\overline{u}_i^2(\sigma_i', \sigma_{-i}) - U_{\texttt{exp\_1}}| \leq f(\kappa)$ for some negligible function $f$ in $\kappa$. It follows that $U_{\texttt{exp\_1}}$ also represents an upper bound for $u_i^2(\sigma_i', \sigma_{-i})$ with some negligible difference. Let $\texttt{abort} \wedge \texttt{stat} := (\texttt{early} \wedge \texttt{stat}) \cup (\texttt{exact} \wedge \texttt{stat})$ for $\texttt{stat} \in \{\texttt{maul}, \overline{\texttt{maul}}\}$. Information-theoretically, we have $\mathrm{Pr}_1[\texttt{exact}|\texttt{abort}, \texttt{stat}] = \beta$ and $\mathrm{Pr}_1[\texttt{early}|\texttt{abort}, \texttt{stat}] = 1 - \beta$ since $\beta$ is independent of $\texttt{stat}$. Using Lemma A10, we have the following bound for $U_{\texttt{exp\_1}}$:

$$U_{\texttt{exp\_1}} = \big[\big[U^+ \times \big(\mathrm{Pr}_1[\texttt{exact}|\texttt{abort}, \overline{\texttt{maul}}] + (\mathrm{Pr}_1[\texttt{true(i)}|\texttt{early}, \overline{\texttt{maul}}]$$
$$\times \mathrm{Pr}_1[\texttt{early}|\texttt{abort}, \overline{\texttt{maul}}])\big)\big]$$
$$+ \big[U^- \times \big(\mathrm{Pr}_1[\overline{\texttt{true(i)}}|\texttt{early}] \times \mathrm{Pr}_1[\texttt{early}|\texttt{abort}, \overline{\texttt{maul}}]\big)\big]\big]$$
$$\times \mathrm{Pr}_1[\texttt{abort}|\overline{\texttt{maul}}] \times \mathrm{Pr}_1[\overline{\texttt{maul}}]$$
$$+ \big[\big[U \times \mathrm{Pr}_1[\texttt{late}|\overline{\texttt{maul}}]\big]\big] \times \mathrm{Pr}_1[\overline{\texttt{maul}}] + (\overline{\texttt{util}}_i(\mathrm{Pr}_1, \texttt{maul}))$$
$$= \big[U^+ \times [\beta + (1/|\mathcal{S}| \times (1 - \beta))]\big] \times \mathrm{Pr}_1[\texttt{abort}|\overline{\texttt{maul}}] \times \mathrm{Pr}_1[\overline{\texttt{maul}}]$$
$$+ \big[U^- \times (1 - 1/|\mathcal{S}|) \times (1 - \beta)\big] \times \mathrm{Pr}_1[\texttt{abort}|\overline{\texttt{maul}}] \times \mathrm{Pr}_1[\overline{\texttt{maul}}]$$
$$+ \big[U \times (1 - \mathrm{Pr}_1[\texttt{abort}, \overline{\texttt{maul}}])\big] \times \mathrm{Pr}_1[\overline{\texttt{maul}}] + [(U^+ \times 1/|\mathcal{S}|)$$
$$+ (U^- \times (1 - 1/|\mathcal{S}|))] \times \mathrm{Pr}_1[\texttt{maul}]$$
$$= \big[\big[U^+ \times \big(\beta + (1/|\mathcal{S}| \times (1 - \beta))\big)\big] + \big[U^- \times (1 - 1/|\mathcal{S}|) \times (1 - \beta)\big] - U\big]$$
$$\times \mathrm{Pr}_1[\texttt{abort}|\overline{\texttt{maul}}] \times \mathrm{Pr}_1[\overline{\texttt{maul}}] + [U \times \mathrm{Pr}_1[\overline{\texttt{maul}}]] + [(U^+ \times 1/|\mathcal{S}|)$$
$$+ (U^- \times (1 - 1/|\mathcal{S}|))] \times (1 - \mathrm{Pr}_1[\overline{\texttt{maul}}])$$

Simplifying the above equations, we have:

$$U_{\texttt{exp\_1}} = U_{\texttt{rand}} + \mathrm{Pr}_1[\overline{\texttt{maul}}] \times \big[[U - U_{\texttt{rand}}] + [(\beta \times U^+) + (1 - \beta) \times U_{\texttt{rand}} - U]$$
$$\times \mathrm{Pr}_1[\texttt{abort}|\overline{\texttt{maul}}]\big]$$

By assumption, we have $U > U_{\texttt{rand}}$ and $[(\beta \times U^+) + (1 - \beta) \times U_{\texttt{rand}} - U] < 0$. Hence, $U_{\texttt{exp\_1}}$ is maximized if $\mathrm{Pr}_1[\overline{\texttt{maul}}] > 0$ and if $\mathrm{Pr}_1[\texttt{abort}|\overline{\texttt{maul}}]$ is minimized. Using the above equations, we define the following:

$$U_{\texttt{exp\_1}|\overline{\texttt{maul}}} = U + [(\beta \times U^+) + (1 - \beta) \times U_{\texttt{rand}} - U] \times \mathrm{Pr}_1[\texttt{abort}|\overline{\texttt{maul}}]$$
$$U_{\texttt{exp\_1}|\texttt{maul}} = U_{\texttt{rand}},$$

so that $U_{\texttt{exp\_1}} = (\mathrm{Pr}_1[\texttt{maul}] \times U_{\texttt{exp\_1}|\texttt{maul}}) + (\mathrm{Pr}_1[\overline{\texttt{maul}}] \times U_{\texttt{exp\_1}|\overline{\texttt{maul}}})$. We now consider differences in utilities between Exp 1 and Exp 2, as well as combine phases 1 and 2 of the protocol. Given any polynomial-time strategy $(\sigma_i^1, \sigma_{-i})$, we have the following expression for $u_i(\sigma_i', \sigma_{-i})$, using the following facts: (1) $U^+ > U > U^-$, and (2) for $\texttt{stat} \in \{\texttt{maul}, \overline{\texttt{maul}}\}$, the sum of $\mathrm{Pr}_1[\texttt{short} \wedge \texttt{true(i)} \wedge \texttt{true(-i)}|\texttt{stat}]$ and $\mathrm{Pr}_1[\texttt{short} \wedge \texttt{true(i)} \wedge \overline{\texttt{true(-i)}}|$

stat]) is equal to the probability $\Pr_1[\texttt{short} \wedge \texttt{true(i)}|\texttt{stat}]$ (and the same applies as well to $\Pr_1[\texttt{short} \wedge \overline{\texttt{true(i)}}]$):

$$u_i(\sigma_i', \sigma_{-i}) \leq (\Pr_1[\texttt{maul}] \times \texttt{util}_i^2(\Pr_1, \texttt{maul})) + (\Pr_1[\overline{\texttt{maul}}] \times \texttt{util}_i^2(\Pr_1, \overline{\texttt{maul}}))$$

where $\texttt{util}_i^2 : \{\Pr_1, \Pr_2\} \times \{\texttt{maul}, \overline{\texttt{maul}}\} \to \mathbb{R}$ is a function defined as:

$$\texttt{util}_i^2(\Pr, \texttt{stat}) := (U^+ \times \Pr[\texttt{short} \wedge \texttt{true(i)}|\texttt{stat}]) + (U^- \times \Pr[\texttt{short} \wedge \overline{\texttt{true(i)}}|\texttt{stat}])$$
$$+ (U_{\texttt{exp\_1}} \times \Pr[\overline{\texttt{short}}|\texttt{stat}])$$

Let $\overline{u}_i(\sigma_i', \sigma_{-i})$ represent an upper bound for $u_i(\sigma_i', \sigma_{-i})$ which the above expression holds with equality. We now define $U_{\texttt{exp\_2}}$ as follows:

$$U_{\texttt{exp\_2}} := (\Pr_2[\texttt{maul}] \times \texttt{util}_i^2(\Pr_2, \texttt{maul})) + (\Pr_2[\overline{\texttt{maul}}] \times \texttt{util}_i^2(\Pr_2, \overline{\texttt{maul}}))$$

From Lemma A8, $\Pr_2[\texttt{true(i)}|\texttt{short}, \overline{\texttt{maul}}] = 1/|\mathcal{S}|$ and $\Pr_2[\texttt{true(i)}|\texttt{maul}] = 1/|\mathcal{S}|$. Using these facts we have:

$$
\begin{aligned}
U_{\texttt{exp\_2}} = &\left[\Pr_2[\overline{\texttt{maul}}] \times \Pr_2[\texttt{short}|\overline{\texttt{maul}}] \times \left[(U^+ \times \Pr_2[\texttt{true(i)}|\texttt{short}|\overline{\texttt{maul}}])\right.\right. \\
&+ (U^- \times \Pr_2[\overline{\texttt{true(i)}}|\texttt{short}|\overline{\texttt{maul}}])] + \left[\Pr_2[\overline{\texttt{maul}}] \times (U_{\texttt{exp\_1}|\overline{\texttt{maul}}} \times \Pr_2[\overline{\texttt{short}}|\overline{\texttt{maul}}])\right] \\
&+ \left[\Pr_2[\texttt{maul}] \times \Pr_2[\texttt{short}|\texttt{maul}] \times \left[(U^+ \times \Pr_2[\texttt{true(i)}|\texttt{short}|\texttt{maul}])\right.\right. \\
&+ (U^- \times \Pr_2[\overline{\texttt{true(i)}}|\texttt{short}|\texttt{maul}])] + \left[\Pr_2[\texttt{maul}] \times (U_{\texttt{exp\_1}|\texttt{maul}} \times \Pr_2[\overline{\texttt{short}}|\texttt{maul}])\right] \\
= &\Pr_2[\overline{\texttt{maul}}] \times \left[[\Pr_2[\texttt{short}|\overline{\texttt{maul}}] \times [(U^+ \times 1/|\mathcal{S}|) + (U^- \times (1 - 1/|\mathcal{S}|))]\right] \\
&+ (U_{\texttt{exp\_1}|\overline{\texttt{maul}}} \times \Pr_2[\overline{\texttt{short}}|\overline{\texttt{maul}}])] + \Pr_2[\texttt{maul}] \times \left[[\Pr_2[\texttt{short}|\texttt{maul}]\right. \\
&\times [(U^+ \times 1/|\mathcal{S}|) + (U^- \times (1 - 1/|\mathcal{S}|))]] + (U_{\texttt{exp\_1}|\texttt{maul}} \times \Pr_2[\overline{\texttt{short}}|\texttt{maul}])] \\
= &\Pr_2[\overline{\texttt{maul}}] \times [(\Pr_2[\texttt{short}|\overline{\texttt{maul}}] \times U_{\texttt{rand}}) + (U_{\texttt{exp\_1}|\overline{\texttt{maul}}} \times (1 - \Pr_2[\texttt{short}|\overline{\texttt{maul}}]))] \\
&+ \Pr_2[\texttt{maul}] \times [(\Pr_2[\texttt{short}|\texttt{maul}] \times U_{\texttt{rand}}) + (U_{\texttt{exp\_1}|\texttt{maul}} \times (1 - \Pr_2[\texttt{short}|\texttt{maul}]))] \\
= &\Pr_2[\overline{\texttt{maul}}] \times [U_{\texttt{exp\_1}|\overline{\texttt{maul}}} + (U_{\texttt{rand}} - U_{\texttt{exp\_1}|\overline{\texttt{maul}}}) \times \Pr_2[\texttt{short}|\overline{\texttt{maul}}]] \\
&+ \Pr_2[\texttt{maul}] \times [U_{\texttt{exp\_1}|\texttt{maul}} + (U_{\texttt{rand}} - U_{\texttt{exp\_1}|\texttt{maul}}) \times \Pr_2[\texttt{short}|\texttt{maul}]] \\
= &\Pr_2[\overline{\texttt{maul}}] \times [U_{\texttt{exp\_1}|\overline{\texttt{maul}}} + (U_{\texttt{rand}} - U_{\texttt{exp\_1}|\overline{\texttt{maul}}}) \times \Pr_2[\texttt{short}|\overline{\texttt{maul}}]] \\
&+ \Pr_2[\texttt{maul}] \times [U_{\texttt{rand}}]
\end{aligned}
$$

where the last line uses the definition $U_{\texttt{exp\_1}|\texttt{maul}} = U_{\texttt{rand}}$. This gives us:

$$U_{\texttt{exp\_2}} = U_{\texttt{rand}} + \Pr_2[\overline{\texttt{maul}}] \times [(U_{\texttt{exp\_1}|\overline{\texttt{maul}}} - U_{\texttt{rand}})$$
$$+ (U_{\texttt{rand}} - U_{\texttt{exp\_1}|\overline{\texttt{maul}}}) \times \Pr_2[\texttt{short}|\overline{\texttt{maul}}]]$$

From Lemma A12, we have $|\overline{u}_i(\sigma_i', \sigma_{-i}) - U_{\texttt{exp\_2}}| \leq f(\kappa)$ for some negligible function $f$ in $\kappa$. It follows that $U_{\texttt{exp}_2}$ represents an upper bound for $u_i(\sigma_i', \sigma_{-i})$ with some negligible difference. Define the equations (note the change from $\Pr_1$ to $\Pr_2$):

$$\hat{U}_{\texttt{exp\_1}|\overline{\texttt{maul}}} = U + [(\beta \times U^+) + (1 - \beta) \times U_{\texttt{rand}} - U] \times \Pr_2[\texttt{abort}|\overline{\texttt{maul}}]$$
$$\hat{U}_{\texttt{exp\_1}|\texttt{maul}} = U_{\texttt{rand}}$$

Using Lemma A12 again, both $\hat{U}_{\mathtt{exp\_1}|\overline{\mathtt{maul}}}$ and $\hat{U}_{\mathtt{exp\_1}|\mathtt{maul}}$ differ from $U_{\mathtt{exp\_1}|\overline{\mathtt{maul}}}$ and $U_{\mathtt{exp\_1}|\mathtt{maul}}$ by a negligible factor, respectively. This gives us the following expression, where $f$ is a negligible function in $\kappa$:

$$\hat{U}_{\mathtt{exp\_2}} + f(\kappa) = U_{\mathtt{exp\_2}} = U_{\mathtt{rand}} + \Pr_2[\overline{\mathtt{maul}}] \times [(\hat{U}_{\mathtt{exp\_1}|\overline{\mathtt{maul}}} - U_{\mathtt{rand}})$$
$$+ (U_{\mathtt{rand}} - \hat{U}_{\mathtt{exp\_1}|\overline{\mathtt{maul}}}) \times \Pr_2[\mathtt{short}|\overline{\mathtt{maul}}]] + f(\kappa)$$

Finally, to prove that $\Pi$ is a computational Nash equilibrium, we have to show that for any polynomial-time strategy $\sigma'_i$ adopted by $p_i$, we have $u_i(\sigma'_i, \sigma_{-i}) \le U + f(k)$ for some negligible function $f$ in $\kappa$. Combining all of the above, we have the following, which proves $\Pi_1$ is a computational Nash equilibrium (i.e., $u_i(\sigma'_i, \sigma_{-i}) \le U + f(\kappa)$ for some negligible $f$ in $\kappa$):

$$
\begin{aligned}
u_i(\sigma'_i, \sigma_{-i}) &\le U_{\mathtt{exp2}} \\
&= \hat{U}_{\mathtt{exp\_2}} + f(\kappa) \\
&= U_{\mathtt{rand}} + \Pr_2[\overline{\mathtt{maul}}] \times [(\hat{U}_{\mathtt{exp\_1}|\overline{\mathtt{maul}}} - U_{\mathtt{rand}}) \\
&\quad + (U_{\mathtt{rand}} - \hat{U}_{\mathtt{exp\_1}|\overline{\mathtt{maul}}}) \times \Pr[\mathtt{short}|\overline{\mathtt{maul}}]] \\
&\quad + f(\kappa) \\
&= U_{\mathtt{rand}} + \Pr_2[\overline{\mathtt{maul}}] \times [(U - U_{\mathtt{rand}}) \\
&\quad + [(\beta \times U^+) + (1 - \beta) \times U_{\mathtt{rand}} - U] \\
&\quad\quad \times \Pr_2[\mathtt{abort}|\overline{\mathtt{maul}}] \\
&\quad + (U_{\mathtt{rand}} - \hat{U}_{\mathtt{exp\_1}|\overline{\mathtt{maul}}}) \times \Pr[\mathtt{short}|\overline{\mathtt{maul}}]] \\
&\quad + f(\kappa) \\
&= [U \times \Pr_2[\overline{\mathtt{maul}}] + U_{\mathtt{rand}} \times (1 - \Pr_2[\overline{\mathtt{maul}}])] \\
&\quad + \Pr_2[\overline{\mathtt{maul}}] \times [(B^- \times \Pr_2[\mathtt{abort}|\overline{\mathtt{maul}}]) \\
&\quad + (C^- \times \Pr[\mathtt{short}|\overline{\mathtt{maul}}])] + f(\kappa) \\
&\le U + f(k)
\end{aligned}
$$

where the last statement uses the following facts:

1. If $\Pr_2[\overline{\mathtt{maul}}] < 1$ we have $[U \times \Pr_2[\overline{\mathtt{maul}}] + U_{\mathtt{rand}} \times (1 - \Pr_2[\overline{\mathtt{maul}}])] < 0$ since $U_{\mathtt{rand}} < U$ by assumption.
2. $B^- := [(\beta \times U^+) + (1 - \beta) \times U_{\mathtt{rand}} - U] < 0$ by assumption.
3. $C^- := U_{\mathtt{rand}} - \hat{U}_{\mathtt{exp\_1}} < 0$ given that $\hat{U}_{\mathtt{exp\_1}}$ contains a $U$ term and $U_{\mathtt{rand}} < U$ by assumption.

This proves that $\Pi_1$ is a computational Nash equilibrium. To show that $\Pi_1$ is also an $(n-1)$-key leakage-tolerant equilibrium, we note that Lemmas A5, A4, A8, A11, and A12 used in the proof above hold even if a player acquires $n-1$ secret keys. For the round complexity, in each round of $\Pi_1$, each participant in the game can simultaneously send $k$ transmissions along $k$ disjoint paths to several other participants. Each transmission takes up to at most `max_l` rounds before it reaches its end-receiver. Phase 1 would then take up to `max_l` rounds, and each iteration in phase 2 takes up to `max_l` rounds. Given $\beta$, the expected value of $r^*$ is $1/\beta$, from which it follows that an average of up to $1/\beta + 1$ rounds will take place in phase 2, and we have that the average round complexity is $2 + 1/\beta$ rounds as stated. Finally, for the communication complexity, the largest amount of bits are communicated by the dealer during phase 1, which amounts to a total of $\nu \times (k + 2n + 1)$ per player. Since there are $n$ players, we have that the maximum amount of bits communicated in a single round would be at most $n \times \nu \times (k + 2n + 1)$, as stated.  $\square$

**Corollary A1.** *Let $p_i$ follow a strategy $\sigma'_i$ such that $\sigma'_i \sim \Pi_1(n, k)$, then $u_i(\sigma'_i, \sigma_{-i}) = U + f(\kappa)$ for some negligible function $f$ in $\kappa$.*

**Proof.** If $\sigma_i' \sim \Pi_1$, then from Lemmas A4 and A5, we have $\Pr[\overline{\texttt{maul}}]$ is equal to 1 with non-negligible probability and $\Pr[\texttt{abort} \vee \texttt{maul}]$ is negligible. The corollary follows from Theorem 1. $\square$

**Proof of Theorem 2.** To show that $\Pi_1$ is a strict Nash equilibrium, suppose that some player $p_i$ plays a polynomial-time strategy $\sigma_i' \neq \Pi$. From Lemmas A4 and A5, we have $\Pr[\texttt{abort} \vee \texttt{short}]$ occur with non-negligible probability, so that for some $c > 0$, we have $\Pr_0[\texttt{abort} \vee \texttt{short}] \geq 1/\kappa^c$ for infinitely many values of $\kappa$. Combining Lemmas A11 and A12, we have $\Pr_2[\texttt{abort} \vee \texttt{short}] \geq 1/\kappa^c$ and $\Pr_2[\texttt{abort} \vee \texttt{short}] \geq 1/\kappa^c$, as well for infinitely many values of $\kappa$. Using the same terms as in the proof for Theorem 1, this implies that $|\hat{U}_{\texttt{exp}_2} - U| \geq 1/\kappa^c$ for infinitely many values of $\kappa$. Given that $u_i(\sigma_i', \sigma_{-i}) \leq \hat{U}_{\texttt{exp}_2}|$ (since $\hat{U}_{\texttt{exp}_2}$ represents an upper bound for $u_i(\sigma_i', \sigma_{-i})$), we have $|u_i(\sigma_i', \sigma_{-i}) - U| \geq 1/\kappa^c$ for infinitely many values of $\kappa$, thereby proving the Theorem. $\square$

**Proof of Theorem 3.** To show that $\Pi_1$ is a $(k-1)$-resilient computational Nash equilibrium, we revise Lemmas A4 and A5 to the following versions that consider coalitions:

> **Coalition Version of Lemma A4**: Given $\Pi_1(n, k)$, denote by $\sigma$ the corresponding set of strategies prescribed by $\Pi_1$. Let $\mathcal{C}$ be a coalition of size at most $k - 1$, such that its members follow a set of polynomial-time strategies $\sigma_{\mathcal{C}} := \{\sigma_{p_i}'\}_{p_i \in \mathcal{C}}$ and let all other players follow $\sigma_{-\mathcal{C}}$. The event $\texttt{short}$ occurs due to $\mathcal{C}$ with non-negligible probability if: (1) some member of $\mathcal{C}$ aborts during phase 1; (2) a transmission originating from outside of $\mathcal{C}$ and is meant for $p_l \in N \setminus \mathcal{C}$ passes through a member $p_i \in \mathcal{C}$, such that $p_i$ does not follow the path encoding of the transmission; or (3) a member $p_i \in \mathcal{C}$ transmits as origin node the information $\{(\hat{g}_i, \hat{h}_i)\}_{i \in [n]}$ to another player $p_l \notin \mathcal{C}$ such that $\{(\hat{g}_i, \hat{h}_i)\}_{i \in [n]} \neq \{(g_i^*, h_i^*)\}_{i \in [n]}$. If $\texttt{short}$ occurs due to $\mathcal{C}$, we have $\sigma_{\mathcal{C}} \not\sim \Pi_1$.

> **Coalition Version of Lemma A5**: Given $\Pi_1(n, k)$, denote by $\sigma$ the corresponding set of strategies prescribed by $\Pi_1$. Let $\mathcal{C}$ be a coalition of size at most $k - 1$, such that its members follow a set of polynomial-time strategies $\sigma_{\mathcal{C}} := \{\sigma_{p_i}'\}_{p_i \in \mathcal{C}}$ and let all other players follow $\sigma_{-\mathcal{C}}$. The event $\texttt{abort}$ occurs due to $\mathcal{C}$ with non-negligible probability if, for some iteration $r \leq r^*$, any of the following occurs: (1) a member $p_i \in \mathcal{C}$ aborts before iteration $r^* + 1$; (2) a transmission originating from outside of $\mathcal{C}$ and is meant for player $p_l \notin \mathcal{C}$ as the end-receiver passes through some member $p_i \in \mathcal{C}$ such that $p_i$ does not follow the path encoding in the transmission; (3) a transmission originating from outside of $\mathcal{C}$ and is meant for player $p_l \notin \mathcal{C}$ as the end-receiver passes through some member $p_i \in \mathcal{C}$ such that $p_i$ sends a modified $h^\circ$ to $p_l \notin \mathcal{C}$ such that $h^\circ \neq h'$ (where $h'$ is from the dealer); (4) a transmission originating from outside of $\mathcal{C}$ and is meant for player $p_l \notin \mathcal{C}$ as the end-receiver passes through some member $p_i \in \mathcal{C}$ such that $p_i$ sends a modified $(\hat{g}_l, \hat{h}_l)$ to $p_l \notin \mathcal{C}$ such that $(\hat{g}_l, \hat{h}_l) \neq (g_l^r, h_l^r)$, or (5) given a member $p_i \in \mathcal{C}$ as the origin-node, $p_i$ sends $(\hat{g}_i, \hat{h}_i)$ to $p_l \notin \mathcal{C}$ such that $(\hat{g}_i, \hat{h}_i) \neq (g_i^r, h_i^r)$.

To prove the coalition versions above, we note that any coalition $\mathcal{C}$ has to be of size at most $k - 1$. From Lemma 1, this implies that for any distinct pair of players $(p_i, p_j)$, any set of $k$-disjoint paths from $p_i$ to $p_j$ has to contain a path that does not contain members of $\mathcal{C}$. This implies that any transmission from some player $p_i \notin \mathcal{C}$ to some other player $p_j \notin \mathcal{C}$, and which passes through some coalition member $p_l \in \mathcal{C}$, such that $p_l$ follows a strategy $\sigma_l' \not\sim \Pi_1$ (i.e., the situations in the above coalition versions of the Lemmas) will be discovered by $p_j$. This is because, as per Lemma 1, the transmission from $p_i$ to $p_j$ passes through one other path which does not contain members of $\mathcal{C}$, and so $p_j$ can use information from this transmission to perform checks against other transmissions that passed through members of $\mathcal{C}$. Using similar arguments as in the proofs of Lemmas A4 and A5, we prove their coalition versions above. Given these coalition versions of Lemmas A4 and A5 and the fact that, given a coalition $\mathcal{C}$ of size at most $k - 1$, the results of Lemmas A8, A11, and A12

hold (given that only up to $k-1$ secret keys can be shared by members of $\mathcal{C}$), the above Theorem follows using a similar proof as in Theorem 1. $\square$

**Appendix G. Technical Results for Protocol $\Pi_2$**

**Lemma A13.** *Given $\Pi_2(n,k)$, denote by $\sigma$ the corresponding set of strategies prescribed by $\Pi_2$. Let $p_i$ follow some polynomial-time strategy $\sigma_i'$ and let all other players follow $\sigma$. The event* `short` *occurs due to $p_i$ with non-negligible probability if: (1) $p_i$ follows $\sigma_i'$ such that* `transmission_ordering_a` *is not followed in phases 1 and 2, or if (2) $p_i$ under $\sigma_i'$ sent an incorrect message that does not match $\Psi$ in phase 1. If* `short` *occurs due to $p_i$, we have $\sigma_i' \not\sim \Pi_2$.*

**Proof.** If $p_i$ itself aborts in phase 1 or in phase 2, then it does not follow the transmission scheme in `transmission_ordering_a` and `short` occurs. If some other player $p_j$ with $i \neq j$, $j \in [n]$ aborts in phases 1 and 2, this event happens if (1) $p_i$ sends a transmission that does not match `transmission_ordering_a` or (2) $p_i$ sends $\Psi' \neq \Psi$ to some other player. If $\Psi' \neq \Psi$ is sent by $p_i$, this will be noticed by some other player given that the other player receives $k-1$ other copies of $\Psi$ according to `transmission_ordering_a` and the other player aborts as required by $\Pi_2$. For the last statement of the Lemma, we have $\sigma_i' \not\sim \Pi_2$ if $\mathrm{view}^{\sigma_i', \Pi_2}$ deviates from $\mathrm{view}^{\Pi_2}$. If $p_i$ does not follow `transmission_ordering_a` or sends $\Psi' \neq \Psi$ in phase 1, then at least one other player $p_j$ notices this. These events imply that $\mathrm{view}^{\sigma_i', \Pi_2} \neq \mathrm{view}^{\Pi_2}$ in the relevant parts involving phases 1 and 2 and $\sigma_i' \not\sim \Pi_2$. $\square$

**Lemma A14.** *Given $\Pi_2(n,k)$, denote by $\sigma$ the corresponding set of strategies prescribed by $\Pi_2$. Let $p_i$ follow some polynomial-time strategy $\sigma_i'$ and let all other players follow $\sigma$. The event* `abort` *occurs due to $p_i$ with non-negligible probability if: (1) $p_i$ follows $\sigma_i'$ such that* `transmission_ordering_b` *is not followed in phase 3 or (2) if $p_i$ sends an incorrect message $(y_i', z_i', \pi_i', \psi_i')$ such that $(y_i', z_i', \pi_i', \psi_i') \neq (y_i^r, z_i^r, \pi_i^r, \psi_i^r)$ for some iteration $r$. If* `abort` *occurs due to $p_i$, we have $\sigma_i' \not\sim \Pi_2$.*

**Proof.** If $p_i$ itself aborts before iteration $r^* + 1$, then it does not follow the transmission scheme in `transmission_ordering_b` and `abort` occurs. If $p_i$ sent correct messages with respect to $(y_i^r, z_i^r, \pi_i^r, \psi_i^r)$ for each iteration $r$, but some other player $p_j$ ($i \neq j$) aborts, this is due to $p_i$ sending a transmission that does not match `transmission_ordering_b` (given that all other players follow $\Pi_2$). If $p_i$ follows `transmission_ordering_b` but some other player $p_j$ ($i \neq j$) aborts, this could only be due to $p_i$ sending a transmission $(y_i', z_i', \pi_i', \psi_i') \neq (y_i^r, z_i^r, \pi_i^r, \psi_i^r)$ for some iteration $r \leq r^*$, which is detected by $p_j$ using the VRF. This is because $\Pi_2$ prescribes that a *unique* $(y_i', z_i', \pi_i', \psi_i')$ be sent by each player at each iteration—using the VRF's properties in Definition A6. For the last statement of the Lemma, we have $\sigma_i' \not\sim \Pi_2$ if $\mathrm{view}^{\sigma_i', \Pi_2}$ deviates from $\mathrm{view}^{\Pi_2}$. If $p_i$ does not follow `transmission_ordering_b` or sends an incorrect transmission for iteration $r \leq r^*$ in phase 3, then at least one other player $p_j$ notices this. These events imply that $\mathrm{view}^{\sigma_i', \Pi_2} \neq \mathrm{view}^{\Pi_2}$ in the relevant parts involving phase 3 and $\sigma_i' \not\sim \Pi_2$. $\square$

Given these terminologies, we state the following Lemmas and definitions.

**Definition A11.** *We now define the following experiments. Let $\sigma_i'$ denote any arbitrary polynomial-time strategy of $p_i$. Define $\Pr_0$ as the probabilities in* `Exp 0`, *by $\Pr_1$ the probabilities in* `Exp 1`, *and by $\Pr_2$ the probabilities in* `Exp 1`.

- ***Exp 0****: This experiment runs $\Pi_2(n,k)$ but with $p_i$ following $\sigma_i'$ and the rest following the prescribed strategies $\Pi_2$. In addition, $p_i$ can acquire $n-1$ secret key pairs $(sk_{i_l}, sk_{i_l}')_{l \in [n]-1}$.*

- ***Exp 1****: This experiment is the same as Exp 0, except that in the initialization phase (i.e., phase 0), the dealer computes $g_i^* = G(i) \oplus v_1$ and $h_i^* = H(i) \oplus v_2$, where $v_1$ and $v_2$ are uniformly sampled from the range of $V_E$.*

***Exp 2***: *This experiment is the same as Exp 1, except that, given the k disjoint paths in phase 2, where $p_i$ receives shares $\{s_{i,1}, s_{i,2}, \ldots, s_{i,k}\}$ and $\{s'_{i,1}, s'_{i,2}, \ldots, s'_{i,k}\}$ to reconstruct $sk_i$ and $sk'_i$, respectively, one pair $(s_{i,j}, s'_{i,j})$ for some $j \in [k]$ is replaced by the dealer in phase 0 with a uniformly sampled pair of numbers in the range of $S_G$.*

**Lemma A15.** *Under $\Pi_2(n, k)$, suppose that $p_i$ deviated and acquired $n - 1$ secret key pairs $(sk_{i'}, sk'_{i'})$ for $i' \in [n]$. Given information $\phi_i(I)$ from any information set $I$ in either phase 1 or any iteration $r \leq r^*$ in phase 2 of $\Pi_2(n, k)$, the distribution of $h_i^*$ and $g_i^*$ for any $i \in [n]$ and the distribution of $H$ and $G$ are indistinguishable from random. In addition, the probability of guessing $r^*$ is $\beta$.*

**Proof.** Given that $V_E$ is pseudorandom, the same proof as that for Lemma A6 would hold word for word after making the appropriate substitutions, i.e., (i) changing $\Lambda$ to $V_E$; (ii) changing from using $sk_i$ for both $g_i^r$ and $h_i^r$ to using $sk_i$ for $g_i^r$ and $sk'_i$ for $h_i^r$; and (iii) adjusting the phase numbers from phase 1 in $\Pi_1$ to phases 1–2 in $\Pi_2$ and from phase 2 in $\Pi_1$ to phase 3 in $\Pi_2$.   □

**Lemma A16.** *Suppose that under $\Pi_2(n, k)$, $p_i$ deviated and acquired $n - 1$ secret key pairs $(sk_{i'}, sk'_{i'})$. If* `maul` *occurred in phase 1 due to $p_i$, the probability of* `true(i)` *and* `true(-i)` *is negligible at any phase.*

**Proof.** Given that $V_E$ is pseudorandom, the same proof as that for Lemma A8 for phase 1 would hold word for word after making the appropriate substitutions, i.e., (i) changing $\Lambda$ to $V_E$; (ii) changing from using $sk_i$ for both $g_i^r$ and $h_i^r$ to using $sk_i$ for $g_i^r$ and $sk'_i$ for $h_i^r$; and (iii) adjusting the phase numbers from phase 1 in $\Pi_1$ to phases 1–2 in $\Pi_2$ and from phase 2 in $\Pi_1$ to phase 3 in $\Pi_2$. For phase 2, if `maul` occurred, the secret key pair $(sk_j, sk'_j)$ of some player $p_j \in N$ is modified to $(\hat{sk}_j, \hat{sk}'_j) \neq (sk_j, sk'_j)$ so that $p_j$ computes:

$$(\hat{y}_i^r = V_E(\hat{sk}_i, r), \hat{z}_i^r = V_E(\hat{sk}'_i, r),$$
$$\hat{\pi}_i^r = V_P(\hat{sk}_i, r), \hat{\psi}_i^r = V_P(\hat{sk}'_i, r))$$

By the properties of the VRF (Definition A6), this implies that with non-negligible probability, we have $V_V(pk_j, r, \hat{y}_i^r, \hat{\pi}_i^r) \neq$ `true` and $V_V(pk'_j, r, \hat{z}_i^r, \hat{\psi}_i^r) \neq$ `true` since, with non-negligible probability, $pk_j \neq \hat{pk}_j$, where $\hat{pk}_j$ denotes the correct public key paired by $V_G$ given a secret key $\hat{sk}_j$. Given that the prior checks would fail for all players, it follows that, with non-negligible probability, under $\Pi_2$, players abort and output a guess for $s$ from the uniform distribution. This proves the Lemma.   □

**Lemma A17.** *Given $\Pi_2(n, k)$, for any polynomial-time strategy $\sigma'_i$ adopted by $p_i$, there exists a negligible function $f$ in $\kappa$ such that we have the following, given a fixed* `stat` $\in \{$`maul`, $\overline{\text{maul}}\}$ *for each statement. This result holds even if $\sigma'_i$ led $p_i$ to acquire less than n secret key pairs:*

1. $|\Pr_0[\text{short}] - \Pr_1[\text{short}] \leq f(\kappa)$
2. $|\Pr_0[\text{exact} \wedge \text{true(i)}] - \Pr_1[\text{exact} \wedge \text{true(i)}]| \leq f(\kappa)$
3. $|\Pr_0[\text{exact} \wedge \overline{\text{true(i)}}] - \Pr_1[\text{exact} \wedge \overline{\text{true(i)}}]| \leq f(\kappa)$
4. $|\Pr_0[\text{late}] - \Pr_1[\text{late}]| \leq f(\kappa)$
5. $|\Pr_0[\text{early} \wedge \text{true(i)}] - \Pr_1[\text{early} \wedge \text{true(i)}]| \leq f(\kappa)$
6. $|\Pr_0[\text{early} \wedge \overline{\text{true(i)}}] - \Pr_1[\text{early} \wedge \overline{\text{true(i)}}]| \leq f(\kappa)$

**Proof.** Given that $V_E$ is pseudorandom and $sk_n$ is unknown for $p_i$, the same proof as that for Lemma A11 would hold word for word after making the appropriate substitutions, i.e., (i) changing $\Lambda$ to $V_E$; (ii) changing from using $sk_i$ for both $g_i^r$ and $h_i^r$ to using $sk_i$ for $g_i^r$ and $sk'_i$ for $h_i^r$; and (iii) adjusting the phase numbers from phase 1 in $\Pi_1$ to phases 1–2 in $\Pi_2$ and from phase 2 in $\Pi_1$ to phase 3 in $\Pi_2$.   □

**Lemma A18.** *Given* $\Pi_2(n, k)$, *for any polynomial-time strategy* $\sigma'_i$ *adopted by* $p_i$, *there exists a negligible function* $f$ *in* $\kappa$ *such that we have the following, given a fixed* stat $\in \{$maul$, \overline{$maul$}\}$ *for each statement. This result holds even if* $\sigma'_i$ *led* $p_i$ *to acquire less than n secret keys:*

1.  $|\text{Pr}_1[\text{short} \wedge \text{true(i)}] - \text{Pr}_2[\text{short} \wedge \text{true(i)}] \leq f(\kappa)$
2.  $|\text{Pr}_1[\text{short} \wedge \overline{\text{true(i)}}] - \text{Pr}_2[\text{short} \wedge \overline{\text{true(i)}}] \leq f(\kappa)$
3.  $|\text{Pr}_1[\text{exact} \wedge \text{true(i)}] - \text{Pr}_2[\text{exact} \wedge \text{true(i)}]| \leq f(\kappa)$
4.  $|\text{Pr}_1[\text{exact} \wedge \overline{\text{true(i)}}] - \text{Pr}_2[\text{exact} \wedge \overline{\text{true(i)}}]| \leq f(\kappa)$
5.  $|\text{Pr}_1[\text{late}] - \text{Pr}_2[\text{late}]| \leq f(\kappa)$
6.  $|\text{Pr}_1[\text{early} \wedge \text{true(i)}] - \text{Pr}_2[\text{early} \wedge \text{true(i)}]| \leq f(\kappa)$
7.  $|\text{Pr}_1[\text{early} \wedge \overline{\text{true(i)}}] - \text{Pr}_2[\text{early} \wedge \overline{\text{true(i)}}]| \leq f(\kappa)$

**Proof.** Given that $V_E$ is pseudorandom and $sk_n$ is unknown by $p_i$, the same proof as that for Lemma A12 would hold word for word after making the appropriate substitutions, i.e., (i) changing $\Lambda$ to $V_E$; (ii) changing from using $sk_i$ for both $g_i^r$ and $h_i^r$ to using $sk_i$ for $g_i^r$ and $sk'_i$ for $h_i^r$; and (iii) adjusting the phase numbers from phase 1 in $\Pi_1$ to phases 1–2 in $\Pi_2$ and from phase 2 in $\Pi_1$ to phase 3 in $\Pi_2$. $\square$

**Proof of Theorem 4.** To prove Theorem 4, we note that the results of Lemmas A4 and A5 for $\Pi_1$ have their equivalent in Lemmas A13 and A14 for $\Pi_2$. The results of Lemmas A11 and A12 for $\Pi_1$ have their equivalent as well in Lemmas A17 and A18 for $\Pi_2$. Given that $V_E$ also has the pseudorandom property similar to $\Lambda$, the proof for Theorem 4 holds word for word for Theorem 4 after making the appropriate substitutions as were performed in the proof of Lemmas A17 and A18. $\square$

**Proof of Theorem 5.** To prove the theorem, we note that $\Pi_2$ prescribes that a *unique* transmission be sent by a *unique* player at each round as shown in Lemmas A14 and A13. The proof for the above Theorem follows that of Theorem 2 by substituting Lemmas A4 and A5 for $\Pi_1$ to their equivalent Lemmas A13 and A14 for $\Pi_2$,and substituting Lemmas A11 and A12 for $\Pi_1$ to their equivalent Lemmas A17 and A18 for $\Pi_2$.

For the average round complexity, we note that phases 1 and 2, take up at most $2 \times$ max_l $\times n \times k$ rounds, given that the dealer sends $\Psi$ to $n$ players along $k$ disjoint paths and that the maximum length of a path is at most max_l. In phase 3, the average value of $r^*$ is $1/\beta$, so that an average of $1 + 1/\beta$ iterations take place under $\Pi_2$. Each iteration in turn takes up at most max_l $\times n^2 \times k$, as each player sends to each other player a transmission along a path of length at most max_l. Finally, the largest communication in a round takes place in phase 1, when the dealer sends $\Psi$ to each participant, which takes up $O(4n\nu)$ bits, which may differ by a constant factor per graph $G$ due to bits taken up by transmission_ordering_a and transmission_ordering_b—both of which depend on the size of $G$. $\square$

**Proof of Theorem 6.** Coalition versions of Lemmas A13 and A14, as performed in the proof of Theorem 3 for Lemmas A4 and A5, can be constructed here using the fact that $\Pi_2$ prescribes that a *unique* transmission be sent by a *unique* player at each round as shown in Lemmas A14 and A13. It follows that any transmission sent by a member of $\mathcal{C}$ to another player outside of $\mathcal{C}$ would be readily checked for deviations from $\Pi_2$. Using these coalition versions of Lemmas A14 and A13, along with the fact that Lemmas A8, A17, and A18 hold in a coalition of size at most $k - 1$ (given that a $k - 1$-sized coalition may only share up to $k - 1$ secret keys among its members), the Theorem follows. $\square$

**Proof of Theorem 7.** From the assumptions of the theorem, each player has acquired the public information $\Psi$ and the pair of secret keys directly from the dealer, and each player has the correct copy of transmission_ordering_b. We note that the difference between protocol $\Pi_2$ and $\Pi_{2.1}$ is that for every transmission in $\Pi_{2.1}$, each node through which the transmission passes checks the correctness of the transmission using the VRF algorithm $V_V$. Given that this is a stronger requirement than $\Pi_2$ (where only end-nodes of the transmission

check for correctness), the results of Lemma A14 readily apply. In addition, the condition in $\Phi$ (where a coalition should be 1-disconnected) implies that for each pair of members $p_i, p_j \in \mathcal{C}$, any transmission from $p_i$ to $p_j$ has to pass through players that are not in $\mathcal{C}$. It follows that all transmissions among members of $\mathcal{C}$ are checked for correctness, and they cannot include additional information in their transmission. In particular, members of $\mathcal{C}$ cannot transmit secret keys to each other as this will violate the VRF checks, and players are constrained to have only 1 secret key, and Lemmas A17 and A18 apply. It follows that all players strictly conform to the strategies prescribed by $\Pi_{2.1}$, and given Lemmas A14, A12, and A18, we apply the same proof as in Theorem 4 to prove the Theorem above. □

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
