# Peer review of "Cryptographic Rational Secret Sharing Schemes over General Networks"

_cryptography, doi:10.3390/cryptography6040050_

Round 1

Reviewer 1 Report

The paper revisits rational (k,n)-secret sharing. On a high level, the utility functions of the players increases when they learn the correct secret, and decreases when others learn the secret. Such schemes are known to exist for n>2. The main focus of the paper is extending the network over which the scheme is implemented, allowing to only send messages between certain pairs of parties (rather than having a link between each pair of parties).The parties are assumed to be computationally bounded.

They obtain several positive results, obtaining sufficient conditions for the topology of networks over which such schemes exist.

=============

The paper appears solid, and its contribution touches a useful extension of an interesting area of rational secret sharing (as a sub-topic) . A slight downside, which I expect to be discussed is why only that particular topology is handled. Are there possible extensions you have in mind. However, this is not a major issue.

More crucially I had trouble understanding much of the details in this work due to several presentation issues discussed below.

1. There is not enough intuition provided on definitions as constructions before the full description is provided. This makes it quite difficult to parse the results. Especially given some typos and inaccuracies. For example, see definition 32, properties 2,3, in which the same text appears by mistake.

Another example is definition 12. U,U- are defined as numbers (rather than functions), however in both cases there is a set of o's that determine them. Do you implicitly assume that the mu_i(o)'s are the same for all such inputs (corresponding to some U_i, and to U-)? If so, this should be stated explicitly.

2. Another aspect of missing high level description is in comaprison of techniques to prior work. You mention [25,24] as handling incomplete networks and for non rational secret sharing and [9] as handling rational secret sharing in the computational setting. [8] Handles asynchronous networks, as do you (which are not rational). It is very interesting to know whether and how you used some of the approaches of these works in your solution on a high level. It seems like you did use ideas from there (e.g VRF from [8], network topology in [25]). It is crucial to understand what are the challenges you faced ? Why don't the results follow "smoothly" by combining techniques from previous work, so the reviewers can better understand your contribution.

3. Non-standard notation and use of primitives. For instance, in (n,k) Shamir secret sharing, you only seem to use it as (k,k) secret sharing. This is confusing. Why not just use additive secret sharing here?

4. The text is rather lengthy, with many definitions that seemingly can be avoided, as they are standard (or better still, put in an appendix) .One example is PRF. This could also make the paper more readable.

Additional concrete comments:

*Line 582, "and for j in [k]" seems redundant.

* Definition 7: "k-path disjoiont" should say the paths are pairwise disjoint. It was not clear to me until I read the proof of Lemma 1.

* Example of k - disj
graphs could be useful. One that comes
to mind are very dense graphs,
such as Cliques, where n > k+1. Do you have any other interesting instances in mind?

* Say explicitly which assumptions about U,U+,U- if any come into play here.

* In your Nash equilibrium, you consider deviations from the protocol both of the type of sending messages that deviate from the protocol, or aborting altogether. It is not clear why abort can not be modeled as sending "garbage" messages, and is instead modeled separately. It could be useful if the number of rounds in the protocol is not fixed apriori. However, it is not clear whether this is the case.

* Some of the standard notions, such as PRF are better briefly recalled in the main text (saying also a standard notion of PRF is used), and put in the appendix for completeness.

Author Response

Thank you for your helpful comments. Please see the attachment for our responses.

Reviewer 2 Report

This paper presents multiple cryptographic rational secret sharing schemes over general networks where the dealer may not have direct connections to each player and players may not have direct connections to each of the other players. The paper also proposes new equilibria concepts for such secret sharing problems. 

The problems and solutions seem valid. However, my main concern is regarding the organization of the paper as in this format it is extremely hard to follow it. 

There are too many definitions! I would suggest remove some of the unnecessary definitions and include them inside the manuscript following the flow of the paper. For example it seems to me that def 21 and 24 can be merged into one definitions. 

I suggest adding a conclusion section to briefly discuss the result of the paper. 

It would be helpful to add a figure to demonstrate the problem and/or proposed protocols. 

The contribution section needs to be explained clearer. Specifically, since this work has similarities with [8] and [25], I suggest discuss the differences in more details.  

Author Response

Thank you for your comments. Please see the attachment for our responses to your comments.
